# RNA-mediated symmetry breaking enables singular olfactory receptor choice

Ariel D. Pourmorady[1,2,3], Elizaveta V. Bashkirova[3,4], Andrea M. Chiariello[5], Houda Belagzhal[6], Albana Kodra[4,7], Rachel Duffié[3], Jerome Kahiapo[8], Kevin Monahan[8], Joan Pulupa[3], Ira Schieren[3], Alexa Osterhoudt[3,4], Job Dekker[6], Mario Nicodemi[5] & Stavros Lomvardas[3,9 ✉]

Olfactory receptor (OR) choice provides an extreme example of allelic competition for transcriptional dominance, where every olfactory neuron stably transcribes one of approximately 2,000 or more OR alleles[1,2]. OR gene choice is mediated by a multichromosomal enhancer hub that activates transcription at a single OR[3,4], followed by OR-translation-dependent feedback that stabilizes this choice[5,6]. Here, using single-cell genomics, we show formation of many competing hubs with variable enhancer composition, only one of which retains euchromatic features and transcriptional competence. Furthermore, we provide evidence that OR transcription recruits enhancers and reinforces enhancer hub activity locally, whereas OR RNA inhibits transcription of competing ORs over distance, promoting transition to transcriptional singularity. Whereas OR transcription is sufficient to break the symmetry between equipotent enhancer hubs, OR translation stabilizes transcription at the prevailing hub, indicating that there may be sequential non-coding and coding mechanisms that are implemented by OR alleles for transcriptional prevalence. We propose that coding OR mRNAs possess non-coding functions that influence nuclear architecture, enhance their own transcription and inhibit transcription from their competitors, with generalizable implications for probabilistic cell fate decisions.

To interact with their environment, cells express diverse receptors that detect chemicals, antigens, photons, heat, magnetic and electric fields, or mechanical stimulation. To perceive the identity and valence of these signals and to elicit appropriate responses, most organisms deploy a 'one receptor type per cell' rule[7], which restricts the cellular receptive field to receptor-specific cues. This recurrent design poses a regulatory challenge, as cells must express one of many receptor genes with similar regulatory sequences that are all transcribed in the same cell type. Lymphocytes solve this problem by VDJ recombination and photoreceptor neurons by placing two mutually exclusive opsin genes in the X chromosome, whereas other cell types have evolved tailored solutions for transcriptional singularity[8]. Among these, olfactory sensory neurons (OSNs) face the most extreme challenge, as they stably express one olfactory receptor (OR) from more than approximately 1,000 available genes in a monogenic and monoallelic fashion[1,9,10]. OR expression in mature OSNs (mOSNs) requires genomic interactions between the active OR allele and an intrachromosomal and interchromosomal network of 63 OR gene-specific enhancers[11] called Greek islands (GIs)[3]. These DNA elements are held together by transcription factors EBF1 and LHX2 and the coactivator LDB1, forming a nucleoprotein complex, the GI hub, which is essential for OR transcription[3,4,12]. Whereas bulk Hi-C experiments indicate that large numbers of GIs associate specifically with the active OR, single-cell Hi-C (Dip-C) has revealed the existence of multiple GI hubs per OSN[13]. Moreover, single-cell RNA sequencing (scRNA-seq) experiments uncovered transient OR co-expression in OSN progenitors, in contrast to the singular OR transcription of mOSNs[14–16]. Together, these observations indicate that differentiating OSNs may have the regulatory capacity for polygenic OR transcription, yet they eventually transition to absolute transcriptional singularity under unknown regulatory mechanisms.

## GI accessibility changes with neuronal differentiation and genomic compartmentalization

To identify genomic changes occurring during the transition from polygenic to singular OR transcription, we performed single-nucleus ATAC-seq (assay for transposase-accessible chromatin using sequencing) and RNA-seq with 10x Genomics, generating a multiome of the main olfactory epithelium (MOE). Data were aligned and processed with Cell Ranger and analysed using the R packages Seurat and Signac[17]. Cells were clustered using combined accessibility and gene expression data by weighted nearest neighbours analysis[18] and visualized by UMAP projection. Various cell populations could be identified, including the neuronal lineage, which contains globose basal cells (GBCs),

[1]Vagelos College of Physicians and Surgeons, Columbia University New York, New York, NY, USA. [2]Department of Neuroscience, Columbia University, New York, NY, USA. [3]Mortimer B. Zuckerman Mind, Brain, and Behavior Institute, Columbia University New York, New York, NY, USA. [4]Integrated Program in Cellular, Molecular and Biomedical Studies, Vagelos College of Physicians and Surgeons, New York, NY, USA. [5]Department of Physics 'Ettore Pancini', University of Naples, and INFN, Napoli, Italy. [6]Department of Biochemistry and Molecular Pharmacology, University of Massachusetts Medical School, Worcester, MA, USA. [7]Department of Genetics and Development, Vagelos College of Physicians and Surgeons, New York, NY, USA. [8]Department of Molecular Biology & Biochemistry, Rutgers School of Arts and Sciences, Robert Wood Johnson Medical School, Piscataway, NJ, USA. [9]Department of Biochemistry and Molecular Biophysics, Vagelos College of Physicians and Surgeons, New York, NY, USA. ✉e-mail: sl682@cumc.columbia.edu

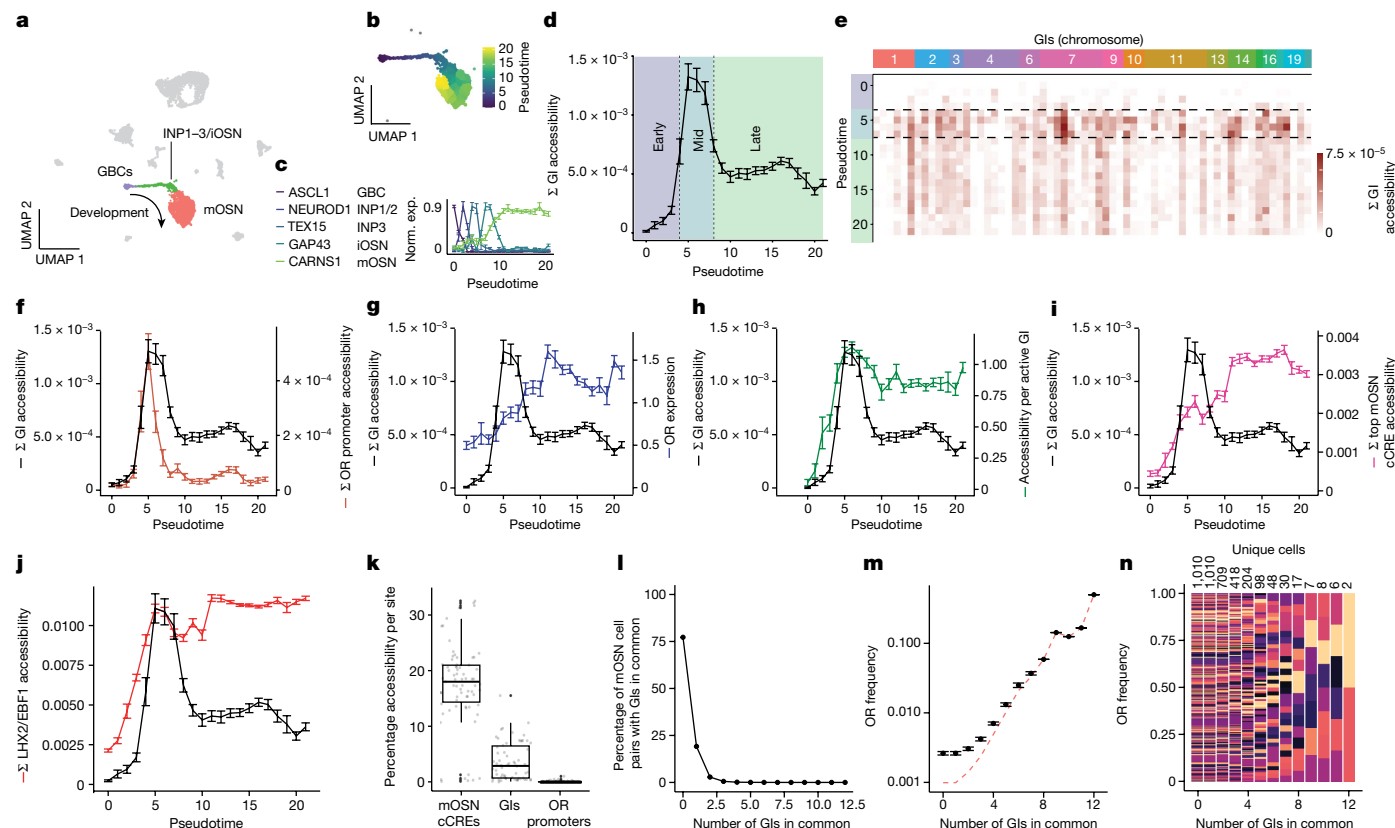

**Fig. 1 | Combined single-nucleus ATAC and single-nucleus RNA-seq uncover gradual GI inactivation. a**, UMAP of the mouse MOE multiome, constructed from weighted nearest neighbours analysis on RNA and ATAC data from 6,497 quality-controlled cells from one mouse (see Extended Data Fig. 1 for an independent replication). The neuronal lineage contains GBCs, INPs, iOSNs and mOSNs. **b**, UMAP projection of the neuronal lineage with cells coloured by pseudotime. **c**, Verification of pseudotime projection using known markers (scaled mean ± s.e.m., $n = 6,497$ cells from one multiome). **d**, Cumulative GI accessibility dynamics, averaged for all cells rounded to the nearest pseudotime, separated into three phases: early (GBC–INP3), mid (INP3–iOSN) and late (mOSN) (mean ± s.e.m. per pseudotime, $n = 2,371$ cells from one multiome). **e**, Individual GI accessibility over pseudotime. **f**, Cumulative GI (black) ($n = 63$ sites, mean ± s.e.m.) and OR promoter (cinnamon) accessibility over pseudotime ($n = 1255$ sites, mean ± s.e.m.). **g**, Expression levels of the most highly expressed OR per cell (blue, sctransform (SCT) normalized counts, mean ± s.e.m.).

**h**, Accessibility per active GI (green, mean ± s.e.m.). **i**, Top mOSN cCREs (pink, $n = 71$ sites, mean ± s.e.m.). **j**, Cumulative accessibility of LHX2 and EBF1 cCREs in mOSNs (red, $n = 4793$ sites, mean ± s.e.m. per pseudotime). **k**, Box plots comparing fraction of cells with accessibility in mOSN cCREs (17.4% ± 7.43%, $n = 71$), GIs (3.64% ± 3.40%, $n = 63$) and OR promoters (1.83 × 10$^{-2}$% ± 5.15 × 10$^{-2}$%, $n = 1,255$) in mOSNs. Each box plot ranges from the upper to lower quartiles with the median as the horizontal line, and whiskers extend to 1.5 times the interquartile range. **l**, A total of 509,545 unique cell pairs, from 1,010 mOSNs, were tested for accessible GIs. **m**, Average frequency of an OR (mean ± s.e.m.) between all unique cells comprising cell pairs sharing between 0 and 12 GIs. The dashed red line represents the expected average OR frequency if the number of unique ORs were to equal the number of unique cells ($n = 1,010$ cells from one multiome). **n**, Frequency of each OR expressed by all unique cells making up cell pairs sharing between 0 and 12 GIs, coloured by OR identity. Norm. exp., normalized expression.

immediate neuronal precursors (INPs), immature OSNs (iOSNs) and mOSNs[19] (Fig. 1a and Extended Data Fig. 1a). We built a pseudotime trajectory of the neuronal lineage (Fig. 1b,c and Extended Data Fig. 1b,c) and measured cumulative GI and OR promoter accessibility for each developmental stage[20]. GIs and OR promoters are initially inaccessible, and their accessibility gradually increases from GBC/early INP to late INP/iOSN, coinciding with transcriptional onset of LHX2 and EBF1 (Extended Data Fig. 2a–c). At the terminal stages of differentiation, during the transition from the iOSN to mOSN stage, GI and OR promoter accessibility both decline sharply (Fig. 1d–f and Extended Data Fig. 1d–f). This decline represents a selective inactivation of most but not all GIs and OR promoters (Fig. 1f,h,k and Extended Data Fig. 1f,h,k), and coincides with transition to robust and singular OR transcription (Fig. 1g and Extended Data Fig. 1g). On the other hand, candidate *cis* regulatory elements (cCREs) for other OSN-expressed genes are 4.8 times more accessible than GIs and preserve their cumulative accessibility in mOSNs, an observation that extends to non-GI cCREs bound by LHX2 and EBF1[12] (Fig. 1i–k and Extended Data Figs. 1i,k and 2d,e).

The rarity of GI activation provides a potential framework for OR selection, where unique combinations of infrequently accessible GIs

could determine the expression of each of the approximately 1,000 OR genes. To explore this, we measured the overlap in OR expression among mOSNs sharing varying degrees of similarity in accessible GIs. As expected, among 509,545 unique mOSN cell pairs, 75% did not share a single common accessible GI, and there was a rapid decline in the number of OSN pairs that shared increasing numbers of common accessible GIs (both *cis* and *trans*) (Fig. 1l and Extended Data Fig. 1l). However, increasing GI overlap did not correlate with increased matching of ORs between 'like' OSN populations (Fig. 1m and Extended Data Fig. 1m). Among OSN pairs that shared up to 12 common active GIs, OR complexity was never reduced below the number of unique cells (Fig. 1n and Extended Data Fig. 1n). Thus, with the caveat that single-cell ATAC (scATAC) is vulnerable to read dropout, these data, at this coverage, indicate that GI accessibility patterns do not correlate with the identity of the chosen OR.

## Dip-C shows distinct features between active and inactive GI hubs

To examine how the differentiation-dependent pruning of accessible GIs influences the assembly of a transcriptionally engaged GI

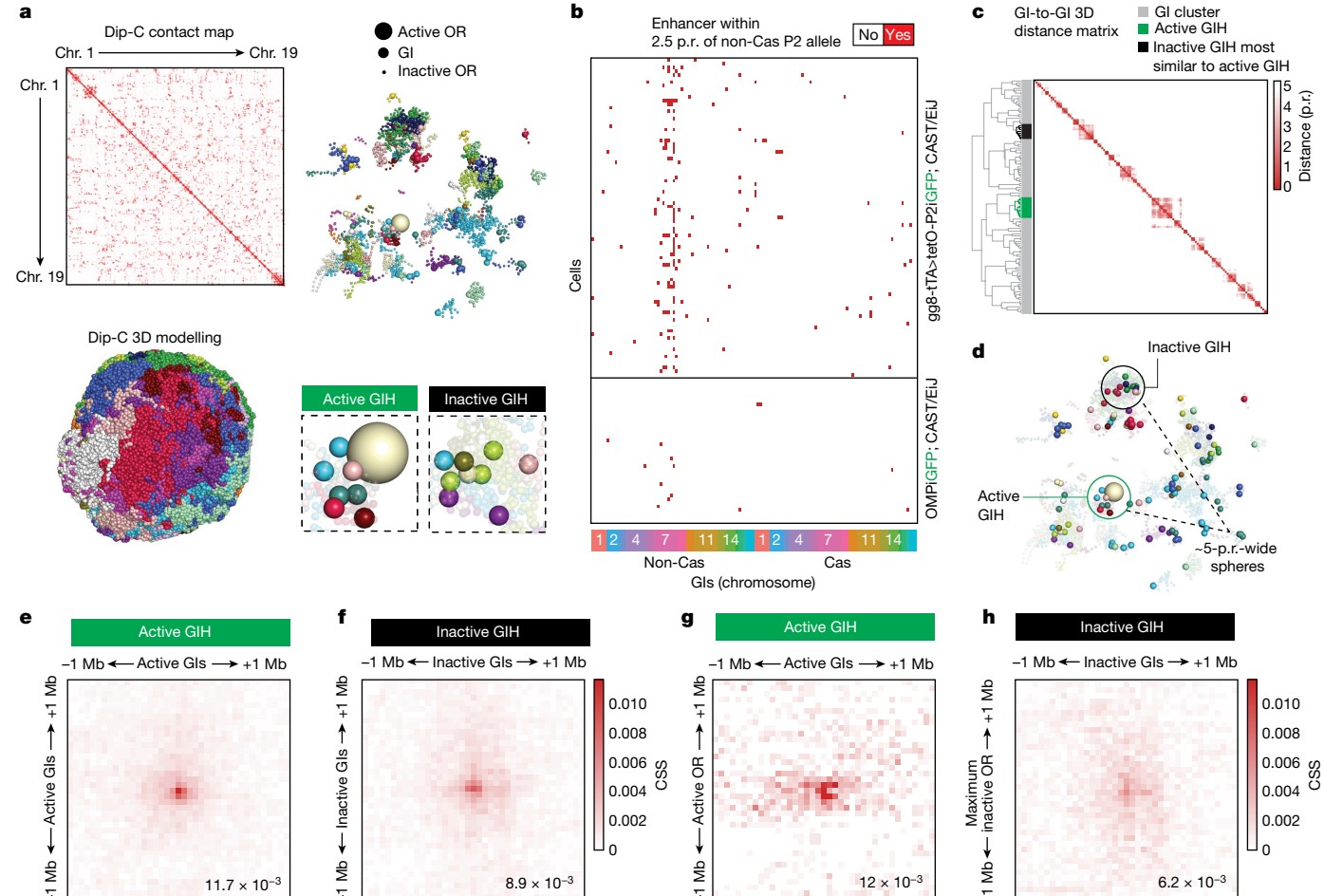

**Fig. 2 | Dip-C reveals differences between active and inactive GI hubs.**
**a**, Dip-C was performed on 161 FAC-sorted GFP⁺ nuclei from *Mor28iGFP* and
*gg8-tTA>tetOP-2iGFP* mice. Three-dimensional models of individual nuclei were
rendered using haplotype-imputed contact maps (top left) and coloured by
chromosome (bottom left). A representative *Mor28iGFP* nucleus stripped of all
genes except inactive ORs (small spheres), the active OR allele (large sphere)
and GIs (medium spheres) is shown, coloured by chromosome (top right). The
active *Mor28* allele was near a cluster of GIs, marking the active hub (bottom
left). Further GI clusters were also detected (bottom right). **b**, Binary arrays
were generated for each cell (rows) and depict GIs within less than 2.5 p.r. of the
P2 allele (non-Cas, chr7) when it is active (top, *gg8-tTA>tetOP-2iGFP* Dip-C, *n* = 87
cells; see Extended Data Fig. 5b for *Mor28iGFP* Dip-C) or inactive (bottom,
*OMPiGFP*, *n* = 40 cells from publicly available data). **c**, Hierarchical clustering of

GI spatial relationships in a P2⁺ nucleus. The dendrogram was cut at 2.5 p.r.
The active GI hub is shown in green and the inactive hub of the most similar size
in black. **d**, Dip-C model depicting the topology of active and inactive hubs
selected for contact analysis. **e**–**h**, Heatmaps of interchromosomal contacts
made between GIs in the active hub (**e**; CSS = 0.0117, *n* = 117 cells with contacts
pooled from two independent experiments); GIs in the inactive hub (**f**; CSS =
0.00891, *n* = 109 cells with contacts pooled from two independent experiments);
GIs and the active OR (**g**; CSS = 0.0120, *n* = 70 cells with contacts pooled from
two independent experiments); and GIs and the maximally engaged inactive
OR in the inactive hub (**h**; CSS = 0.00621, *n* = 106 cells with contacts pooled
from two independent experiments). CSS was measured by dividing the contacts
in any 50 kb bin by the sum of all contacts in the 2 × 2 Mb square, where the CSS
at the focus of interaction is noted in the bottom right corner. GIH, GI hub.

hub, we performed Dip-C[21] on OSNs expressing a known OR allele.
We sorted OSNs from the progeny of *Mor28iGFP* and *gg8-tTA>tetO-P2*
mice crossed to Castaneous (Cas) mice, where F₁ hybrids would have
known single-nucleotide polymorphisms[22–25] (Extended Data Fig. 3).
MOR28 (encoded by *Or4e5*, also known as *Mor28*) is one of the most
frequently chosen ORs, whereas *gg8-tTA>tetO-P2* knock-in mice express
the P2 allele in most mOSNs, owing to tTA-dependent induction of this
allele in OSN progenitors[22]. Transcriptional priming results in biased GI
hub assembly over the P2 locus and stable tTA-independent expression
in approximately 75% of mOSNs[22]. Haplotype-imputed single-cell Hi-C
contact maps were generated from Dip-C libraries and used to render
three-dimensional models of 161 individual OSN nuclei (Fig. 2a and
Extended Data Fig. 3), on the basis of a DNA polymer model[21]. A repre-
sentative model of a *Mor28iGFP* nucleus stripped of all genes except
for ORs and GIs illustrates multichromosomal OR compartments and
the GI hub associated with the active *Mor28* allele (Fig. 2a and Extended
Data Fig. 4a,b).

We defined the active hub in our Dip-C model as the collection of GIs
that contain the active OR allele within an approximately 5 particle radii
(p.r.) span. We chose this limit because beyond 5 p.r., spatial GI distri-
bution between active and inactive P2 and *Mor28* alleles, respectively,
became indistinguishable (Extended Data Fig. 4c,d). Comparing Dip-C
data from pure P2⁺ and *Mor28*⁺ OSNs with those from a mixed mOSN
population[22] confirmed the association of *trans* GIs within 2.5 p.r. of
these two OR alleles preferentially in the OSNs that transcribed them
(Fig. 2b and Extended Data Figs. 4c,d and 5a,b). We observed that most
P2⁺–P2⁺ or *Mor28*⁺–*Mor28*⁺ OSN pairs rarely shared common *trans* GIs
in their active hubs, with P2-containing active hubs being as different
from each other as from *Mor28*-containing active hubs (Extended Data
Fig. 5c). Thus, consistent with the multiome data, OR choice is not linked
to the combinatorial identity of *trans* GIs in a hub, and OR alleles may
indiscriminately use *trans* GIs that happen to be nearby.

We also detected further GI hubs in each OSN (Fig. 2a), as previously
described[13]. To compare the genomic organization of active and inactive

hubs, we devised an unbiased strategy for hub definition in each OSN, by hierarchical clustering of GI spatial relationships using the Dip-C model of each nucleus (Fig. 2c and Extended Data Fig. 6a). We define an inactive hub as any cluster of GIs residing within a span of 5 p.r. that does not overlap with the active hub (Extended Data Fig. 6a). Active GI hubs contain $5.42 \pm 3.00$ GIs ($n = 161$ GI clusters), whereas all other inactive GI complexes contain $2.39 \pm 1.84$ GIs on average ($n = 7,990$ GI clusters; Extended Data Fig. 5d). However, although there was a clear distinction between active and inactive hubs in the population, individual OSNs frequently contained inactive hubs with a similar or higher number of converging enhancers compared with the active GI hub. If two GI hubs can have a similar enhancer concentration, why is only one transcriptionally engaged?

We explored genomic differences between active and inactive hubs, using the actual Hi-C contacts from each hub (Extended Data Figs. 5e and 6a). For each nucleus, we identified the active GI hub and the inactive hub that it was most similar to with respect to the number of GIs within the same diameter using the polymer model (Fig. 2c,d and Extended Data Fig. 6a). We then extracted genomic contacts from the active and inactive hub from each nucleus and generated averages among the 161 nuclei (Extended Data Fig. 6a). Having confirmed that we were comparing active and inactive hubs with similar topologies (Extended Data Fig. 6d,e), we calculated the average interaction specificity (contact specificity score; CSS) of GIs in an active or inactive hub. GIs in the active hub made 1.3× more specific contacts with each other (CSS = 0.0117) than did GIs in an inactive hub (CSS = 0.0089) (Fig. 2e,f). Contact specificity between OR alleles, however, was independent of the transcriptional status of the hub and lower than the GI–GI contact specificity (Extended Data Fig. 6b,c). Notably, active and inactive GI hubs exhibited the largest differences when we analysed contacts between GIs and the OR alleles they contained. To fairly compare contact specificity of OR–GI interactions between equivalent active and inactive GI hubs, we challenged ourselves to find the inactive OR gene in the selected inactive GI hub making the most contacts with GIs in every cell, thus biasing our analysis against finding differences in contact specificity. However, even when selecting the inactive OR gene that would be the best competitor to the active OR, we found that contact specificity between the inactive OR and GIs in the inactive hub (CSS = 0.0062) was almost two times lower than that between the active OR and the active hub (CSS = 0.012) (Fig. 2g,h and Extended Data Fig. 6f,g). Notably, in the active hub, contacts between active GIs and the active OR (CSS = 0.012) mirrored the specificity observed between active GI–GI contacts (CSS = 0.0117). Thus, whereas the DNA polymer model identifies inactive hubs that seem identical to active GI hubs, GIs contact the transcriptionally engaged OR allele in a more specific and focused fashion than they do inactive ORs, indicating that there may be distinct chromatin features and biochemical properties between active and inactive GI hubs.

### Active and inactive hubs possess distinct biochemical properties and histone modification features

To characterize differences between transcriptionally engaged and inactive hubs in each OSN, we interrogated the biochemical and histone modification properties of the two types of GI hub, using liquid Hi-C[26]. In liquid Hi-C, genomic interactions in euchromatin show a greater loss in contact specificity than those in heterochromatin, a result confirmed by time-course liquid Hi-C in P2[+] sorted cells[27] (Extended Data Fig. 7a,b). During this time course, we examined CSS changes in the active and inactive hubs (Fig. 3a). We used P2–*trans* GI and inactive OR–*trans* GI Hi-C contacts as proxies for the active and inactive hubs, respectively. Indeed, the two types of interaction are distinct: P2–*trans* GI contacts are highly focused, generating a dot at the centre of the heatmap, whereas inactive OR–*trans* GI contacts produce a stripe owing to homogeneous GI interactions with all the ORs of a cluster.

The reduced specificity in GI-inactive OR contacts in bulk was consistent with the comparison of GI–OR contact specificity in active and inactive hubs at the single-cell level. Notably, within 5 min of predigestion, the active hub experienced a decrease in contact specificity double that observed in inactive hubs, a trend that persisted at 30 min and became significant at 60 min (63.2% versus 38.1%, $P < 0.05$) (Fig. 3b and Extended Data Fig. 7b). The complete lack of stereotypy in the constitution of active and inactive GI hubs precluded normalization of our liquid Hi-C experiments to digestion efficiency[26]. However, even if inactive hubs were more stable owing to reduced predigestion, this would confirm the distinct biochemical properties of active and inactive GI hubs.

We also performed protein-directed mapping of genome architecture through H3K27ac Hi-C chromatin immunoprecipitation (HiChIP)[28]. P2[+] sorted cells were processed for HiChIP[27,29] (Extended Data Fig. 7c,d). Differences in contact specificity between Hi-C and H3K27ac HiChIP over GI hubs were measured to infer H3K27ac status. Whereas contacts in the active hub were enriched for H3K27ac (131% increase in contact specificity), other GI–OR contacts were reduced (18% reduction in contact specificity) (Fig. 3c,d). Furthermore, total contacts made to GIs showed that whereas the active OR gene significantly increased its contacts to GIs by 2.7-fold, both inactive OR genes and other GIs significantly decreased their GI contacts (Extended Data Fig. 7e–h). Thus, the results of Dip-C, liquid Hi-C and H3K27ac HiChIP indicate that the active GI hub has distinct biochemical properties distinguishing it from the other GI hubs.

### OR transcription facilitates 'symmetry breaking' and singular OR gene choice

Hi-C experiments from fluorescence-activated cell (FAC)-sorted MOE populations show that GIs initiate *trans* contacts with each other and with OR genes at the onset of polygenic OR transcription (Extended Data Fig. 8a–d). DNA fluorescence in situ hybridization (FISH) and Dip-C[13] show that these early contacts reflect the assembly of multiple GI hubs (Extended Data Fig. 8e–g). Further, scRNA-seq and Dip-C experiments (LimCA) described in a recent preprint[30] indicate that GI hubs formed during differentiation are actively engaged in polygenic OR transcription. Thus, multiple GI hubs drive OR co-expression in OSN progenitors, but only one of these hubs remains transcriptionally active in mOSNs (Fig. 4a). We propose a 'counting' mechanism that stochastically eliminates all but one hub during differentiation. Such a counting process has been described for X-chromosome inactivation[31], and it has been speculated that it may occur through a 'symmetry-breaking' process[32,33] that protects one and silences the other X chromosome(s). Recent observations in *gg8-tTA>tetO-P2* mice, whereby P2 induction during polygenic OR transcription results in preferential choice of this OR in mOSNs[22] (Fig. 4b), are consistent with a similar transcription-mediated symmetry-breaking process in OR gene expression.

A symmetry-breaking process predicts that each mOSN can only have a single transcriptionally engaged GI hub; thus, induction of robust P2 transcription in mOSNs should shut off the prevailing GI hub or recruit it over the P2 locus. Indeed, P2 induction in mOSNs using *OmpitTA* silenced the previously chosen ORs and promoted strong GI contacts with P2 in the P2-expressing mOSNs (Fig. 4c). To determine whether P2 hijacks the previously active GI hub or simply silences it, we combined the induction of P2 in mOSNs with a tracing strategy that permanently marks OSNs that have previously chosen a different OR, namely *Mor28*. We crossed *OmpitTA>tetOP-2iGFP* mice to *Mor28icre>tdT fl/+* mice, whereby all cells that have ever transcribed *Mor28icre* are tdT[+], all cells actively expressing *P2iGFP* are GFP[+] and cells that have switched from *Mor28icre* to *P2iGFP* are double GFP[+]tdT[+] (Fig. 4d,f and Supplementary Information Fig. 1).

We sorted GFP[+], tdT[+] and GFP[+]tdT[+] cells and performed ATAC-seq, RNA-seq and Hi-C. GFP[+] and GFP[+]tdT[+] cells possessed a highly accessible

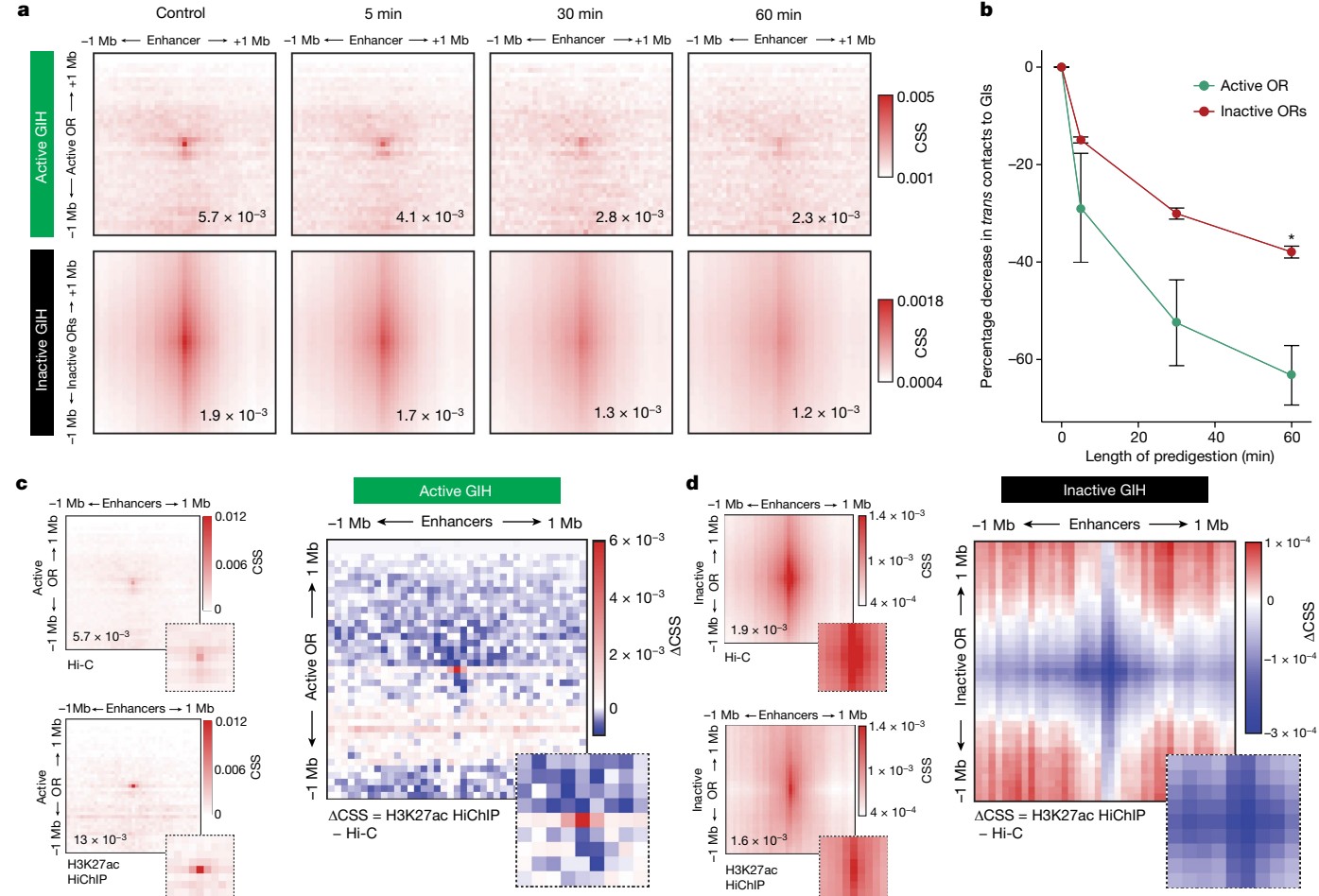

**Fig. 3 | Active and inactive GI hubs possess distinct chromatin properties.**
**a**, Liquid Hi-C was performed on *gg8-tTA>tetOP-2iGFP* GFP⁺ sorted cells for
0 min (control), 5 min, 30 min and 60 min to measure the differential stability
of inactive hubs (inactive ORs to GIs, bottom) and active hubs (active OR to GIs,
top). All heatmaps depict interchromosomal interactions, at 50 kb resolution,
in a 1 Mb radius surrounding a hub. Heatmaps represent merged data from
three biological replicates of liquid Hi-C generated at each time point. **b**, The
mean ± s.e.m. percentage change in CSS (Methods) at 5 min (active OR,
−28.9 ± 19.4%, inactive OR −14.9 ± 1.06%), 30 min (active OR, −52.5 ± 15.3%,

inactive OR −30.1% ± 1.97%) and 60 min (active OR, −63.2 ± 10.6%, inactive
OR −38.0 ± 2.1%) was quantified over the time course of DpnII predigestion
(two-sided Welch's *t*-test *P* = 0.048, *n* = 3 biological replicates per time point).
All points are plotted as mean ± s.e.m. **c,d**, H3K27ac HiChIP was performed on
two biological replicates of *gg8-tTA>tetOP-2iGFP* GFP⁺ sorted cells; then, the
results were merged and differences in CSS compared with Hi-C were assessed
over the active GI hub (**c**; ΔCSS = 7.391 × 10⁻³) and inactive GI hubs (**d**; ΔCSS =
−0.341 × 10⁻³) at 50 kb resolution. Inset heatmaps show contacts zoomed in
to a 200 kb radius surrounding a hub.

P2 locus, robustly transcribed P2 and formed a GI hub around the P2
allele (Fig. 4g,h,i,k and Extended Data Fig. 9a). In tdT⁺ cells, the *Mor28*
locus was also highly accessible, actively transcribed and supported
by a GI hub (Fig. 4g,h,m and Extended Data Fig. 9a). Surprisingly, tdT⁺
cells had elevated P2 accessibility, increased levels of P2 mRNA and a
GI hub over the P2 locus (Fig. 4j and Extended Data Fig. 9a). This was
a different hub from the one surrounding *Mor28*; we did not detect
increased contact specificity of P2–*Mor28* interactions, which would be
expected if they shared a hub (Extended Data Fig. 9b). Therefore, tdT⁺
cells represent newly differentiating mOSNs that initiate tTA-mediated
transcription of P2 but have not yet reached sufficient levels of GFP
expression for fluorescence detection. Thus, we are 'catching' OSNs
at the state of gene switching, a metastable state in which both P2 and
*Mor28* alleles are in contact with distinct functional hubs. This state
may be tolerated only as long as only one OR is highly transcribed; once
P2 expression increases enough to permit GFP detection (GFP⁺tdT⁺
cells), *Mor28* becomes inaccessible and loses contacts with its own
GI hub, and its mRNA levels drop (Fig. 4g,h,n). Thus, whereas low OR
transcription may be sufficient for GI hub engagement and compat-
ible with polygenic OR expression, robust OR transcription breaks
symmetry and terminates transcription of other ORs[34].

## OR protein-independent symmetry breaking indicates potential non-coding OR RNA functions

Although OR protein translation preserved the singularity of OR tran-
scription, we could not explain how it would bias the choice between
competing GI hubs. Thus, we reasoned that symmetry breaking
could be independent of the OR protein-elicited feedback, which is
generated in the endoplasmic reticulum. This would ascribe to OR
transcription per se or to the nascent OR RNAs roles previously sug-
gested for non-coding RNAs in organizing genomic interactions[35] and
nuclear compartments[36,37]. A fundamental difference of our model
is that we attribute nuclear regulatory functions to protein-coding
mRNAs, representing approximately 4% of the mouse genes. We tested
this first by using CRISPR-mediated non-homologous end-joining to
create a non-coding, 'sterile' *tetO-P2* allele (*tetO-P2(nc)*), circumvent-
ing full-length and functional OR protein-elicited feedback. A 25 bp
deletion was induced at the 5′ end of the P2 coding sequence (CDS),
resulting in the production of the full-length P2 transcript but no P2
protein (Extended Data Fig. 10a).

We induced the sterile *tetO-P2(nc)* allele in mOSNs, using *OMPitTA>
tetO-P2(nc)* mice, observing the same induction frequency as that of

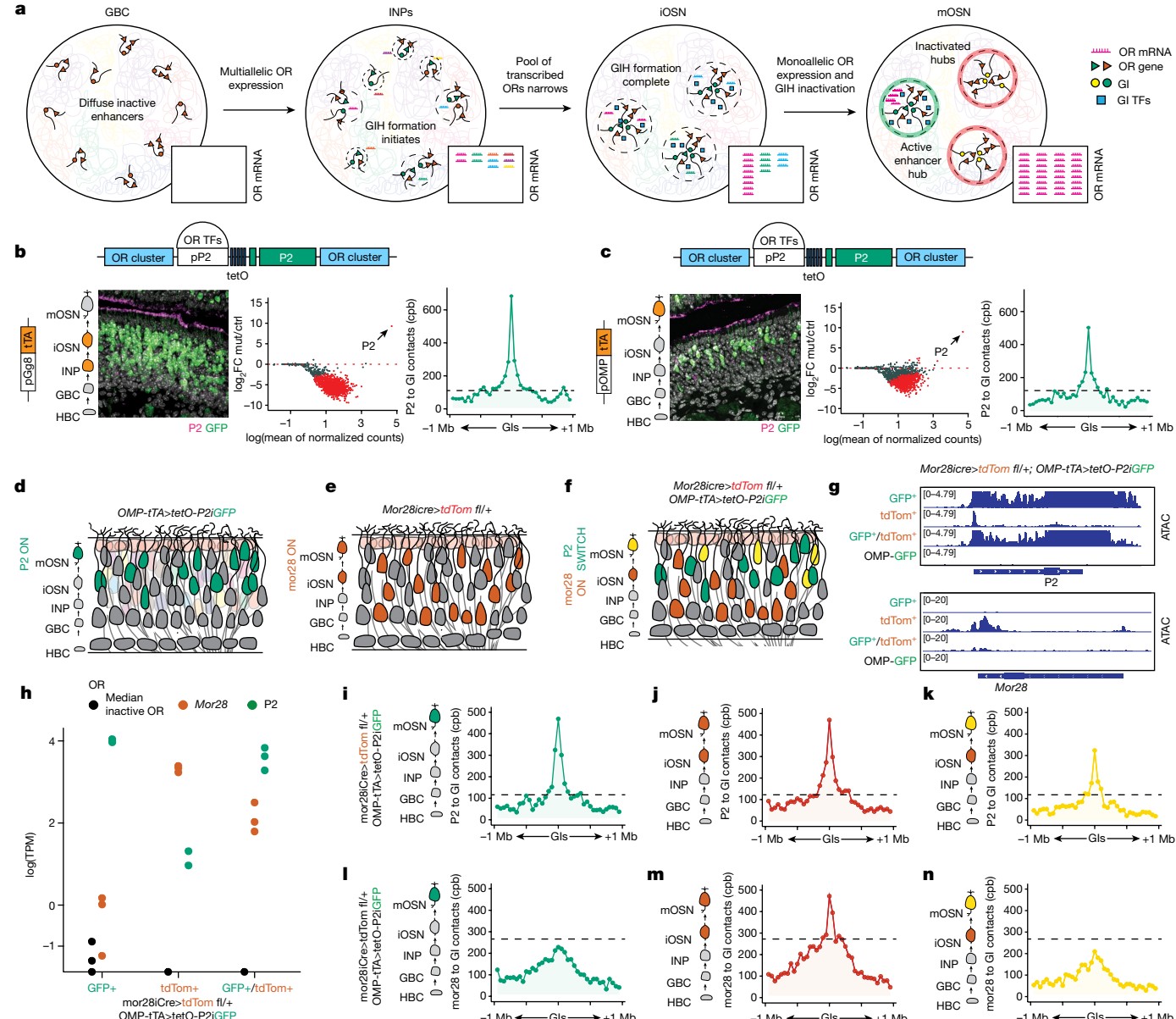

**Fig. 4 | Symmetry breaking as a model for singular OR gene choice.**
**a**, A model of GI 3D reorganization, chromatin remodelling and OR transcription during OSN development. **b,c**, Left, immunofluorescence assays targeting P2 protein (magenta) and GFP (green) in the MOEs of *gg8-tTA>tetOP-2iGFP* (**b**) and *OMP-tTA>tetOP-2iGFP* (**c**) mice. On the left of each immunofluorescence panel, the tTA driver and its expression stage are shown by the orange cells. The schematic at the top of each panel depicts the design of the mutant P2 allele residing in its endogenous locus. Middle: RNA-seq MA plots depicting DeSeq2 normalized OR gene counts versus *OMP-tTA>tetO-GFP*. Significantly changed ORs are shown in red (adjusted *P* < 0.05). Right: Hi-C in GFP+ cells measuring aggregate contacts per billion (cpb) between P2 and a 1 Mb radius surrounding all *trans* GIs. The dashed black line represents P2 to GI contacts in *OMPiGFP* OSNs (negative control). **d**–**f**, Schematics of olfactory epithelia from *OMP-tTA>tetOP-2iGFP* (**d**), *Mor28icre>tdTom fl/+* (**e**) and *OMP-tTA>tetOP-2iGFP; Mor28icre>tdTom fl/+* (**f**) mice. **g**, ATAC-seq over the P2 (top) and the *Mor28* locus (bottom) in GFP+, tdT+ and GFP+tdT+ cells from the quadruple transgenics or GFP+ cells from *OMPiGFP* mice. For each condition, ATAC-seq was performed in three separate biological replicates, and the results were merged. **h**, RNA-seq reads (TPM) of *Mor28*, P2 and the median inactive OR in all three cell types in three biological replicates. **i**–**n**, In situ Hi-C in GFP+ (**i** and **l**), tdT+ (**j** and **m**) and GFP+tdT+ (**k** and **n**) cells measuring aggregate cpb between the P2 (**i**–**k**) or *Mor28* (**l**–**n**) locus and a 1 Mb radius surrounding all *trans* GIs. The dashed black line represents contacts between P2 (top) or *Mor28* (bottom) and GIs in *OMPiGFP* OSNs (negative control). For each condition, in situ Hi-C was performed in three separate biological replicates, and the results were then merged.

the functional *tetO-P2* allele (Fig. 4c, left, and Fig. 5a, left). Moreover, Hi-C on the GFP+ mOSNs showed strong associations with GIs (Fig. 5a, right), and similarly with the functional P2 allele (Fig. 4c, right, and Extended Data Fig. 10d). Notably, RNA-seq on the GFP+ OSNs showed that induction of the sterile P2 allele is sufficient to shut down transcription of the previously chosen ORs, mimicking the intact P2 allele (Fig. 5a, middle, Fig. 4c, middle, and Extended Data Fig. 10c). Furthermore, we generated an inducible OR M71 transgene that produces sterile M71

RNA (*tetOM71(nc)iGFP*) solely under the control of a tetO promoter in mOSNs (Extended Data Fig. 10e,f). RNA-seq on GFP+ OSNs from these transgenic mice showed a significant reduction in OR mRNA levels compared with those of multiple mOSN controls (Extended Data Fig. 10g–i). Thus, synthesis of a sterile OR RNA that contains only the OR CDS also suppresses OR transcription. Notably, transcriptional reduction of endogenous ORs is not as strong as the one observed by the *tetO-P2(nc)* allele. This may be owing to the lower expression levels of this sterile

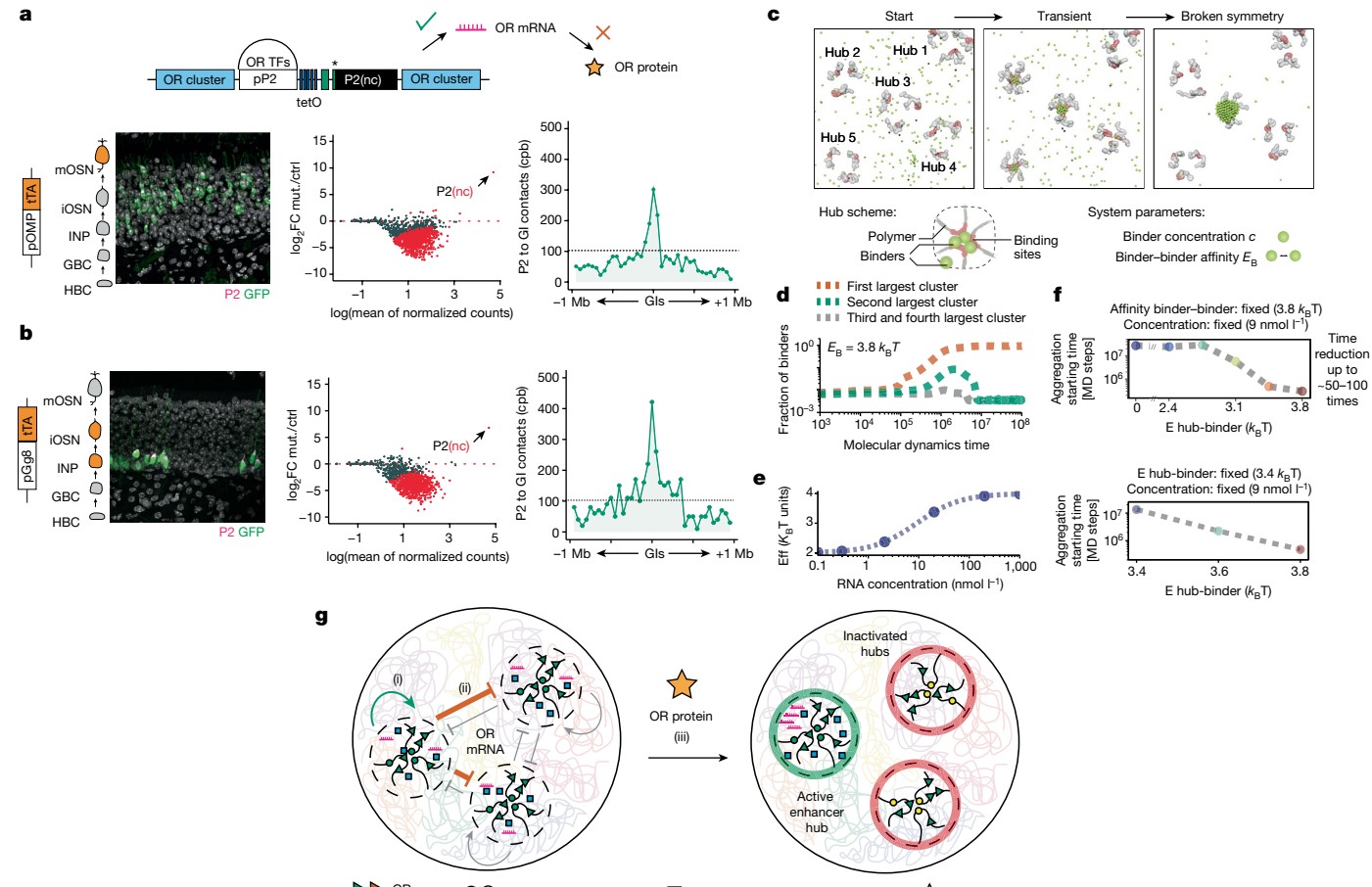

## Fig. 5 | OR RNA synthesis promotes transition to singularity.

**a**, Immunofluorescence targeting P2 protein (magenta) and GFP (green) in the MOE of *OMP-tTA>tetO-P2(nc)iGFP* mice, which expresses P2 mRNA but not protein (left) in mOSNs. MA plot depicting DEseq2 normalized OR gene counts, normalized to *OMPitTA>tetO-GFP*. Significantly changed OR genes are shown in red (adjusted *P* < 0.05). In situ Hi-C (right) in GFP⁺ cells measuring aggregate cpb between P2 and a 1 Mb radius surrounding all interchromosomal GIs. TFs, transcription factors. **b**, Immunofluorescence targeting P2 protein (magenta) and GFP (green) shown in the MOE of *gg8-tTA>tetO-P2(nc)iGFP* mice. MA plot (middle) depicting DeSeq2 normalized OR gene counts, normalized to *gg8itTA>tetO-GFP*. Significantly changed ORs in red (adjusted *P* < 0.05). In situ Hi-C (right) in GFP⁺ sorted cells measuring aggregate cpb between P2 and a 1 Mb radius surrounding all interchromosomal GIs. **c**, Molecular dynamics simulation snapshots from different stages of the symmetry-breaking process. **d**, Symmetry-breaking dynamics captured by monitoring fraction of binders in the largest clusters during time. The fraction of binders in the largest clusters are shown for a symmetry-breaking event ($E_B > E_{BT}(c)$, $c$ = 9 nmol l⁻¹), where only one large cluster self-assembles at equilibrium (red curve). **e,f**, If the binder has affinity for the nascent OR mRNA, increased RNA concentration will increase binder–binder or binder–hub affinity (**e**), resulting in acceleration of the symmetry-breaking process (**f**). **g**, Schematic summary: OR RNA synthesis (i) biases GI hub formation over the transcribed P2, re-enforces the GI hub from which it is produced and (ii) inhibits transcription from competing hubs. Subsequently, P2 protein expression (iii) stabilizes this choice, enabling continuous P2 expression in the absence of tTA. ctrl, control; mut., mutant.

---

transgene, the lack of 5′ and 3′ untranslated region sequences of the OR mRNA, the absence of native OR promoter sequences and the inability to compete for GI hub recruitment. Nevertheless, this result further supports a previously unappreciated role of OR RNA synthesis in the regulation of OR gene choice.

Although our genetic manipulations demonstrate a role of OR RNA synthesis in transcriptional singularity, they also confirm a critical role of the OR protein in this process, as induction of *tetO-P2(nc)* by *gg8-tTA* does not result in stable choice of the sterile P2 allele in mOSNs, unlike the intact *tetO-P2* allele (Fig. 5b, left). GFP⁺ OSNs are restricted to the basal MOE layer and differentiate up to the ATF5⁺ iOSN stage, which is coincident with the induction of OR protein feedback (Extended Data Fig. 10b). However, these iOSNs express only the sterile P2 allele and exhibit GI contacts with the P2 locus, further supporting the non-coding functions of the OR mRNA (Fig. 5b, middle and right, and Extended Data Fig. 10d). It is most likely that without the OR protein feedback, these iOSNs fail to stabilize GI hub–P2 interactions and switch to a different

OR when tTA expression stops. This would explain why the putative non-coding functions of OR RNAs were only revealed when we disentangled OR transcription from the protein-elicited feedback. Thus, singularity is imposed by a two-step process: OR transcription breaks the symmetry between competing GI hubs, and the OR protein-elicited feedback makes this choice permanent, preventing hundreds of non-coding OR pseudogenes from having stable expression in mOSNs.

## Discussion

We propose a physics-based symmetry-breaking model[32] where the self-affinity of GI hub-binding factors in a single prevailing cluster results in phase separation and transcriptional selection of one OR allele among many competing ones (Fig. 5e,h). Symmetry breaking can explain how low polygenic OR expression from multiple hubs could culminate in robust singular OR expression from a single hub, through GI hub-binding factors that aggregate into multiple small

foci before organizing into a single large cluster (Fig. 5c and Extended Data Fig. 10j,k). This process probably starts from *cis* GI–OR interactions at early differentiation stages[30], initiating low-level polygenic OR transcription that signals for *trans* GI recruitment[4] and assembly of competing GI hubs. At this stage, spontaneous fluctuations in local RNA synthesis could be the signal for a phase transition that reinforces transcription in one hub and silences the others (Fig. 5c–f). One explanation for an 'auto' enhancing and 'allo' repressing action of the nascent OR RNA is that it contributes to the efficient recruitment of a limited diffusible transactivator (Fig. 5e). We propose the hypothesis that this transactivator is transcribed at low levels in mOSNs and exhibits selectivity for the sequence and/or structure of the OR RNA, affinity for LHX2, EBF1 or LDB1, and concentration-dependent phase separation properties[38,39]. In this vein, OR RNA will act as a local hub enhancer by recruiting this limited factor and as a global hub repressor by sequestering it from other hubs. With this dual function, small fluctuations in OR RNA synthesis could rapidly break the symmetry between hubs, accelerating the transition to singular OR transcription (Fig. 5e–g). The appeal of a symmetry-breaking model is that it only allows two modes of OR transcription: low and polygenic transcription from multiple hubs (INPs to iOSNs) or singular and robust transcription from one (mOSNs). In other words, it is the high rate of OR transcription in the prevailing interchromosomal GI hub that enforces singular OR expression during differentiation. Given the ever-expanding list of genes forming interchromosomal compartments in neurons[40,41], it will be interesting to investigate the non-coding role of other coding mRNAs in mutually exclusive cell fate decisions.

Although we favour an RNA-mediated symmetry-breaking process, we cannot ignore other explanations of our data. Transcription-enabled chromatin remodelling of the OR locus, which may facilitate transcription factor binding on P2 DNA and GI hub assembly, may also contribute to biased P2 choice upon tTA induction. Similarly, tTA may synergize with endogenous transcription factors on the P2 promoter, facilitating GI hub recruitment to the P2 locus. However, in both scenarios, the competing OR–GI hub interactions dissipate only when P2 RNA levels reach a threshold, supporting a direct role of the OR mRNA in symmetry breaking. We also acknowledge that tTA-induced P2 transcription at the polygenic state (INPs, iOSNs) is stronger than the transcription of competing endogenous ORs, which may artificially bias P2 choice. However, tTA-driven P2 transcription in mOSNs is not as high as the transcription of the already chosen OR, yet it also hijacks the OR choice apparatus. Thus, it is likely that the transcriptional advantage that tTA induction confers on P2 mimics the advantage that different endogenous ORs have along the dorsoventral axis of the MOE, breaking symmetry in a biased, positionally informed fashion[22].

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

# Methods

## Mice

Mice were treated in compliance with the rules and regulations of the Institutional Animal Care and Use Committee of Columbia University under protocol number AABG6553. Mice were euthanized using $CO_2$ followed by cervical dislocation. Both male and female mice were used for experiments. All experiments were performed on dissected olfactory epithelium tissue or on dissociated cells prepared from whole olfactory epithelium tissue. This study used several mouse lines (*Mus musculus*) on mixed C57BL/6J and 129 backgrounds. For Dip-C, H3K27ac HiChIP and liquid Hi-C, cells expressing the OR P2 were obtained by crossing *tetO-P2-IRES-GFP* mice to *Gng8(gg8)-tTA* mice[42] and sorting GFP+ cells from dissociated MOE. For Dip-C, *Gng8tTA>tetO-P2* and *Mor28-IRES-GFP*[25] mice were crossed to CAST/EiJ mice (Jax strain 000928) to generate $F_1$ hybrids where known single-nucleotide polymorphisms could be used for haplotype imputation. For the Hi-C data shown in Supplementary Fig. 8a–f, horizontal basal cell and INP analyses were performed on previously published Hi-C data[4], iOSNs were isolated by performing Hi-C on heterozygous *Atf5-IRES-RFP*[43] *OMP-IRES-GFP* mice, sorting RFP+GFP− cells, and GFP+ cells from *OMP-IRES-GFP* mice[25] were used to isolate mOSNs. For ATAC-seq, RNA-seq and in situ Hi-C results shown in Fig. 4g–n, *Mor28-IRES-cre*[25], *Rosa26(LSL-tdTomato/+)*[44], *OMP-ires-tTA* and *tetO-P2* alleles were crossed to create mice heterozygous for all alleles. For immunofluorescence, Hi-C and RNA-seq, *tetO-P2(nc)* mice were generated by performing CRISPR/non-homologous end-joining on heterozygous *tetO-P2* embryos with the following guide targeting the 5′ region of the P2 CDS (5′-GGGAAACTGGACAACTGTCA-3′). Verification of frameshift was done by performing TIDE analysis on PCR amplicons of the unmutated and mutated *tetO-P2* sequence from gDNA of $F_1$ pups of founder mice and stock *tetO-P2* mouse lines. For immunofluorescence and RNA-seq, *tetOM71(nc)* mice were generated by first assembling a tetOM71(nc)-IRES-GFP construct made by performing an NEB HiFi assembly using an *M71(nc)-IRES-GFP* gene block made with Integrated DNA Technologies (IDT, https://www.idtdna.com/pages) and a pTRE Tight tetO-Fv2E-Perk plasmid (gift from H. Shayya). The M71 CDS was rendered non-coding by changing the 11th amino acid to a stop codon and mutating all in-frame methionine codons to another missense codon that would result in few modifications to RNA secondary structure, thereby preventing any in-frame translation. NheI restriction digest released a fragment containing the *tetOM71(nc)* construct, which was used for pronuclear injection in B6CBAF1 zygotes. Tail biopsy and PCR were used to identify founder mice containing the transgene; these were crossed to *Omp-irestTA*[45] animals to screen for both germline transmission and tTA-dependent transgene expression in mOSNs. *tetOM71-LacZ* mice[46] and *tetO-GFP* mice were also crossed to OMP-tTA and/or Gng8tTA drivers for immunofluorescence and RNA-seq experiments. For all experiments, mice were between 5 and 12 weeks of age.

**Fluorescence-activated cell sorting.** Cells were prepared for FAC sorting as previously described[4] by dissociating olfactory epithelium tissue with papain for 40 min at 37 °C according to the Worthington Papain Dissociation System. Cells were washed twice with cold PBS before being passed through a 40 μm strainer. Live (DAPI-negative) fluorescent cells were collected for RNA-seq and liquid Hi-C. Alternatively, for Hi-C and HiChIP, cells were fixed for 10 min in 1% formaldehyde in PBS at room temperature, quenched with glycine and washed with cold PBS before sorting of fluorescent cells. For Dip-C, cells were fixed in 2% formaldehyde in PBS at room temperature for 10 min, inactivated with 1% bovine serum albumin (BSA) and washed with cold 1% BSA in PBS before sorting of fluorescent cells. All cells were sorted on a Beckman Coulter Low Flow Astrios EQ.

## Olfactory epithelium immunofluorescence

Immunofluorescence assays were performed as previously described[43]. In brief, dissected MOEs were fixed in 4% (w/v) paraformaldehyde in PBS for 1 h at 4 °C and then washed three times for 10 min each time in PBS. Olfactory epithelia were decalcified overnight at 4 °C in 0.5 M EDTA (pH 8) and washed again in PBS. MOEs were cryoprotected overnight at 4 °C in 30% (w/v) sucrose in PBS, embedded in OCT, frozen over an ethanol/dry ice slurry and stored at −80 °C until sectioning. To ensure full coverage of the MOE, tissue was serially sectioned in the coronal plane, moving from the flat posterior surface to the anterior surface. Six slides were prepared with four sections per slide, of 15 mm sections collected on slides starting at the moment when turbinate 3 separated from the dorsalmost aspect of the epithelium[47]. Slides were frozen at −80 °C until the day of staining experiments, when they were thawed, washed for 5 min in PBS and postfixed for 10 min at room temperature in 4% (v/v) formaldehyde (Thermo Fisher) in PBS. Tissue was then washed three times (5 min each time, in PBS + 0.1% Triton X-100 (Sigma)) and blocked for 1 h at room temperature in 4% (v/v) donkey serum (Sigma) + 1% Triton X-100 in PBS. Primary antibodies against GFP (chicken anti-GFP ab13970, 1:2,000), P2 (Olfr17 antibody were raised in guinea pig, 1:2,000), M71 (1:3,000)[11] and/or LacZ (abcam ab4761, 1:16,000) were diluted in block solution and used for incubation overnight at 4 °C. The following day, sections were washed, incubated with secondary antibodies (Jackson Immunoresearch, 1:500 in block solution) for 1 h at room temperature, washed again and mounted using VECTASHIELD Vibrance (Vector Labs) mounting medium. Images were rendered with ImageJ 2.0.0.

## In situ Hi-C, liquid Hi-C and H3K27ac HiChIP

**In situ Hi-C and liquid Hi-C.** In situ Hi-C was performed exactly as previously described[4]. The liquid Hi-C protocol[26] was integrated into our Hi-C protocol to perform liquid Hi-C in OSNs. In brief, MOE was dissociated from *gg8-tTA>tetO-P2* mice, and 400,000 GFP+ cells were sorted as described above per condition per replicate, with three biological replicates per time point. After sorting, cells were pelleted at 600*g*, for 10 min at 4 °C, and resuspended in 300 μl chilled lysis buffer (50 mM Tris pH 7.5, 0.1% Igepal, 150 mM NaCl, protease inhibitor in water). Samples were then pelleted for 7 min at 700*g* and 4 °C and then resuspended in 105 μl DpnII-MasterMix (DpnII Buffer, 250 U DpnII) and placed on a preheated thermomixer at 37 °C with shaking at 900 rpm for 5 min, 30 min or 60 min. Samples were immediately placed on ice for 10 min after predigestion. For 0 min liquid Hi-C, after lysis, cells were immediately processed for fixation. For fixation, samples were diluted into 1% formaldehyde in PBS, rotated on a rotisserie for 10 min at room temperature and quenched with 1/10 volume of 1.25 M glycine. Samples were pelleted at 2,500*g*, for 5 min at 4 °C, washed with PBS and then resuspended in nuclear permeabilization solution (as described in the in situ Hi-C protocol). All subsequent steps and the library preparation were performed as previously described[4]. Samples were sequenced paired-end 50 bp or 100 bp on Illumina NextSeq 550, Illumina NovaSeq2000 or Illumina NextSeq2000. Three biological replicates were created for all liquid Hi-C experiments; once libraries had been confirmed to be similar, they were merged. Heatmaps were generated from merged cooler files, and Welch's two-sample *t*-tests on CSS scores were performed on unmerged replicates.

**H3K27ac HiChIP.** The HiChIP protocol was given by the Chang laboratory and integrated into our Hi-C protocol for H3K27ac HiChIP on OSNs[28]. MOE from 5–7 *gg8-tTA>tetO-P2* mice were dissociated to obtain 4 million GFP+ cells per replicate, for a total of two replicates. Cells were processed according to the in situ Hi-C protocol with the following exceptions: nuclei were digested for only 2 h instead of overnight, and complete nuclei digestion was verified by running reverse cross-linked digested nuclei on a DNA agarose gel. After ligation, nuclei were pelleted

at 2,500g, for 5 min at 4 °C, and stored overnight at −20 °C. The next day, nuclei were resuspended in 130 µl of HiChIP nuclear lysis buffer (50 mM Tris pH 7.5, 10 mM EDTA, 1% sodium dodecyl sulfate, protease inhibitor in water) and sheared on a Covaris S220 with the following parameters: duty cycle, 2%; PIP, 140; cycles/burst, 200; time, 4 min. After shearing, samples were precleared, immunoprecipitation was performed with 1 µg H3K27ac antibody per 4 million cell input (Abcam GR323193701) and libraries were prepared exactly as previously described[28]. Samples were sequenced paired-end 50 bp on an Illumina NextSeq2000.

**In situ Hi-C, liquid Hi-C and HiChIP alignment and data preprocessing.** Alignment and data preprocessing were performed exactly as previously described[22]. In brief, reads were aligned to the mm10 genome using the distiller pipeline (https://github.com/mirnylab/distiller-nf, requirements: java8, nextflow and Docker); uniquely mapped reads (mapq > 30) were retained, and duplicate reads were discarded. Contacts were then binned into matrices using cooler[48]. Data pooled from two to three biological replicates were analysed, after the results of analyses of individual replicates had been confirmed to be similar.

## RNA-seq
**RNA extraction and library preparation.** All RNA-seq experiments were performed under RNA clean conditions. For RNA-seq, live cells were sorted into RNase-free PBS, pelleted at 600g, for 5 min at 4 °C, then resuspended in 500 µl TRIzol, flash-frozen in liquid nitrogen and stored overnight at −80 °C. RNA extraction was performed the next day. TRIzol suspensions were thawed on ice, 1/5 V of 1-bromo-3-chloropropane was added, and tubes were shaken vigorously to combine phases. Phases were allowed to separate for 2 min at room temperature, then tubes were centrifuged at 10,500 rpm, for 15 min at 4 °C, in an Eppendorf centrifuge C5424R. We collected the upper aqueous phase and transferred to a new tube. Then, 1/2 V of isopropanol and 1 µl of linear polyacrylamide (Sigma Aldrich 56575) were added, the tube was inverted to mix the contents, and RNA was allowed to precipitate for 10 min at room temperature. Tubes were centrifuged for 10 min at 10,500 rpm and 4 °C. The supernatant was removed, and 1 V of 75% ethanol was added to the pellet, which was dislodged by flicking the tube. Tubes were centrifuged for another 5 min, at 10,500 rpm and 4 °C. Ethanol was removed, and tubes were allowed to air dry for 5 min until the pellet turned clear. Next, we added 26 µl of RNase-free water, 3 µl of Ambion DNase I 10× buffer and 1 µl of DNase I (AM2222) to remove all DNA and incubated tubes at 37 °C for 30 min. RNA was purified by a 1.5× AMPure bead clean-up, measured on a nanodrop and used as the input for library preparation with a SMARTER Stranded Total RNA-Seq Kit - Pico Input Mammalian v2 (TaKaRa Bio USA). *OMP-tTA>tetO-GFP*, *gg8-tTA>tetO-GFP* and two *gg8-tTA>tetO-P2* libraries were prepared with the TruSeq kit. However, mOSN samples were compared with both *OMP-tTA>tetO-GFP* (TruSeq prep) and *OMP-IRES-GFP* (TaKaRa Bio USA), which label the same neurons, and produced the same results (Extended Data Fig. 10c,g–i). Libraries were sequenced on either a NextSeq550 or a NextSeq2000 and were sequenced to a targeted coverage of approximately 25 million reads. All RNA-seq experiments were performed with two to three biological replicates.

**RNA-seq data processing and analysis.** Data processing and analysis was performed as previously described[12]. In brief, adaptor sequences were removed from raw sequencing data with CutAdapt. RNA-seq reads were aligned to the mouse genome (mm10) using STAR[49]. SAMtools was used to select uniquely aligning reads by removing reads with alignment quality alignments below 30 (-q 30). RNA-seq data were analysed in R with the DESeq2 package[50]. For MA plots, DESeq2 normalized gene counts were compared between control and knockout mice, and significantly changed genes were identified with an adjusted *P* value cutoff of 0.05. DESeq2 normalized counts were used to examine expression

levels of genes (Extended Data Fig. 2a–c). Principal component analysis on all genes except *Olfr* genes was performed on RNA-seq datasets, to separate cells according to their developmental cell stage (Extended Data Fig. 10b).

## ATAC-seq
**ATAC-seq library preparation.** ATAC-seq libraries, data processing and bigwig generation were performed exactly as previously described[12]. In brief, cells were pelleted (500g, 5 min, 4 °C) and then resuspended in lysis buffer (10 mM Tris-HCl, pH 7.4, 10 mM NaCl, 3 mM MgCl2, 0.1% IGEPAL CA-630). Nuclei were immediately pelleted (1,000g, 10 min, 4 °C). Pelleted nuclei were resuspended in transposition reaction mix prepared from Illumina Nextera reagents (for 50 µl: 22.5 µl water, 25 µl 2× TD buffer, 2.5 µl Tn5 transposase). The volume of the Tn5 transposition reaction was scaled to the number of cells collected: 1 µl mix per 1,000 cells. If fewer than 10,000 cells were collected by FACS, 10-µl-scale reactions were performed. Transposed DNA was column purified using a Qiagen MinElute PCR cleanup kit (Qiagen). The transposed DNA was then amplified using barcoded primers and NEBNext High Fidelity 2× PCR Master Mix (NEB). Amplified libraries were purified using Ampure XP beads (Beckman Coulter) at a ratio of 1.6 µl of beads per 1 µl of library and eluted in 30 µl of elution buffer (10 mM Tris-HCl pH 8, 0.1 mM EDTA). Libraries were sequenced on either a NextSeq550 or a NextSeq2000 and were sequenced to a targeted coverage of approximately 25 million reads.

**ATAC-seq data processing.** Adaptor sequences were removed from raw sequencing data with CutAdapt, and reads were aligned to the mouse genome (mm10) using Bowtie2. Default settings were used, except that a maximum insert size of 1,000 (-X 1,000) was allowed for ATAC-seq. PCR duplicate reads were identified with Picard and removed with SAMtools. SAMtools was used to select uniquely aligning reads by removing reads with alignment quality alignments below 30 (-q 30). For ATAC-seq, regions of open chromatin were identified by running HOMER peak calling in 'region' mode, with a fragment size of 150 bp and a peak size of 300 bp. For ATAC-seq signal tracks, the results of replicate experiments were merged, and HOMER was used to generate 1 bp resolution signal tracks normalized to a library size of 10,000,000 reads. Reads were shifted 4 bp upstream to more accurately map the Tn5 insertion site. Reads were extended to the full fragment length, as determined by paired-end sequencing. Bigwigs were visualized with the Integrated Genome Browser 9.0.0.

## Dip-C generation
**Dip-C and data preprocessing.** Cas mice were crossed to *gg8-tTA>tetO-P2-IRES-GFP* or *Mor28-IRES-GFP* heterozygous $F_1$ hybrids. Dip-C and data preprocessing were performed exactly as previously described[22] and following the quality control metrics as previously described[13], with the following exceptions. Each Dip-C library was sequenced on a single lane of an Illumina NovaSeq 6000. Reads were trimmed with CutAdapt v.1.17, and Dip-C libraries were aligned with BWA 0.7.17. Haplotype-imputed single-cell contacts were generated using the dip-c package (https://github.com/tanlongzhi/dip-c; requirements: hickit r291 and k8-Linux K8: 0.2.5-r80. We excluded cells that had fewer than around 400,000 contacts, a low contact-to-read ratio, or high variability in three-dimensional structure across computational replicates. Overall, the median number of contacts across nuclei was 715,690 contacts per cell for 74 cells for *Mor28-IRES-GFP* Dip-C and 694,462 contacts per cell for 84 cells for *gg8-tTA>tetO-P2-IRES-GFP* Dip-C, for a total of 161 cells. Three-dimensional reconstruction of Dip-C models was performed in PyMOL 2.5.3 as previously described[21].

## DNA FISH
Oligopaint probes specific for 20 kb encompassing the 30 most interacting GIs (based on bulk Hi-C results) and for the P2 locus were

generated using oligominer scripts (https://github.com/brianbeliveau/OligoMiner). Sections of the MOE were fixed, denatured and hybridized as previously described[51,52]. Imaging was performed using the Vutara VXL at the Zuckerman Institute Imaging Platform.

## Multiome generation

**Purification of nuclei.** Nuclei must be purified under RNA clean conditions. A cell suspension of mouse MOE was obtained from an adult mouse following the dissociation conditions previously described[12]. Cell pellets were immediately resuspended in 300 μl of cold RNAse-free lysis buffer (10 mM Tris-HCl, pH 7.4, 10 mM NaCl, 3 mM MgCl$_2$, 0.1% IGEPAL CA-630), and nuclei were pelleted in an Eppendorf 5810R centrifuge at 1,000$g$ for 10 min at 4 °C. Nuclei were resuspended in 500 μl 10× homogenization buffer (100 mM Trizma base, 800 mM KCl, 100 mM EDTA, 10 mM spermidine trihydrochloride, 10 mM spermidine tetrahydrochloride in double-distilled H$_2$O), and the pH was adjusted to 9–9.4 with NaOH. Instructions for preparation of homogenization buffer can be found in Zhang et al.[53]. RNAse inhibitor (NEB MO314L) was added, followed by 500 μl 82% OptiPrep solution (4.1 ml OptiPrep solution (Sigma Aldrich D1556-250ML), 25 μl 1 M CaCl$_2$, 15 μl 1 M magnesium acetate, 50 μl 1 M Tris pH 8, 810 μl water), and the mixture was placed on ice. Then, 1 ml homogenate was carefully added on to 1 ml of 48% OptiPrep solution (2.4 ml OptiPrep solution, 800 μl 1 M sucrose, 25 μl 1 M CaCl$_2$, 15 μl 1 M magnesium acetate, 50 μl 1 M Tris pH 8, 1,710 μl water) and spun down in a precooled swinging bucket centrifuge (Eppendorf 5810R) at 32,00$g$ for 20 min at 4 °C, with acceleration 5/9 and deceleration 0/9 (no break)[54]. The supernatant was aspirated and disposed of without dislodging the pellet. The pellet was air-dried and resuspended in 500 μl PBS diluted with 0.04% BSA with RNAse inhibitor. Cell concentration was measured for accurate loading into the 10× pipeline. Two independent multiomes were generated from a 12 week old (Fig. 1, wild-type background) and a 5-week-old mouse (Extended Data Fig. 1; *gg8-tTA>tetO-P2(nc)* background) and analysed separately. Both multiomes produced the same findings.

**10x Genomics scATAC and scRNA library generation.** Joint scRNA-seq and scATAC-seq libraries were prepared in collaboration with the Columbia Genome Center using the 10x Genomics Single Cell Multiome ATAC + Gene Expression kit according to the manufacturer's instructions. Both 10X Single-Cell Expression (GEX) and ATAC libraries were sequenced to around 350 million reads on an Illumina NovaSeq 6000 150PE.

**Generation of aligned multiome data.** Raw sequencing data were demultiplexed with cellranger-arc mkfastq and aligned with cellranger-arc count. An mm10 fasta file and a custom GTF with extended OR annotations[55] were used to generate a reference package for alignment with cellranger-arc mkref. Our multiome contained an estimated 8,856 cells (12,936 cells for independent replicates; Extended Data Fig. 1) from the MOE, with a median of 2,671 high-quality ATAC fragments per cell (median 9,078 high-quality ATAC fragments per cell for independent replicates; Extended Data Fig. 1) and a median of 1,316 GEX genes per cell (1,006 GEX genes per cell for independent replicates; Extended Data Fig. 1). All multiome data were analysed in R v.4.1.3 using packages Signac v.1.6.0 and Seurat v.4.1.0.

## Molecular dynamics simulations of GI hubs in OSNs

To investigate the symmetry-breaking mechanism of GI hubs occurring in OSNs, classical molecular dynamics simulations were used[56]. Each hub was made of three distinct polymers, modelled as standard self-avoiding-walk strings composed of $N = 30$ beads. Each polymer was equipped with three binding sites, located in the central region. Polymer ends in a specific hub were anchored to the vertices of a hexagon (Fig. 5c) to ensure hub specificity and spatial separation between the polymers in the hub. Other geometries (for instance, triangular) gave

similar results. Binding sites could attractively interact with binders with an affinity $E_P$ and binder total concentration $c$. In addition, binders could interact among themselves with affinity $E_B$. For the sake of simplicity, polymer bead and binders had the same diameter $\sigma$ and mass $m$, which were both set to 1 (dimensionless units)[56]. All particles interacted with a repulsive Lennard–Jones (LJ) potential to take into account their excluded volume, with diameter $\sigma$ and energy scale $\varepsilon = 1k_BT$, where $T$ is the temperature and $k_B$ is the Boltzmann constant. Between two consecutive beads of a polymer, a finite extensible nonlinear elastic[56] potential was used, with length constant $R_0 = 1.6\sigma$ and elastic constant $K = 30k_BT/\sigma^2$, as previously described[57].

The interactions among binders, as well as the interactions between binders and binding sites, were modelled as a truncated, shifted LJ potential[57]: $V_{LJ}(r) = 4\varepsilon\left[\left(\frac{\sigma}{r}\right)^{12} - \left(\frac{\sigma}{r}\right)^{6} - \left(\frac{\sigma}{R_{int}}\right)^{12} + \left(\frac{\sigma}{R_{int}}\right)^{6}\right]$ for $R_{int} < 1.3\sigma$ and 0 otherwise, where $r$ is the distance between particle centres, and $\varepsilon$, sampled in the range 8–12 $k_BT$, regulates the interaction intensity. The affinities $E_B$ shown in Fig. 5c,d correspond to the minimum of $V_{LJ}$. For the sake of simplicity, the interaction between binder and binding sites was kept constant ($E_P = 3.5k_BT$). To map the length scale $\sigma$ in physical units, we equalized the average interhub distance of nearest neighbouring hubs with the median interhub distance of ~2 μm; this was estimated by measuring the average inter-GI distance in Dip-C nuclei, which was 33.4 p.r., obtaining $\sigma = 60$ nm. Binder concentrations were computed as previously described[57], using $c = N_B/VN_A$, where $N_B$ is the number of binders, $V$ is the volume (in litres) of the simulation box and $N_A$ is the Avogadro number.

The system was in contact with a thermal bath at temperature $T$; therefore, positions evolved according to the Langevin equation[58], with the following standard parameters: friction coefficient $\zeta = 0.5$, temperature $T = 1$ and timestep d$t = 0.012$ (ref. 57). Integration was performed with a velocity Verlet algorithm using the LAMMPS software[59]. The simulation was performed in a cubic box (linear size $D = 64\sigma$) with boundary periodic conditions to avoid finite size effects. For each parameter setting, we performed ten independent simulations. The system was initialized with polymers in random self-avoiding-walk states and binders randomly located in the simulation box and then equilibrated up to $10^8$ time = steps. Configurations were logarithmically sampled up to the equilibrium sampling frequency, that is, every $10^5$ timesteps.

**Phase diagram and symmetry-breaking dynamics.** The phase diagram was obtained by considering several different combinations of system control parameters, that is, binder self-interaction affinity $E_B$ and binder concentration $c$. Symmetry-breaking events were called if, at equilibrium, a large and stable aggregate of binders in a GI hub was detected. To this end, we performed standard hierarchical clustering applied directly to the coordinates of binders, using their Euclidean distance as a metric[60]. Clustering was performed using the linkage function from the Python package scipy.cluster. Then, a distance threshold $R_{thr} = 1.3\sigma$ (as large as the attractive LJ distance cutoff) was set, and a cluster was defined as the set of binders whose cophenetic distance was lower than $R_{thr}$.

To study the dynamics of symmetry-breaking events associated with the formation of a stable cluster in a single GI hub, we considered system configurations from the starting state to the equilibrium state. For each sampled timestep, we applied the clustering procedure described above and then selected the largest clusters, that is, those containing the highest fractions of binders. We then used averaging over independent runs to generate the curves shown in Fig. 5d.

## Statistics

All statistical analyses used Welch's two-sample $t$-test. All averages are reported as mean ± s.e.m. In plots with error bars, points are centred on the mean, and error bars indicate the s.e.m.

## Reporting summary

Further information on research design is available in the Nature Portfolio Reporting Summary linked to this article.

## Data availability

Data that support the findings of this study have been deposited in a GEO superseries with accession number GSE230380. Dip-C data from previously published work from our laboratory were used for Fig. 2b and Extended Data Fig. 5b (GSE158730). Dip-C data from Tan et al. were used to render principal component analyses on single-cell chromatin compartments in Extended Data Fig. 8e,f (GSE121791). Previously published Hi-C data from our laboratory were used for Extended Data Fig. 8a–f and are publicly available at https://data.4dnucleome.org/ under accession numbers 4DNESH4UTRNL (https://data.4dnucleome.org/experiment-set-replicates/4DNESH4UTRNL/?redirected_from=%2F4DNESH4UTRNL), 4DNESNYBDSLY (https://data.4dnucleome.org/experiment-set-replicates/4DNESNYBDSLY/?redirected_from=%2F4DNESNYBDSLY), 4DNES54YB6TQ, 4DNESRE7AK5U (https://data.4dnucleome.org/experiment-set-replicates/4DNES54YB6TQ/?redirected_from=%2F4DNES54YB6TQ), 4DNES425UDGS (https://data.4dnucleome.org/experiment-set-replicates/4DNES425UDGS/?redirected_from=%2F4DNES425UDGS) and 4DNESEPDL6KY (https://data.4dnucleome.org/experiment-set-replicates/4DNESEPDL6KY/?redirected_from=%2F4DNESEPDL6KY). Genome assembly for the mm10 genome that was used for deep-sequencing read alignment can be found at www.ncbi.nlm.nih.gov/datasets/genome/GCF_000001635.20/. Source data are provided with this paper.

## Code availability

All code will be placed in the GitHub repository https://github.com/arielpourmorady/Pourmorady_etal.git and made available upon request. The methodology for all deep-sequencing data analysis can be found in Supplementary Note 1.

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

**Acknowledgements** All mouse experiments were performed in compliance with Institutional Animal Care and Use Committee protocol AC-AABG6553. We thank R. Axel, P. Sims, T. Maniatis, J. Cunningham, A. Rizvi and L. Abbott for critical discussions and suggestions, as well as members of the Lomvardas laboratory. We also thank D. Paterka and the Zuckerman Imaging Platform for help with DNA FISH imaging and analysis, and the Columbia Genome Center for their help in generating 10x Genomics libraries. Finally, we thank H. Chang and K. Kraft for help with the HiChIP protocol. A.D.P. was supported by F30DC020900-01; J.D. by HG003143 and HG011536; and S.L. by NIDCD grant R01DC018744, NIH 4DNucleome consortium grant U01DA052783, and Roy and Diana Vagelos.

**Author contributions** S.L. conceived the study and, together with A.D.P., designed the experiments. A.D.P. performed all Dip-C, liquid Hi-C, H3K27ac HiChIP and ATAC-seq, developed code and analysed the data. In situ Hi-C and RNA-seq were performed by A.D.P., E.V.B., R.D., K.M. and J.K. An unpublished liquid Hi-C protocol and technical assistance were provided by H.B., and J.D., A.D.P. and J.P. conceived genetic strategies for generating non-coding OR reporter mice and performed quality control for the non-coding transgene. E.V.B. generated the *gg8-tTA>tetO-P2iGFP* mice. A.M.C. and M.N. performed all physical modelling analyses. Multiomes were generated by A.D.P. in collaboration with the Columbia Genome Center. All cell sorting was performed by I.S., and A.K. performed the DNA FISH analysis.

**Competing interests** The authors declare no competing interests.

**Additional information**
**Correspondence and requests for materials** should be addressed to Stavros Lomvardas.

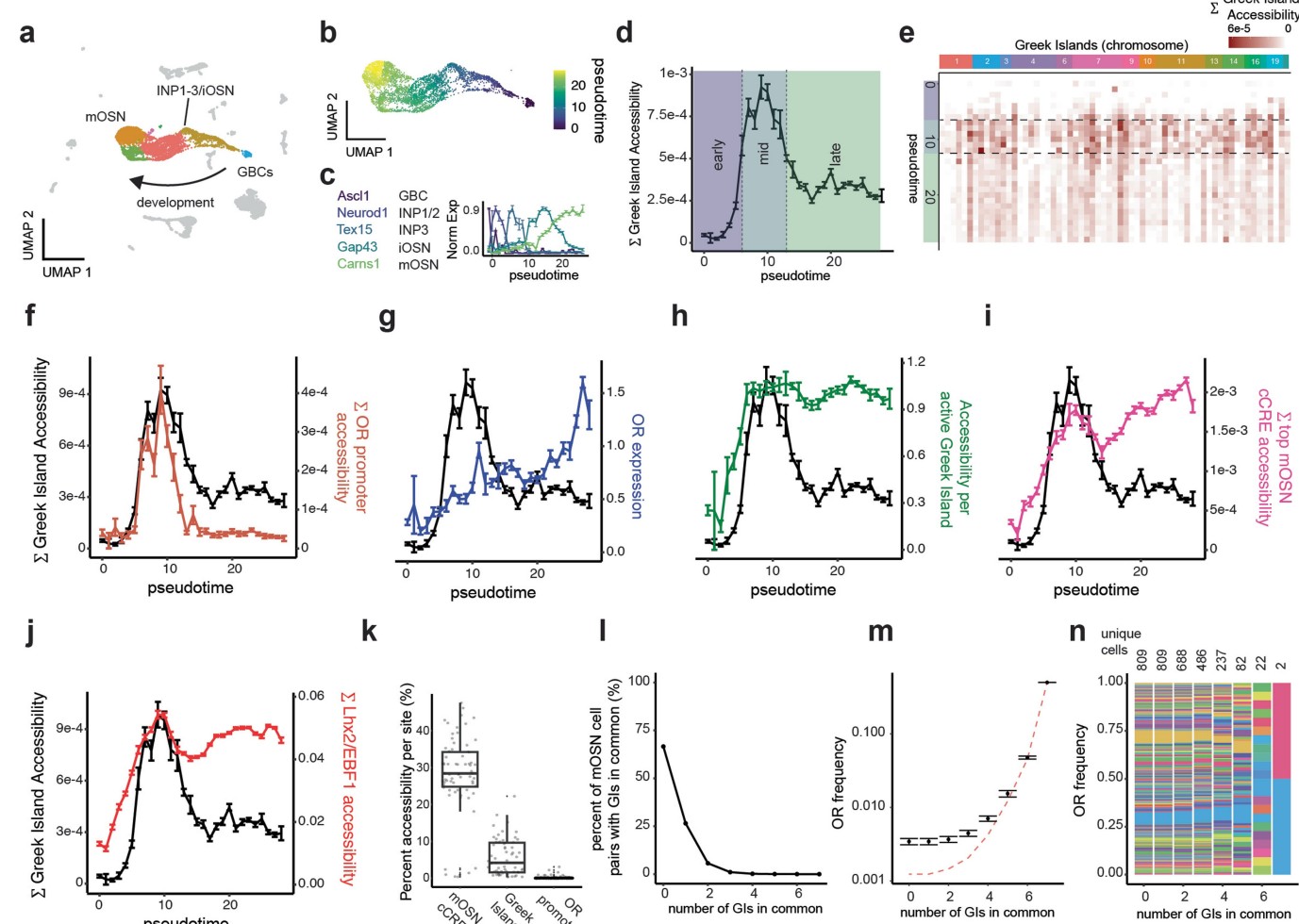

**Extended Data Fig. 1 | Greek Island inactivation during the transition to transcriptional singularity.** (**a**) Two separate multiomes were constructed from independent mice and analyzed separately (see Fig. 1). UMAP representation of the mouse MOE multiome, constructed from weighted nearest neighbors analysis on RNA and ATAC data from 9,034 quality-controlled cells from one mouse. The neuronal lineage contains globose basal cells (GBCs), immediate neuronal precursors (INPs), immature (iOSNs) and mature olfactory sensory neurons (mOSNs). (**b**–**c**) Pseudotime projection onto the neuronal lineage was verified with the expression of known marker genes (scaled mean ± SEM, n = 9,034 cells from one multiome). (**d**) Cumulative GI accessibility dynamics, averaged for all cells rounded to the nearest pseudotime, separate into 3 phases: early, from GBC to INP3; mid, from INP3 to iOSN; and late, mOSN (mean ± SEM per pseudotime, n = 2,371 cells from one multiome). (**e**) GI accessibility over pseudotime per GI. (**f**) Cumulative GI accessibility over pseudotime plotted against cumulative accessibility of OR promoters (cinnamon, n = 1255 sites, mean ± SEM), (**g**) expression level of the most

highly expressed OR per cell (blue, SCT normalized counts, mean ± SEM), (**h**) accessibility per active GI (green, (**i**) top mOSN cCREs (pink, n = 71 sites, mean ± SEM), and (**j**) cumulative accessibility of Lhx2 & EBF1 bound sites in mOSNs (red, n = 4793 sites, mean ± SEM per pseudotime). (**k**) Boxplots comparing fraction of cells with accessibility in mOSN cCREs (27.9% ± 10.9%, n = 71), GIs (5.70% ± 4.94%, n = 63) and OR promoters (3.66e-2% ± 1.36e-1%, n = 1255) in mOSNs. Each boxplot ranges from the upper and lower quartiles with the median as the horizontal line and whiskers extend to 1.5 times the interquartile range. (**l**) 326,836 unique cell pairs, from 809 mOSNs, were tested for overlap in their accessible GIs. (**m**) Average frequency of an OR (mean ± SEM), between all unique cells making up cell pairs sharing between 0 and 7 GIs. The dashed red line represents the expected average OR frequency if the number of unique ORs were to equal the number of unique cells (n = 809 cells from one multiome). (**n**) Frequency of each OR expressed by all unique cells making up cell pairs sharing between 0 and 7 GIs, colored by OR identity.

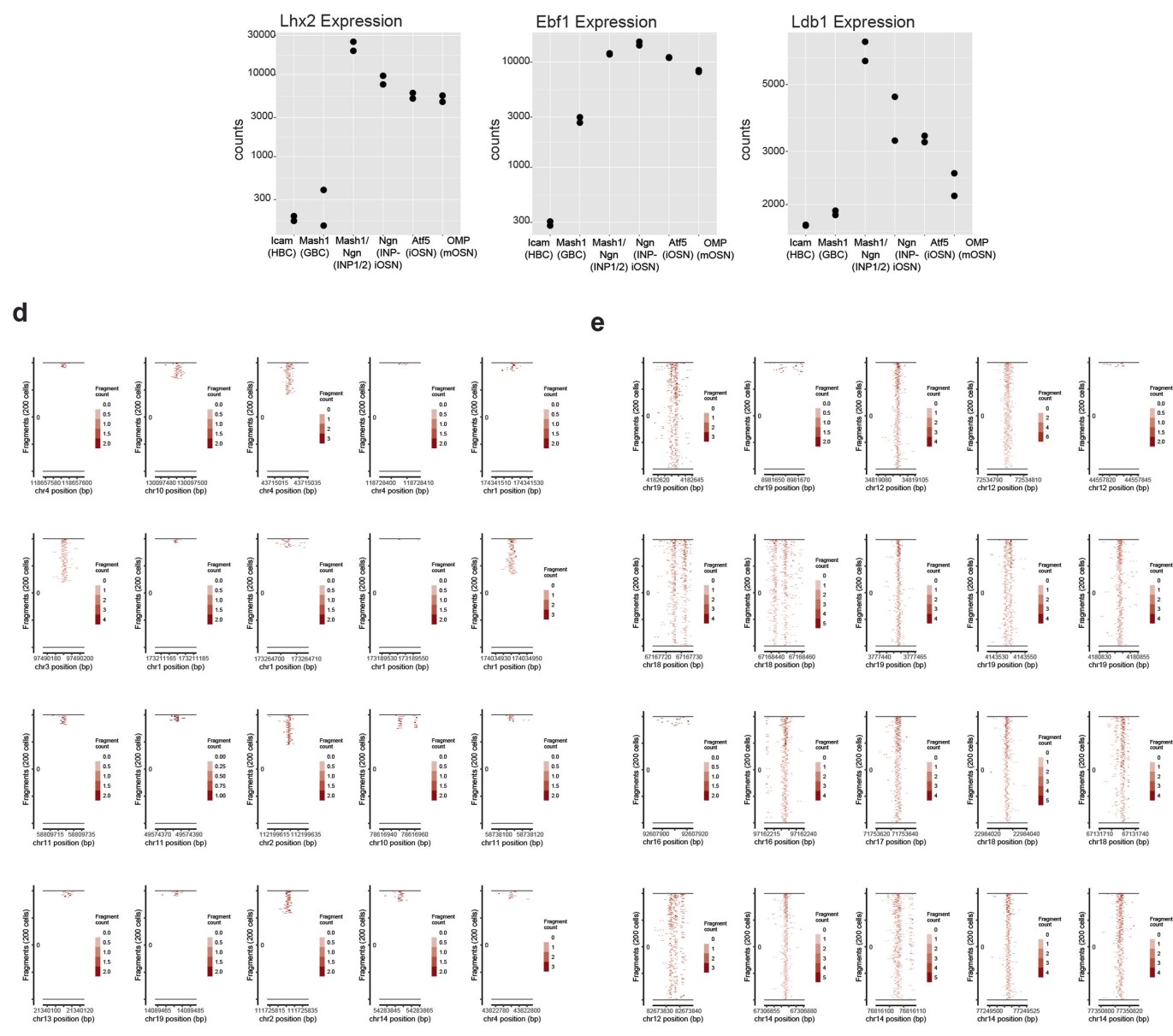

**Extended Data Fig. 2 | Gene expression data and single-cell ATAC analysis of Greek Islands and mOSN cCREs. (a–c)** DEseq2 normalized gene counts of Lhx2 (a), Ebf1 (b), and Ldb1 (c) expression plotted across different cell stages in the olfactory neuronal lineage. **(d–e)** Tileplots were generated depicting Tn5 insertion events in (d) 20 representative GIs and (e) 20 cCREs for the top 200 mOSNs having accessibility at these loci.

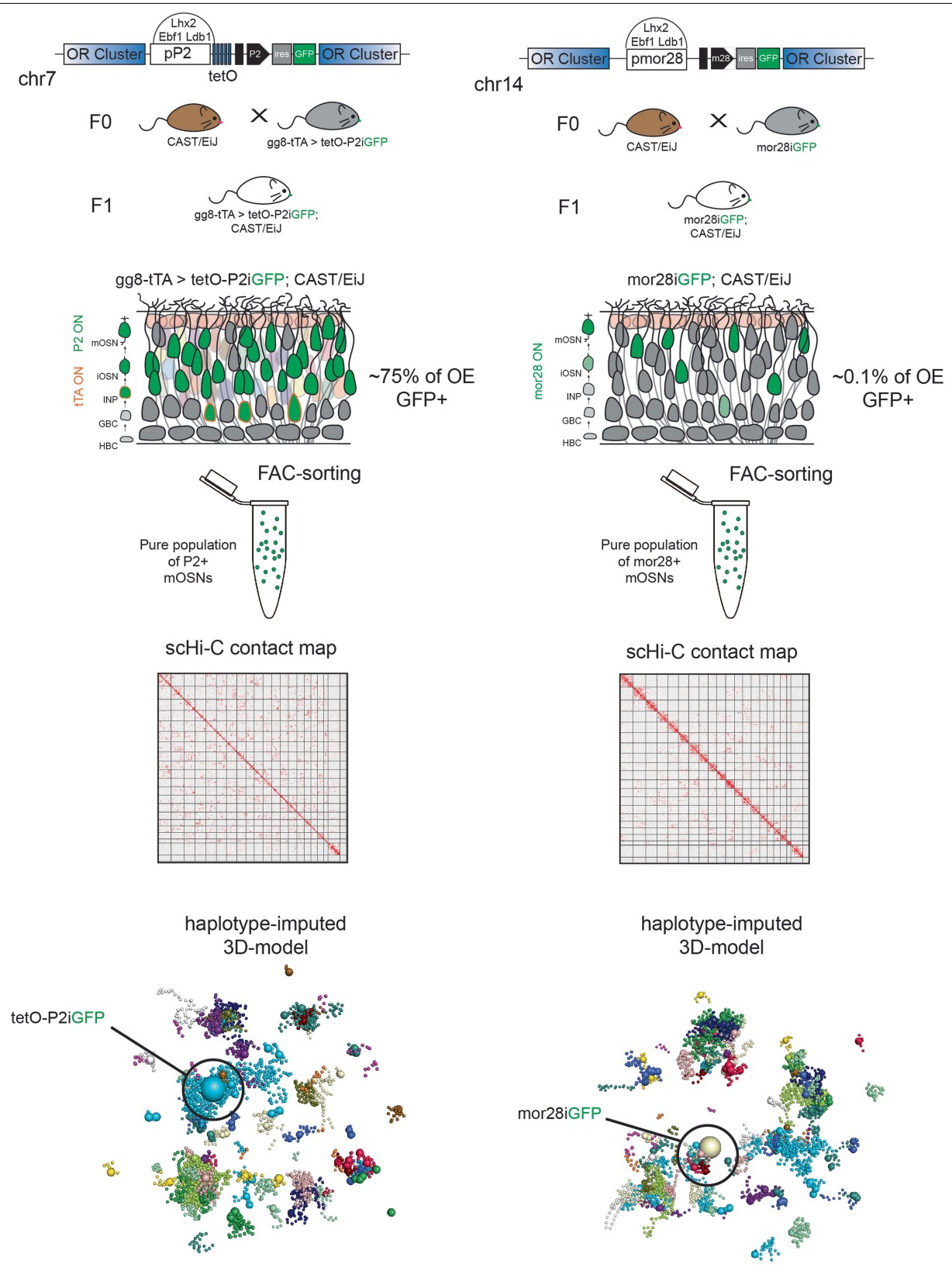

**Extended Data Fig. 3** | See next page for caption.

**Extended Data Fig. 3 | Dip-C experimental strategy.** To perform Dip-C on pure populations of OSNs expressing a known OR from a known allele, two distinct genetic strategies were used. (Left) A mouse containing a driver specific to developing OSNs, *gg8-tTA*, was crossed to a reporter mouse which labels cells actively expressing the OR P2 with GFP, called *tetO-P2iGFP* (*tetOP2*). In *gg8-tTA>tetOP2* mice, P2 expression is briefly induced in developing OSNs through tTA binding on a knocked-in tetO promoter, inserted immediately downstream of the endogenous P2 promoter (pP2). P2 transcription during development biases OR choice, and results in most mOSNs of the MOE expressing the OR P2, which can be identified by co-expression of GFP. *Gg8-tTA>tetOP2* mice are crossed to CAST/EiJ mice which have known SNPs, to generate hybrid mice where maternal and paternal alleles can be identified. Once F1 hybrid mice reach adult age, GFP+ mOSNs are isolated for by FAC-sorting and processed for Dip-C. Through the Dip-C data analysis pipeline, haplotype imputed single-cell contacts are processed to generate single-cell contact maps for each cell, which are used to render 3D-models of each nucleus. The location of the active GI hub in every nucleus is determined by first identifying the location of the active OR allele, which by definition is located within the active GI hub. (Right) In *mor28iGFP* mice, an IRES-GFP sequence is knocked-in immediately downstream of the mor28 coding exon. mOSNs expressing mor28 at this allele, will also produce GFP, whose fluorescence can be used for FAC-sorting. These mice are crossed to CAST/EiJ mice and processed for Dip-C exactly as *gg8-tTA>tetOP2* hybrids.

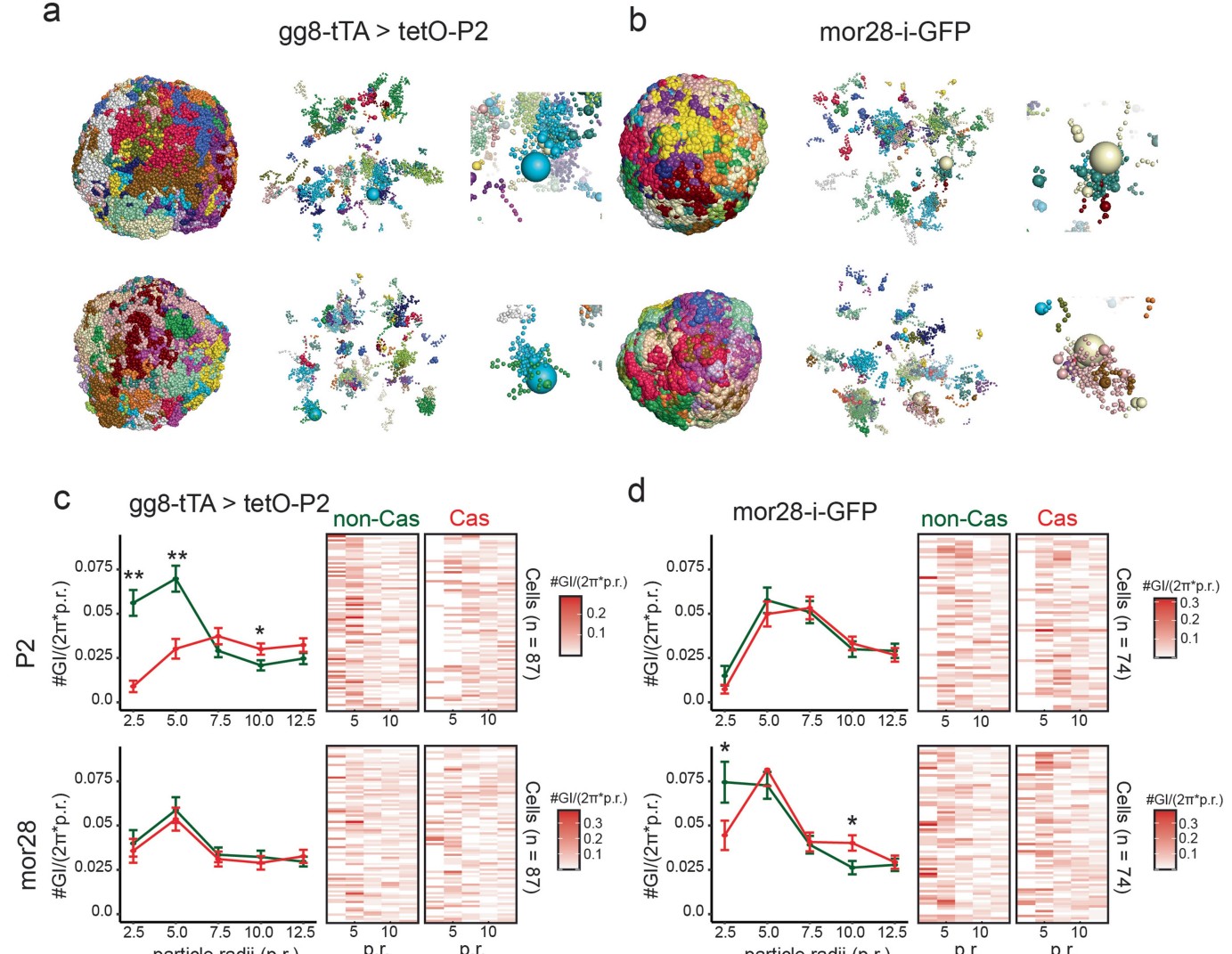

**Extended Data Fig. 4 | Determining the dimensions of the GI hub.**
(**a**,**b**) Representative models of the whole genome (left), OR genes and GIs (middle), and the active GI hubs (right) are shown for 4 *gg8tTA>tetOP2* and *mor28-i-GFP* nuclei colored by chromosome. (**c**,**d**) Left, line plots depict the mean ± SEM number of *trans* Greek Islands, normalized to radial distance, at binned distances away from P2 gene or mor28 gene on non-Cas (green, active in GFP⁺ cells) or Cas (red, always inactive) chromosomes. Right, heatmaps depict the number of *trans* Greek Islands, normalized to radial distance, at

binned distances away from *Olfr17* or *Olfr1507* alleles for each individual cell in a dataset. Dip-C was performed over 2 independent experiments on FAC-sorted GFP⁺ cells pooled from *gg8tTA>tetOP2* mice and *mor28-i-GFP* mice to generate a total of 161 high quality cells (Welch two-sample t-test; *gg8tTA>tetOP2*, P2 locus, 2.5 p.r., p = 3.4e-8; *gg8tTA>tetOP2*, P2 locus, 5 p.r., p = 3.1e-5; *gg8tTA>tetOP2*, P2 locus, 10 p.r., p = 0.03; *mor28-i-GFP*, mor28 locus, 2.5 p.r., p = 0.036; *mor28-i-GFP*, mor28 locus, 10 p.r., p = 0.017; n = 87 P2 cells and n = 74 mor28 cells).

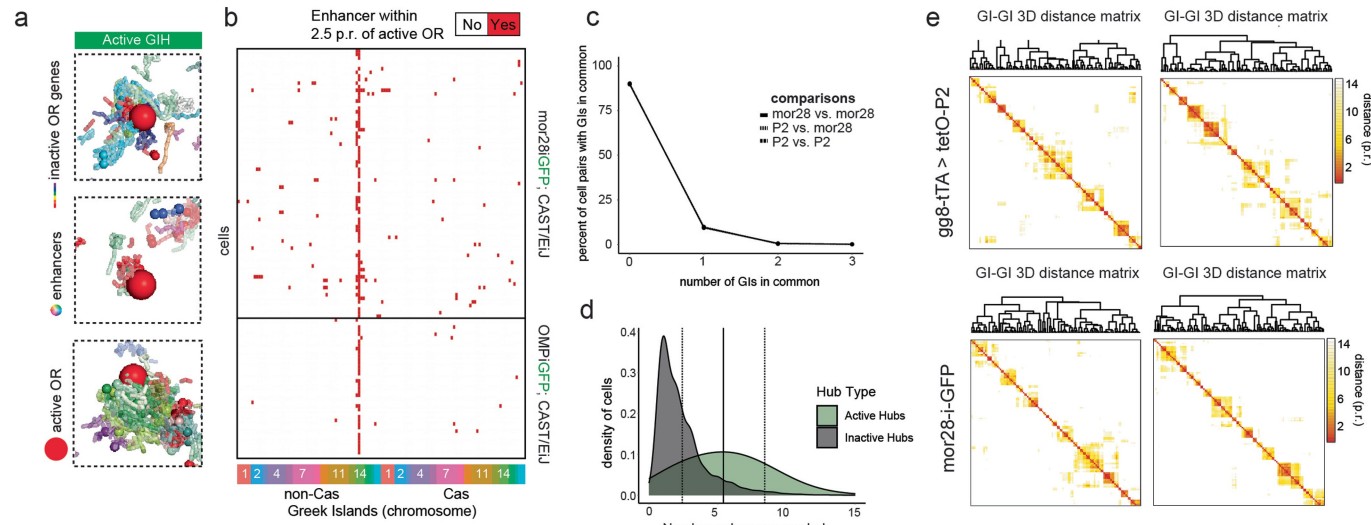

**Extended Data Fig. 5 | Deciphering the variability of active GI hubs.**
(**a**) GI hubs encompassing the active P2 allele in 3 gg8tTA>tetOP2 representative nuclei. (**b**) Binary arrays generated for each cell (rows) depict GIs within <2.5 p.r. of the mor28 allele (non-Cas, chr14) when it is active (top, mor28-i-GFP, n = 74 cells), or inactive (bottom, OMP-i-GFP, n = 40 cells). (**c**) 36 mor28iGFP nuclei and 42 *gg8tTA>tetOP2iGFP* nuclei were compared between themselves and each other for overlap in the combination of *trans* GIs in their active GI hub

(mor28-vs-mor28, n = 630 cell pairs; P2-vs-P2, n = 861 cell pairs; mor28-vs-P2, n = 1453 cell pairs). GIs from chr7 and chr14 are excluded from this analysis to examine GIs that are *trans* to both P2 and mor28. (**d**) Hierarchical clustering of GI spatial relationships was performed on Dip-C models to extract the relative size of the active GI hub (5.42 ± 3.00 GIs, n = 161) vs. inactive GI complexes (2.39 ± 1.84 GIs, n = 7,990) across all cells. (**e**) Hierarchical clustering of GI spatial relationships in a 2 gg8tTA>tetOP2 and 2 mor28-i-GFP nuclei.

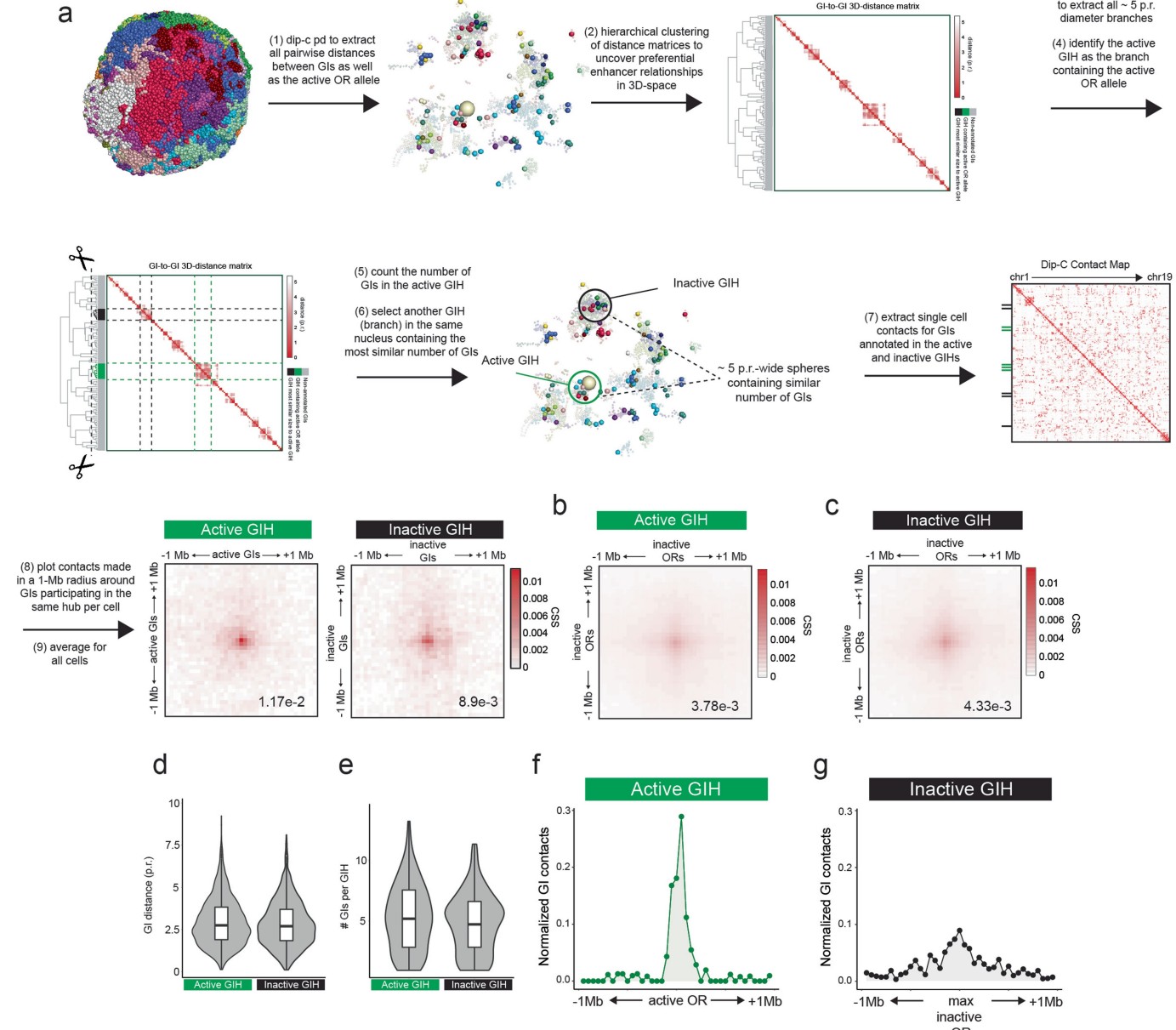

**Extended Data Fig. 6 | Comparing genome architecture of active and inactive GI hubs.** (**a**) Methodology for extracting active and inactive GI hubs of most similar size and concentration from modeling and examining contact specificity. (**b**–**c**) Heatmaps of interchromosomal contacts (CSS) made between (e) OR genes in the active GIH (CSS = 0.00378, n = 132 cells with contacts pooled from 2 independent experiments) and (f) OR genes in the inactive GIH (CSS = 0.00433, n = 113 cells with contacts pooled from 2 independent experiments). (**d**) Pairwise distance relationships between all GIs in active and inactive GI hubs were measured to confirm the similar size of extracted active and inactive GI hubs (active GI hub, 3.10 ± 1.46 p.r., inactive GI hub, 2.95 ± 1.39 p.r., n = 156 cells pooled from 2 independent Dip-C experiments). (**e**) The number of GIs per

GI hub per cell were measured to confirm the similar density of active and inactive GI hubs (active GIH, 5.59 ± 2.88 GIs, inactive GIH, 5.18 ± 2.48 GIs, n = 156 cells pooled from 2 independent Dip-C experiments). Cells with no GIs in the hub were excluded from this analysis. Each boxplot ranges from the upper and lower quartiles with the median as the horizontal line and whiskers extend to 1.5 times the interquartile range. (**f**–**g**) Mean *trans* GI contacts at 50-kb resolution to the average active OR in the active GI hub (f) or the average inactive OR making the most contacts to the most similar inactive GI hub (g) (n = 161 cells pooled from 2 independent experiments). After contacts between GIs and 2-Mb around OR genes in all cells are aggregated, contacts in each bin are normalized to the total contacts made in the 2-Mb span.

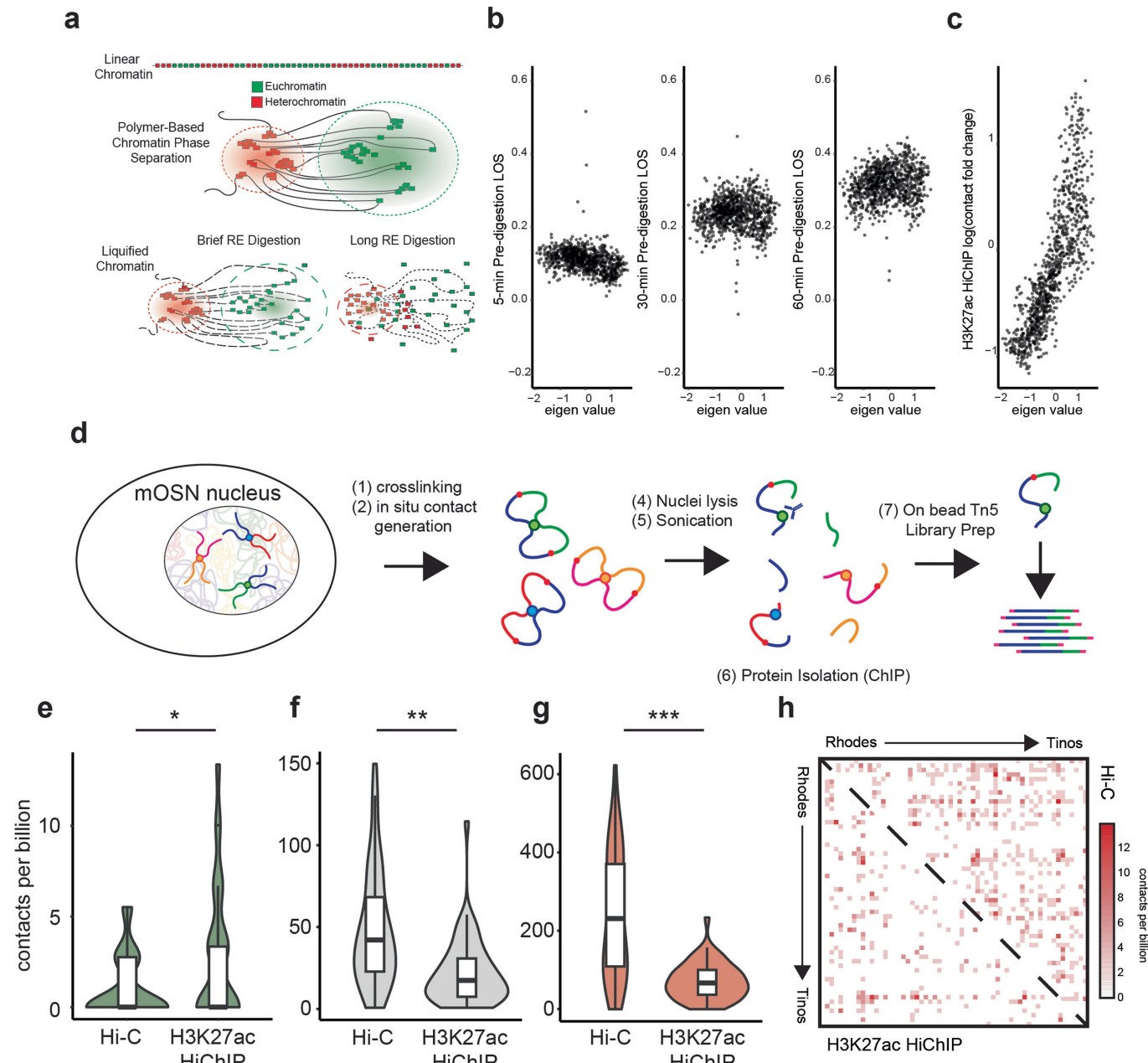

**Extended Data Fig. 7 | Liquid HiC and HiChIP in olfactory neurons.**
(**a**) A schematic of Liquid Hi-C. In Liquid Hi-C, nuclear substructures are distinguished by their varied response to a time course of restriction enzyme digestion preceding Hi-C (pre-digestion). In general, genomic interactions within euchromatin will display a greater loss in contact specificity than in heterochromatin. (**b**) Compartment scores (eigen <0, B compartment, eigen, > 0 A compartment) are plotted against loss in contact specificity (LOS) after 5 m, 30 m, and 60 m of Liquid Hi-C for 1000 randomly sampled bins on chr2. (**c**) Compartment scores (eigen <0, B compartment, eigen > 0 A compartment) are plotted against logFC in contacts after H3K27ac HiChIP, compared to Hi-C, for 1000 randomly sampled bins on chr2. (**d**) Schematic of HiChIP: after cross-linking and contact generation, contacts distinguished by desired

protein marks are immunoprecipitated, to enrich for nuclear substructures containing a protein of interest. (**e**) Interchromosomal contacts (contacts per billion) made by the active OR (left, Two-sided Welch's t-test p = 0.0035, control, 1.03 ± 1.74 cpb, HiChIP, 2.76 ± 3.91 cpb, n = 51 GIs), Greek Islands (middle, Two-sided Welch's t-test p = 1.94e-5, control, 237 ± 160 cpb, HiChIP, 72.0 ± 46.1 cpb, n = 51 GIs), and inactive ORs (right, Two-sided Welch's t-test p = 2.07e-9, control, 47.5 ± 35.3 cpb, HiChIP, 21.6 ± 20.1 cpb, n = 51 GIs), to the 51 *trans* Greek Islands at 10-kb resolution in Hi-C and H3K27ac HiChIP. (**f**) Individual interchromosomal GI-GI contacts at 10-kb resolution for Hi-C and H3K27ac HiChIP. All HiChIP represent averages from two biological replicates. Each boxplot ranges from the upper and lower quartiles with the median as the horizontal line and whiskers extend to 1.5 times the interquartile range.

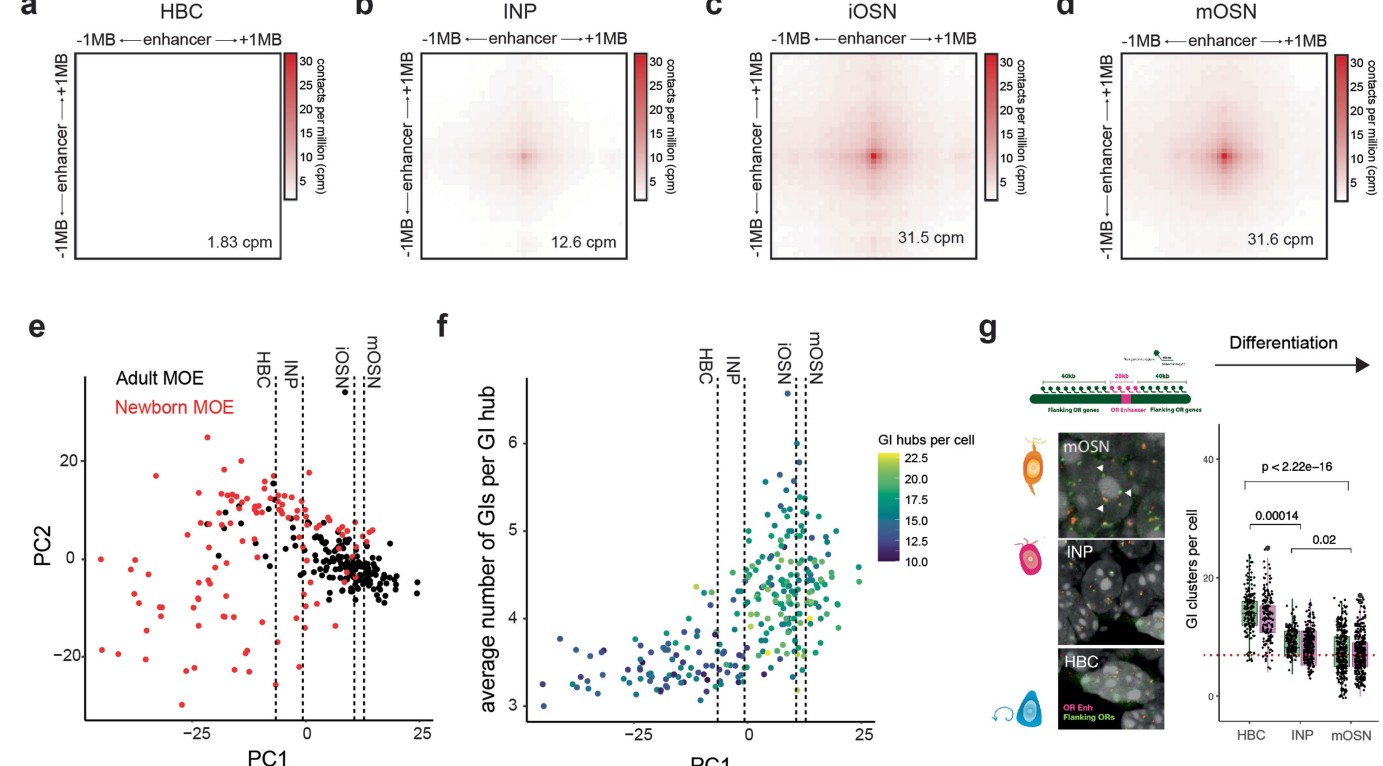

**Extended Data Fig. 8 | Gradual GI hub assembly during OSN differentiation.**
(**a-d**) Interchromosomal GI-GI contacts (contacts per million) in horizontal basal cells (a, HBCs), intermediate neural progenitors (b, INPs), immature OSNs (c, iOSNs), and mOSNs (d). (**e**) PCA on vectors of genome compartment score generated from Dip-C on cells extracted from newborn and adult MOE published in Tan et al. 2019. As demonstrated in Tan et al. 2019, bulk Hi-C data projected onto the PCA reveals that PC1 acts as an axis for OSN development. (**f**) Hierarchical clustering of GI spatial relationships was performed on all MOE cells from Tan et al. 2019, where the number of GI complexes (GI hubs) and the average number of GIs per hub was counted in every cell and projected across

PC1. (**g**) DNA FISH analysis performed in adult MOE sections with a complex probe that detects 30 GIs at once. Cells were annotated as HBCs (n = 371 cells, across 2 independent samples) INPs (n = 179 cells across 2 independent samples), and mOSNs (n = 755 cells, across 2 independent samples), by their morphology and localization across the apical-basal axis of the olfactory epithelium. Each boxplot ranges from the upper and lower quartiles with the median as the horizontal line and whiskers extend to 1.5 times the interquartile range (Welch's two-sample t-test, HBC vs. INP, p = 1.4e-4; HBC vs. mOSN, p < 2.22e-16; INP vs. mOSN, p = 0.02).

## a

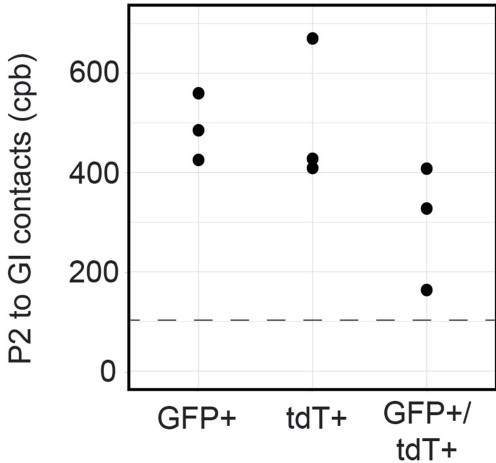 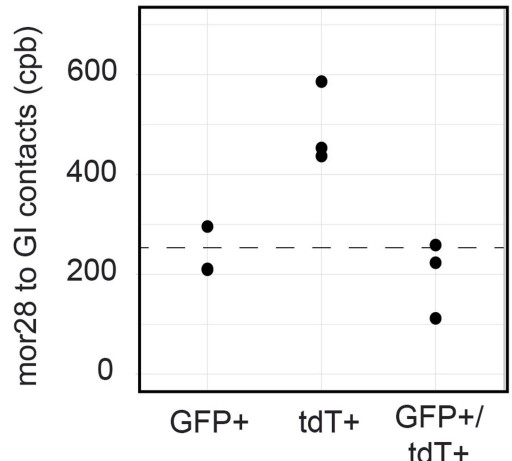

## b

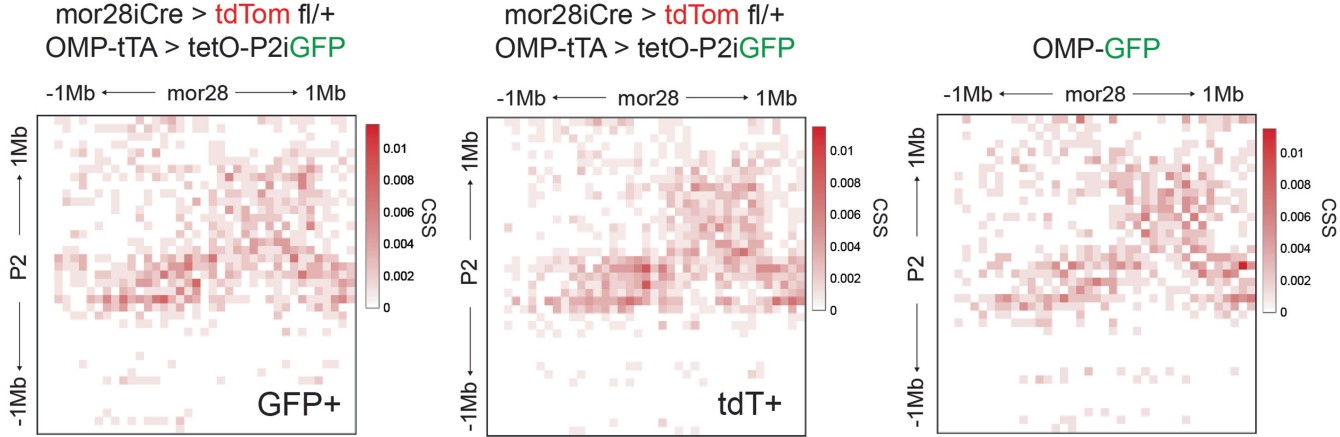

**Extended Data Fig. 9 | P2 to mor28 contacts in OR switching model.**
(**a**) *In-situ* Hi-C in 3 independent biological replicates of GFP⁺, tdT⁺, and GFP⁺/tdT⁺ cells sorted from *mor28iCre>tdT fl/+; OMPtTA>tetOP2iGFP* mice measuring cpb between the P2 (left) or mor28 (right) locus and all *trans* GIs per biological replicate (n = 3). The dashed black line represents contacts between P2 (left) or mor28 (right) and GIs in *OMPiGFP* OSNs (negative control). Contacts between P2 and *trans* GIs are as follows: GFP + 490 ± 67.3 contacts; tdT+ 502 ± 146 contacts; GFP + /tdT+ 299 ± 125 contacts. Contacts between mor28 and *trans* GIs are as follows: GFP + 239 ± 49.5 contacts; tdT+ 492 ± 81.9 contacts; GFP + /tdT+ 198 ± 76.6 contacts. (**b**) Here we show normalized contacts (CSS) in a 2-Mb diameter surrounding P2 and mor28 between (left) GFP+ cells from *mor28iCre>tdT fl/+; OMPtTA>tetOP2iGFP* mice, tdT+ cells from *mor28iCre>tdT fl/+; OMPtTA>tetOP2iGFP* mice, and GFP+ cells from *OMP-GFP* mice.

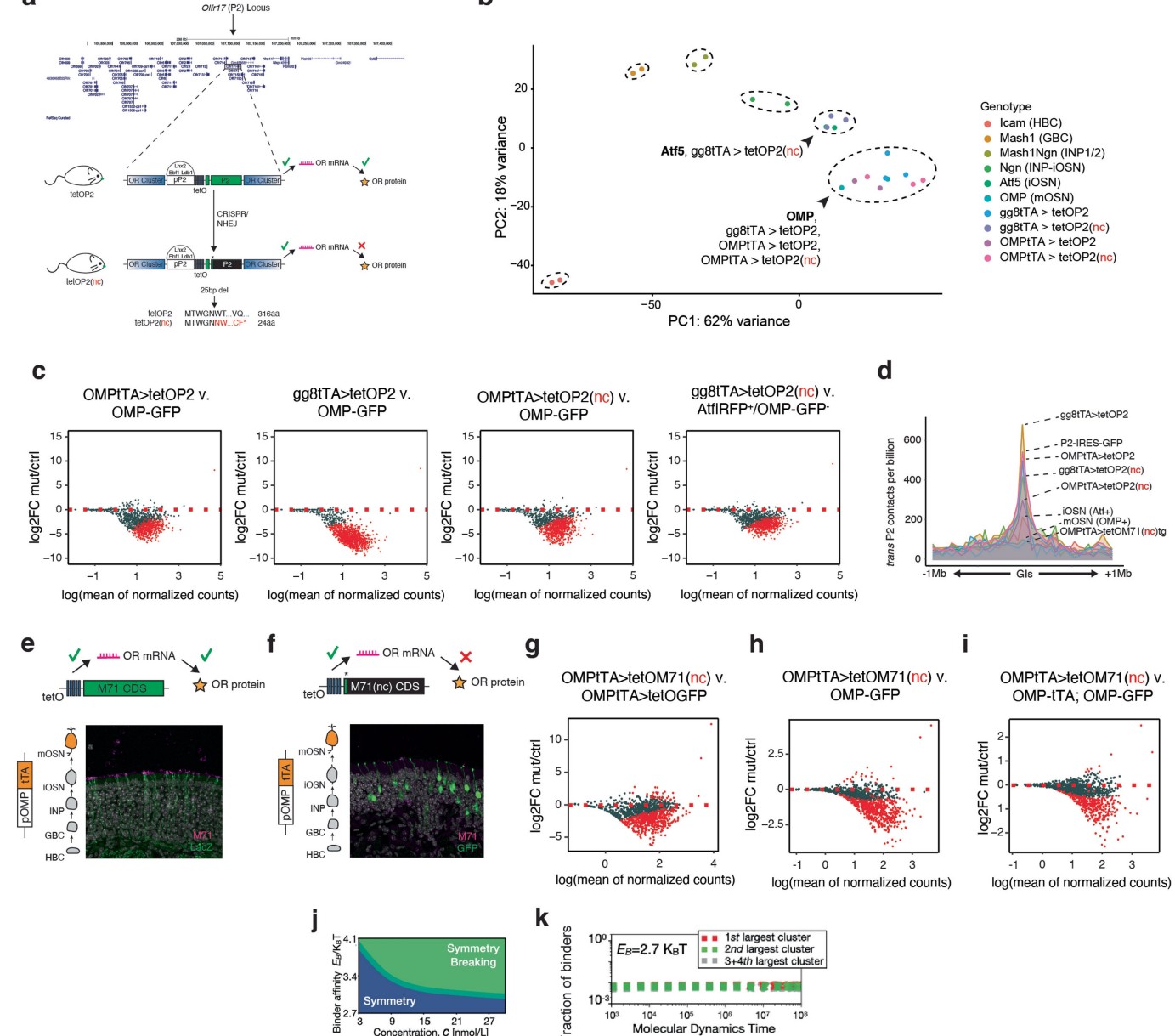

**Extended Data Fig. 10 | Evidence for OR RNA-mediated symmetry breaking.**
(**a**) Schematic of tetO-P2, and tetO-P2(nc) design. tetO-P2 contains a tetO
sequence immediately downstream of the P2 promoter, as well as an IRES-GFP
sequence immediately downstream of the P2 coding exon. Using CRISPR
(see methods), we generated a tetO-P2(nc) allele, which had a 25 bp deletion at
the start of the coding exon, preventing P2 protein production (**b**) PCA of gene
expression data from the olfactory neuronal lineage including P2 mutants:
HBCs (Icam +), GBCs (Mash1 +), early INPs (Mash1/Ngn +), INPs-iOSNs (Ngn +),
iOSNs during OR choice (Atf5 +), and mature OSNs (OMP +). (**c**) MA plots of
DESeq2 normalized OR gene counts generated from RNA-Seq on GFP+ cells
from *OMP-tTA>tetO-P2*, *gg8-tTA>tetO-P2*, *OMP-tTA>tetO-P2(nc)*, and
*gg8-tTA>tetO-P2(nc)*, compared to either GFP+ from *OMP-i-GFP* mice or RFP + /
GFP- cells from *Atf5-i-RFP*; *OMP-i-GFP* mice. (**d**) Interchromosomal GI contacts
to a 2-Mb region surrounding the P2 gene in gg8-tTA>tetOP2 (684 cpb),
P2-IRES-GFP (547 cpb), OMP-tTA>tetO-P2 (503 cpb), OMP-tTA>tetO-P2(nc)

(302 cpb), gg8-tTA>tetO-P2(nc) (421 cpb), iOSNs (217 cpb), mOSNs (103 cpb),
and OMP-tTA>tetO-M71(nc)tg (79 cpb, contacts per billion). (**e–f**) IF of the MOE
from *OMPitTA>tetOM71tg* (**e**) and *OMPitTA>tetOM71(nc)tg* (**f**) mice (M71, magenta;
LacZ, green; GFP, green). Genetic designs of transgenes are illustrated above
images. (**g–i**) MA plots of DESeq2 normalized OR gene counts generated from
RNA-Seq on *OMPitTA>tetOM71(nc)tg* GFP+ cells, compared to 3 GFP+ cells from
independent controls (g) *OMPitTA>tetOGFP*, (h) *OMP-i-GFP*, and (i) *OMP-tTA*;
*OMP-i-GFP*. (**j**) Phase diagram of the system. For affinities $E_B$ and concentrations
$c$ below transition threshold, binders form multiple small clusters and no GIH
symmetry breaking is observed. Conversely, if they are above threshold, at
equilibrium one stable cluster associates to only one GIH and the symmetry is
broken. (**k**) When $E_B < E_{BT}(c)$, the different clusters have all comparable sizes,
their fraction of binders is flat (bottom panel) and symmetry breaking does
not occur.

# Reporting Summary

## Statistics

For all statistical analyses, confirm that the following items are present in the figure legend, table legend, main text, or Methods section.

| n/a | Confirmed | |
|---|---|---|
| ☐ | ☒ | The exact sample size (*n*) for each experimental group/condition, given as a discrete number and unit of measurement |
| ☐ | ☒ | A statement on whether measurements were taken from distinct samples or whether the same sample was measured repeatedly |
| ☐ | ☒ | The statistical test(s) used AND whether they are one- or two-sided <br> *Only common tests should be described solely by name; describe more complex techniques in the Methods section.* |
| ☒ | ☐ | A description of all covariates tested |
| ☒ | ☐ | A description of any assumptions or corrections, such as tests of normality and adjustment for multiple comparisons |
| ☐ | ☒ | A full description of the statistical parameters including central tendency (e.g. means) or other basic estimates (e.g. regression coefficient) AND variation (e.g. standard deviation) or associated estimates of uncertainty (e.g. confidence intervals) |
| ☐ | ☒ | For null hypothesis testing, the test statistic (e.g. $F$, $t$, $r$) with confidence intervals, effect sizes, degrees of freedom and $P$ value noted <br> *Give P values as exact values whenever suitable.* |
| ☒ | ☐ | For Bayesian analysis, information on the choice of priors and Markov chain Monte Carlo settings |
| ☐ | ☒ | For hierarchical and complex designs, identification of the appropriate level for tests and full reporting of outcomes |
| ☒ | ☐ | Estimates of effect sizes (e.g. Cohen's *d*, Pearson's *r*), indicating how they were calculated |

*Our web collection on statistics for biologists contains articles on many of the points above.*

## Software and code

Policy information about availability of computer code

Data collection | Zeiss Zen2012 SP1 (v8.1.9.484) was used for capturing confocal images

Data analysis
BWA 0.7.17
hickit r291
k8-Linux K8: 0.2.5-r80
Bedtools2 v2.26.0
conda 4.11.0
Python v2.7.13
PyMOL(TM) 2.5.3
dip-c (https://github.com/tanlongzhi/dip-c) (81dc83ea824c3503f835ba7b0d333163c46b9aca, Jan 9, 2019)
cellranger-arc-2.0.1
macs2 2.2.7.1
Signac 1.6.0
Seurat 4.1.0
R 4.1.3
monocle3 1.0.0
java 8
nextflow version 19.07.0.5106
Docker version 19.03.2, build 6a30dfc
distiller-nf 0.3.3
cooler, version 0.8.10

Integrated Genome Browser 9.0.0
CutAdapt v1.17
Bowtie2 v2.3.2
Samtools v1.4.1
STAR v2.5.3a
ImageJ 2.0.0
Picard 1.137
DeSeq2 1.24.0

Scripts for all data analysis were written in R and is available under the following github repository (https://github.com/arielpourmorady/Pourmorady_etal.git)

For manuscripts utilizing custom algorithms or software that are central to the research but not yet described in published literature, software must be made available to editors and reviewers. We strongly encourage code deposition in a community repository (e.g. GitHub). See the Nature Portfolio guidelines for submitting code & software for further information.

## Data

Policy information about availability of data

All manuscripts must include a data availability statement. This statement should provide the following information, where applicable:
- Accession codes, unique identifiers, or web links for publicly available datasets
- A description of any restrictions on data availability
- For clinical datasets or third party data, please ensure that the statement adheres to our policy

Data that support the findings of this study are deposited in a GEO superseries with the following accession number: GSE230380. Dip-C data from previously published work from our lab were used for Fig. 2b and Fig. S5b (GSE158730). Dip-C data from Tan et al. were used to render PCAs on single-cell chromatin compartments in Fig. S8e-f (GSE121791). Previously published Hi-C data from our lab were used for Fig. S8a-f, and are publicly available at https://data.4dnucleome.org/ under the following accession numbers: 4DNESH4UTRNL, 4DNESNYBDSLY, 4DNES54YB6TQ, 4DNESRE7AK5U, 4DNES425UDGS and 4DNESEPDL6KY. Genome assembly for mm10 genome which was used for deep-sequencing read alignment can be found at https://www.ncbi.nlm.nih.gov/datasets/genome/GCF_000001635.20/.

## Human research participants

Policy information about studies involving human research participants and Sex and Gender in Research.

| | |
|---|---|
| Reporting on sex and gender | N/A |
| Population characteristics | N/A |
| Recruitment | N/A |
| Ethics oversight | N/A |

Note that full information on the approval of the study protocol must also be provided in the manuscript.

# Field-specific reporting

Please select the one below that is the best fit for your research. If you are not sure, read the appropriate sections before making your selection.

☒ Life sciences          ☐ Behavioural & social sciences          ☐ Ecological, evolutionary & environmental sciences

For a reference copy of the document with all sections, see nature.com/documents/nr-reporting-summary-flat.pdf

# Life sciences study design

All studies must disclose on these points even when the disclosure is negative.

| | |
|---|---|
| Sample size | Dip-C was performed on sorted cells pooled from at least three mice and performed independently on gg8tTA>tetO-P2 and mor28iGFP mouse lines. Dip-C sample size was based off Tan et al. 2019, where authors also examined Dip-C models and contacts of nuclei in 409 cells which represented a mixed population of cell types of the MOE including mOSNs. Since we performed Dip-C on a pure population of sorted mOSNs from the MOE, we targeted our sample size to 192 cells and generated 161 high quality cells. The multiome was generated from one mouse and reproduced independently from a second mouse (shown is supplement). For bulk high throughput sequencing experiments, a sample size of two to three independent biological replicates was selected.  For Hi-C experiments (including HiChIP and Liquid Hi-C), at least two independent samples were generated as per 4DNucleome guidelines (https://drive.google.com/file/d/1-NEldtpuDuYXcbWngETltNPCRv0RZlIK/view). Furthermore, for HiChIP and ATAC-seq experiments, at least two independent samples were generated as per ENCODE guidelines (https://doi.org/10.1101/gr.136184.111). For RNA-Seq, 2-3 biological replicates were selected because multiple statistical approaches have been developed to allow identification of significantly changed loci or genes from two biological replicates of high throughput sequencing data (e.g. edgeR, DEseq2). These approaches work by using the large number of genes/loci measured to |

analyze and model the dispersion and variance within and between replicates, thereby allowing the identification of genes/loci with significant differences between conditions. Wherever possible, additional biological replicates were included.

| Data exclusions | For the multiome, cells was quality controlled by retaining cells with a TSS enrichment > 1, nucleosome signal < 2, ATAC counts between 1,000 and 10,000, and RNA counts between 1,000 and 25,000. For scATAC data from the multiome, peaks within ENCODE black-listed regions (https://sites.google.com/site/anshulkundaje/projects/blacklists) were excluded from analysis. This exclusion was pre-established. For Dip-C, we excluded cells that had contacts < ~400,000 contacts, cells that had a low contact-to-read ratio, and cells that had high variability in 3D structure across computational replicates. |
|---|---|
| Replication | At least two independent biological replicates were performed for each experiment. Replicate experiments yielded the same results. |
| Randomization | No experiments were performed with live animals. For the purpose of purifying primary cells, animals of similar age were grouped by genotype and sorted together. |
| Blinding | Animals were used as a source of tissue and primary cells, so knowledge of genotype was required for proper handling and cell sorting. Every genotype analyzed contained multiple transgenic alleles, consequently only a small fraction of mice had the experimental and control genotypes; therefore, knowledge of genotype was essential for deciding which animals to use for FAC-sorting and downstream experiments. Performing the reported experiments in a blind fashion would increase the cost of the analysis by 10-fold, and would result in unnecessary preservation of mice that should be euthanized. Finally, because our analysis was performed in transgenic mice, blinding is not possible because the genotypes become apparent in the analysis itself. |

# Reporting for specific materials, systems and methods

We require information from authors about some types of materials, experimental systems and methods used in many studies. Here, indicate whether each material, system or method listed is relevant to your study. If you are not sure if a list item applies to your research, read the appropriate section before selecting a response.

## Materials & experimental systems

| n/a | Involved in the study |
|---|---|
| ☐ | ☒ Antibodies |
| ☒ | ☐ Eukaryotic cell lines |
| ☒ | ☐ Palaeontology and archaeology |
| ☐ | ☒ Animals and other organisms |
| ☒ | ☐ Clinical data |
| ☒ | ☐ Dual use research of concern |

## Methods

| n/a | Involved in the study |
|---|---|
| ☒ | ☐ ChIP-seq |
| ☐ | ☒ Flow cytometry |
| ☒ | ☐ MRI-based neuroimaging |

## Antibodies

| Antibodies used | GFP (chicken anti-GFP ab13970)<br>M71 (Lomvardas et al. 2006, Gilad Barnea Brown University)<br>P2 (Olfr17 antibody were raised in guinea pig, Gilad Barnea Brown University)<br>LacZ (abcam ab4761)<br>anti-chick IgG conjugated to Alexa-488 (Jackson ImmunoResearch, Code 103-434-155, RRID: AB_2337390 polyclonal).<br>anti-guinea pig IgG conjugated to Cy3 (Jackson ImmunoResearch, Code 106-165-003, RRID: AB_2337423 polyclonal).<br>H3K27ac antibody (Abcam GR323193701) |
|---|---|
| Validation | LacZ (abcam ab4761) - Manufacturer states that antibody is validated for immunofluorescenc<br>GFP (chicken anti-GFP ab13970) - Manufacturer states that antibody is validated for immunofluorescence<br>anti-guinea pig IgG conjugated to Cy3 - Manufacturer states that antibody is validated for immunofluorescence<br>anti-chick IgG conjugated to Alexa-488 - Manufacturer states that antibody is validated for immunofluorescence<br>H3K27ac antibody (Abcam GR323193701) - Manufacturer states that antibody is validated for immunofluorescence<br>M71 antibody - Validated in Lomvardas et al. 2006 and in this study by positive staining in the cilia of neurons expressing the tetO-M71-LacZ transgene and negative staining in tetO-M71(KO)-IRES-GFP mice.<br>P2 antibody - Validated in this study by positive staining in the cilia of neurons expressing the tetO-P2iresGFP mutant gene and negative staining in tetO-P2(KO)iresGFP. |

## Animals and other research organisms

Policy information about studies involving animals; ARRIVE guidelines recommended for reporting animal research, and Sex and Gender in Research

| Laboratory animals | The mice are housed in individually ventilated cages (IVC), and the rooms are maintained at 72F and 30%RH. The light cycle is 12:12 (lights on at 7:00am off at 7:00pm). Mice are fed irradiated PicoLab Rodent Diet 20, and are housed in cages with irradiated corn cob bedding. This study used several mouse lines (mus musculus) on mixed C57BL/6J and 129 backgrounds. Experimental mice were generated by crossing mice from the following lines together in different combinations. All experiments were performed on male and female adult mice between 5-12 weeks of age. |
|---|---|

Experimental genotypes were:

Gng8(gg8)-tTA - Male and female adult mice were used
OMPitTA - Male and female adult mice were used
tetO-P2-IRES-GFP - Male and female adult mice were used
tetO-P2(KO)-IRES-GFP - Male and female adult mice were used
tetO-M71-LacZ - Male and female adult mice were used
tetO-M71(KO)-IRES-GFP - Male and female adult mice were used
tetO-GFP - Male and female adult mice were used
Olfr1507-ires-GFP - Male and female adult mice were used
CAST/EiJ - Male and female adult mice were used
Atf5irep - Male and female adult mice were used
OMP-ires-GFP - Male and female adult mice were used
Ngn1-GFP - Male and female mice were used a 1-2 weeks of age.
mor28-ires-cre - Male and female adult mice were used
Rosa26(LSL-tdTomato/+) - Male and female adult mice were used

Data published in Horta et al. 2019 were generated from the following mice and used in this paper:

Krt5Cre;R26R-tdtomato-GFP - Male and female adult mice were used
Olfr17-ires-GFP - Male and female adult mice were used

| | |
|---|---|
| Wild animals | None |
| Reporting on sex | All experiments were performed on both sexes and findings apply to both sexes. |
| Field-collected samples | None |
| Ethics oversight | IACUC - Columbia University |

Note that full information on the approval of the study protocol must also be provided in the manuscript.

# Flow Cytometry

## Plots

Confirm that:

☒ The axis labels state the marker and fluorochrome used (e.g. CD4-FITC).

☒ The axis scales are clearly visible. Include numbers along axes only for bottom left plot of group (a 'group' is an analysis of identical markers).

☒ All plots are contour plots with outliers or pseudocolor plots.

☒ A numerical value for number of cells or percentage (with statistics) is provided.

## Methodology

| | |
|---|---|
| Sample preparation | The FACS plot in Supplemental Figure 9a is a representative FACS plot illustrating our gating strategy for purifying GFP+/tdT-, GFP-/tdT+, and GFP+/tdT+ cells. This gating strategy was also used to isolate GFP+ cells in on our other experiments from other mouse lines. Briefly, for this plot, MOE were dissected from OMPtTA>tetOP2iGFP, mor28iCre>tdTom fl/+, mice and dissociated to obtain a single cell suspension of the MOE which was suspended in sort solution of 2% FBS/1xPBS, 4 mM MgCl2, and 1:1000 DAPI. 300 uL of sort solution was used per OE. |
| Instrument | Beckman Coulter Low Flow Astrios EQ |
| Software | Acquisition Software: Summit v3.2.1 |
| Cell population abundance | For OMPtTA>tetOP2iGFP, mor28iCre>tdTom fl/+, experiments: GFP+ 5.44%, tdT+ 0.1%, and GFP+/tdT+ 0.01%. |
| Gating strategy | FSC/SSC gates are set to eliminate small debris as well as larger and more granular cells that are not olfactory neurons. Doublet discrimination is performed by plotting SSC Height vs. SSC area to eliminate events with larger area, indicating they are doublets. This is also done by comparing SSC Height vs. SSC width to eliminate events with greater width which would represent doublets. Gating is also performed to eliminate DAPI positive cells. Both GFP+ and tdT+ cells are three decades brighter than negative cells and are clear cell populations which can be easily defined. |

☒ Tick this box to confirm that a figure exemplifying the gating strategy is provided in the Supplementary Information.

