## [Peer Review File · Nature]

Manuscript Title: A symmetry breaking process proposes non-coding functions for olfactory receptor RNAs

Reviewer Comments & Author Rebuttals

Reviewer Reports on the Initial Version:

Referees' comments:

Referee #1 (Remarks to the Author):

Lomvardas and colleagues aim to address a fundamental question about how olfactory receptor neurons transcribe only one of ~2000 distinct olfactory receptor genes. This olfactory receptor choice model represents a paradigm example of allelic competition where stochastic choice leads to a quantitative and deterministic outcome that is essential for function. Yet, what mechanisms exist to explain such random counting problems in olfactory receptor choice or other contexts is still largely unknown. To address this, the authors generate a comprehensive single cell atlas of 3D genome conformation changes and chromatin accessibility that culminates in monoallelic olfactory receptor (OR) gene expression. Olfactory receptor genes organize around enhancer dense regions called "Greek Islands" (GI). The authors show that the active GI localizes specifically with the active OR, whereas inactive GIs form less specific interactions with inactive ORs. To explore how competition of protein factors between GIs might impact stochastic choice, they present a physical model whereby concentration leads to a phase transition and symmetry breaking driving accumulation over a single GI. The authors explore the idea that the olfactory receptor pre-mRNA might be required for this symmetry breaking. They test this using transgenes with coding and frameshifted ORs to distinguish between the RNA and protein products. Based on these observations, the authors propose a model whereby the RNA transcribed from the olfactory receptor gene acts to select the monoallelically expressed allele and silence the other olfactory receptor alleles.

Overall, this paper and the proposed model represents an elegant and important new mechanism to explain a long-standing enigma in gene regulation. The idea that the RNA product of a protein-coding gene can play a direct role in symmetry breaking – reinforcing activation of itself and ensuring silencing of the other alleles – is a novel mechanism that will have important implications far beyond the olfactory receptor system. While I am extremely enthusiastic about the concepts presented in this work, I do have some concerns about how this paper is presented and some of the technical aspects of the experiments that I think will be important to address in order to fully demonstrate this model.

As currently written, the paper appears to represent two distinct parts that do not obviously connect. The first part (Figures 1-3) that focuses on the single cell atlas of structural changes and enhancer interactions between the GIs and the second part (Figures 4 and 5) which focuses on the physical model and the role of RNA in monoallelic expression. While the authors may have some logic of why they felt the former components were critical in leading to the model and the RNA components, it was not obvious to me in reading the paper why these descriptions of GI dynamics were relevant to this mechanism or how they connected. None of these specific dynamics were utilized in establishing the model or exploring the RNA role.

Related to this point, it was also not obvious how they went from the proposed model to the RNA hypothesis that they tested. Based on my understanding of the model as presented, there is no "RNA component" included. As such, how the idea of GIs serving as a scaffold for a self-aggregating protein led to them testing a role for RNA is unclear. Indeed, the Figure 4C illustration

suggests that it's the GI hubs that drives aggregation/symmetry breaking. More generally, the authors never test the predictions of this model beyond the gradual silencing of all but one GI. The central mechanistic claims of the model, including whether there is a concentration-limited protein factor, whether this protein factor changes affinity in a local hub, or whether this factor is essential for symmetry breaking, remain untested. As such, it is not clear to me why this model is presented in the Results section since it is not directly tested and does not directly relate to the previous mapping observations or the subsequent RNA hypothesis.

An alternative approach would be to explore this model (which is an adaptation of the symmetry breaking model presented by Nicodemi and Prisco 2007 in the context of X inactivation) in the Discussion as a potential explanation linking their observed 3D genome and RNA results. This would allow the authors to more thoroughly explore/highlight many of the remaining unknowns that still need to be determined and tested. For example, how the RNA component might actually lead to symmetry breaking. Based on my inference from their model, there is presumably a missing protein factor that the RNA likely binds to. What is this protein factor? While I understand that asking the authors to find this protein and characterize it is beyond the scope of this paper, they should certainly explain what some properties of such a protein might be (e.g. RNA binding, ability to undergo concentration-dependent phase transition, etc.) and note the need to identify these components in future work.

Finally, I have a technical issue with the authors' claim that the non-coding transgene experiment demonstrates the centrality of the RNA to OR choice. The authors perform a series of clever experiments to distinguish between the RNA and protein products. The key experiment uses a Tet-inducible OR allele that they know is preferentially selected when overexpressed to make a version of this allele that impacts protein production but leaves the remaining sequence otherwise mostly intact. While the non-coding allele leads to lower expression of other OR genes, it does not lead to Tet-independent expression of the transgene itself. If indeed the RNA is critical for monoallelic expression, the non-coding transgene should reduce expression elsewhere and sustain expression of itself. The authors hypothesize that the encoded protein is likely required for sustained expression of the active OR gene, if not for its initial selection. This is certainly a reasonable explanation, but it is never actually tested. Without this demonstration, the authors have not actually demonstrated that the RNA can drive monoallelic choice. One simple experiment that would address this is to express the OR protein exogenously (e.g. by mRNA or protein transfection) alongside the mutant allele and see if this causes sustained expression of the non-coding OR transgene. There might be other experiments to address this that the authors are probably better suited to define in their system and I would be amenable to any that would address this concern.

Referee #2 (Remarks to the Author):

In a tour de force effort, Pourmorady et al address an important unanswered question in genome biology, of which the answer will have broad impact across multiple biological systems. Specifically, in olfactory sensory neurons (OSNs) a single olfactory receptor gene must be chosen and expressed in each neuron, and this requires the selection from 126 possible enhancers (so-called Greek islands – GIs) distributed across the genome. In post-mitotic OSN progenitors, multiple OR genes are expressed, and then an unknown mechanism later in OSN maturation kicks in to ensure that only one OR allele is expressed for the remaining lifetime of each given neuron. In this work, the authors chase down the answer to this mechanism using multiple bleeding edge single-cell and ensemble sequencing technologies as well as biophysical modeling. The impact of the work is clearly at the level appropriate for Nature, and the improvements for revision can be focused on gathering critical multimodal single-cell data to test the authors' compelling hypothesis. Below we provide comments and ideas which we hope are useful to the authors in revising the work.

Comments

>>Figure 1 text is well written and important conclusions are shown. First, using multi-modal single-nucleus ATAC-seq and RNA-seq in the same nucleus for neurons from the main olfactory epithelium (MOE), authors show that: (1) Greek island enhancers are much more variable in accessibility across a cohort of single mOSNs compared to conventional enhancers. (2) GI enhancers exhibit the highest accessibility during the INP \diamond iOSN developmental stage when multiple OR alleles are known to be expressed. (3) Overall summed GI enhancer accessibility declines substantially in mOSNs during which the OR single allele choice is maintained. Overall, the data together are consistent with the authors' proposed model that the reduction in total summed enhancer accessibility (pruning). A bit of additional re-analysis will help to further support the concept that sustained accessibility of a single enhancer/GI across single cells coincides with single OR allele choice.

1. Figure 1a- could clusters be determined only from the RNAseq, thus allow pure clusters from transcriptional markers and then pure analysis of ATAC at promoters and GIs in 1b and following figures? The reason this might be helpful is that ATACseq signal at promoters is often not a correlative read out of expression, and ATAC signal at enhancers is only predictive of enhancer activity in key developmental contexts (as might be the case in the OSN allele choice system). Minor: It might be helpful to explicitly state the arrow is the direction of lineage within the figure caption.
2. Figure 1b is rather exciting – readers would benefit from a supplementary figure containing 1b implementation for all 126 GI enhancers as well as for the promoters of all OR gene alleles. Annotating 1b to be clear it is only in mOSN cell type and not for the earlier lineages would be helpful.
3. It would be useful and informative to see the same type of Figure 1e in the supplement but for a line plot for every GI and OR promoter overlaid in the same 1-2 plot(s). Also, if this format replaced Figure 1f, there might be less mental overhead of seeing the trends match the average/sum in Figure 1e.
4. Do the colors (blue, green, red) match the colors in the line plot & marker genes in Figure 1d and 1e? I would have expected green iOSNs to be placed prior to blue mOSNs in pseudotime in Figure 1e.
5. The critical concept in 1g-i is that the computation involves cumulative/summed accessibility. In pink, the accessibility over the top 71 GIs is shown as plateauing in the last mOSN stage – yet in blue the same pattern is shown only for the top OR. Is the idea that these 71 enhancers regulate 71 different ORs across this cellular population (1 GI per 1 OR per cell – many cells)? If so, it might be more relevant to show in blue the plot for the top ORs that would be relevant for these 71 GIs.
6. Since this particular data is at single cell resolution, can the authors show that in each cell there is a top GI and a top OR that correlate with high accessibility in mOSNs – and that this is different in each cell? To me, this seems like a small reanalysis which would have outsized exciting impact on the paper.

>>Figure 2 – Using elegant transgenic models, the authors conclude that for hubs of enhancer-to-OR gene contacts the number of hubs per cell and the size of hubs (number of GI enhancers per hub per cell) does not change whether the OR gene is active or inactive. They also conclude that cis contacts between the OR gene and GI enhancers are stronger in hubs when the OR genes are active vs. inactive.

7. Readers would benefit from a different resolution for panel Figure 2c – difficult to see text and data
8. The 5 particle limit was based on imaging data that suggests that enhancer-promoter distances below 300-400nm correlate with transcription. A supplemental figure showcasing this relationship would be very helpful for the readers.

9. The writing on the cellular populations was not crystal clear for a generalized audience and would benefit from a schematic. I could gather that there were two OR genes under consideration – P2 and mor28. After reading the description of the genetic system several times, I could guess that mOSNs expressing either of these 2 OR genes were labeled with GFP via an IRES. For mor28, it is a naturally highly expressed OR gene and GFP+ cells could be sorted. For P2, it is tet inducible and the P2 OR allele remains on long after tet is removed, so the mOSNs with this gene on stay GFP positive and can be sorted. The breeding to another mouse genetic background allows alleles to be separated. As a reader, it took a lot of time to think through this independent from the brief text, so revising this part for clarity on the models might keep readers engaged to keep reading the rest of the paper.

10. It was not fully clear what should be deduced from Figure 2c – is it meant to claim that a GI enhancer on chr14 is physically proximal (proximity score as predicted by the polymer model) to the mor28 gene in the majority of sorted cells with the mor28 gene expressed? And the GI enhancer on chr7 is physically proximal to the P2 in the majority of sorted cells with the P2 gene expressed? I think I would have benefited from knowing more clearly if this plot was measuring proximity between the one enhancer and the OR gene allele of interest – or if it was measuring proximity among multiple GIs on that chromosome and not assessing contact with the OR gene at all. This figure would increase in value if the same plots were made for a negative control of 70-80 sorted cells that are not GFP+ -- they should not have the pattern shown in Figure 2c. Would the signal on chr14 and chr7 be gone completely or only reduce in the number of cells?

11. The purpose of Figure 2d might be achieved more directly by showing the full distributions of the number of active vs. inactive hubs per cell. I struggled to interpret data showing the distribution only within or above a certain average threshold – I'd prefer to see all of the data and the comparison across conditions. Would like to see explicitly the metrics of Number of Hubs per cell and Number of Enhancers per Hub per cell – full distributions for both Inactive and Active hubs.

12. The text for this section was a bit difficult to parse in terms of weighing it against the figure. My understanding is that the main conclusion is that there is very similar distributions of the number of enhancers per hub/cluster per cell whether the hub is inactive (i.e. with an inactive OR gene) vs. active (i.e. with an active OR gene). However, when I look at panel 2h/2i – the inactive OR genes are not forming contacts with inactive GIs – so is there any evidence for inactive GI – inactive OR hubs from these data? All I could gather is that 2f shows inactive GIs are together, 2f shows inactive ORs can be together, but 2h-I shows inactive GIs and inactive ORs are actually not in hubs. Could this be clarified to support the claims with a bit of tweaks in the analysis?

13. How are inactive GIH determined in DIP-C data since it is not multimodal? Is it assumed that only one OR gene can be active (the sorted population GFP allele) – so all other GI-OR contacts can automatically be inferred to be inactive?

14. Figure panels 2h/i are not labeled in the 2g caption

15. Figure 2j – add yellow sphere as inactive GI enhancers to key – I did not see congruence between the drawn model and results from Figure 2g-i. At first glance, inactive ORs are not in hubs with inactive GIs?

16. Supplemental Figure 2: One thing I am curious on is why it is first needed to project DIP-C onto a polymer model to identify active and inactive hubs? Is it not possible to identify hubs directly from the Dip-C contact matrix? It would be excellent to see and observe the same hubs directly via the Dip-C contact matrix – it would be useful to clarify in the text if this is not possible in single cell data and the intermediate of the polymer model is required for transitioning a sparse matrix to insight.

>>Figure 3 – the overarching conclusion is that hubs with active OR alleles are enriched in H3K27ac, whereas hubs with inactive OR alleles are enriched in heterochromatin.

17. Can a single-cell H3K9me3 Cut and Run assay confirm that heterochromatin is enriched at GI enhancers when they are in a hub with inactive OR alleles? This would be a direct test of the hypothesis. There might be a technical barrier to doing this because you would need a multi-modal assay to do single-allele gene expression and heterochromatin ChIP in the same cell. If so, it

would make sense that this is not feasible at this time.

>>Figure 4 – the conclusion and model from Figure 4 is perhaps the most important of the paper. The model asserts that GI enhancers are ubiquitously activated at a low level along with sporadic OR transcription in the INP stage, and then subsequently a few choice enhancers and OR alleles are upregulated along with multi-enhancer transcriptional hubs in iOSNs. It is then only in mOSNs that many hubs are shut off and the genes silenced, whereas one hub of enhancers and the singular chosen OR allele stays on whilst others are all silenced.

18. The model is compelling and exciting – yet, the data to support is sparse in the current form. This is the one part of the paper that would really benefit from addition of another experimental assay given the sweeping impact of the model.

19. It is particularly difficult to infer the model from the data shown in Figure 4a. Although not introduced explicitly, I inferred that the authors are showing either scDipC data or new ensemble Hi-C data for these 4 developmental populations introduced in Figure 1a (same cells also used for the 10x multiome analysis). Can the authors make it clear what data was generated and used to make these images? Did they go back to the 4 cell types from Figure 1 and create more DipC libraries?

20. Figure 4a and 4b – these two analyses do not help support the models because we cannot parse inactive OR-inactive GI hubs from active OR-active GI hubs, nor active GI-GI hubs from inactive GI-GI hubs. Yet, if this was explicitly done, it would transform the paper and back up the conclusions. There is indeed a technical challenge of gathering the data to support this claim – as it would require allele level gene expression in the same cell as single-cell Hi-C like measurements between all GI enhancers and all OR gene isoforms in an allele-specific manner. Since this model is very important to the paper: Can the authors pursue a multi-modal genomics or imaging technology to drive these conclusions home? For example, sequential Oligopaints for a few OR alleles and the GI enhancers coupled with merFISH for the RNA of those chosen OR alleles? If the authors deem that a bulk multiomic technique might be useful to gather the data support – Hi-Car is another option: <https://www.sciencedirect.com/science/article/pii/S1097276522000983>

21. If the authors deem that conducting multimodal techniques is out of scope, the wording can be revised to indicate that Figure 4A is a model that is of high impact yet cannot be fully tested to conclusion at this time in science and technology development.

22. The authors present the intriguing idea "at the earlier stage of polygenic OR transcription when Lhx2 and Ldb1 concentrations are highest (Fig. S5a) each of the multiple hubs may have similar protein composition. As the concentration of these two factors decreases with differentiation, the symmetry between hubs becomes broken, with Lhx2/Ebf/Ldb1 accumulation in one GI hub". Multimodal imaging of Lhx2/Ldb1 along with Oligopaints would likely be needed for single cell evidence of equal Ldb1 concentration at all GI-OR hubs prior to single allele choice. Can the wording be revised ?

Referee #3 (Remarks to the Author):

The manuscript by Lomvardas and colleagues uses an impressive array of technically dizzying single cell genomics techniques to probe the mechanisms driving the singularity of odorant receptor (OR) expression by mature olfactory sensory neurons (mOSNs). Previous work by this group has demonstrated the presence of ~70 OR cluster enhancers called Greek Islands (GIs) that mediate intra- and inter-chromosomal interactions that organize the chromatin of mature OSNs into discrete clusters or Greek Island Hubs (GIHs) in an Lhx2/Ebf/Ldb1-dependent mechanism. This architecture is required for high-level expression of a single OR per mOSN. A curiosity arises from other studies, which demonstrated that during the transition from immediate neuronal precursors (INPs) to immature OSNs (iOSNs), multiple ORs are expressed at a low level before one is selected for singular, high-level expression in mOSNs. The authors posit that the properties of the GIHs must change during this transition.

Consistent with this hypothesis, ATAC-seq shows peak accessibility of GIs at the INP-iOSN transition, corresponding to the onset of multi-OR expression per cell, with accessibility falling yet maintained at supra-basal levels through the mOSN stage. Using Dip-C to reconstruct the 3D nuclear chromatin architecture, they further find that each mOSN has on average five GIHs: one active hub that includes the OR chosen for expression (experimentally defined, as these were the cells that were purified by FACS), and ~four “inactive” hubs containing GIs in which the active OR allele is absent. Further analysis by Hi-C, liquid Hi-C, and H3K27ac ChIP-seq demonstrate differences in the number of inter- and intra-chromosomal contacts and biochemical properties of active and inactive GIHs.

The above observations raise the question of how the active GIH is ultimately favored over the other GIHs that wind up being suppressed. The authors conceived of a symmetry breaking model to account for the transition to enhanced contacts within the active GIH over those in the inactive GIHs. On one level, the model seems attractive and plausible enough. However, as intriguing as they may be, it is unclear to this reviewer whether the experiments that follow address the model, at least in terms of specifics. Using the endogenous MOR28 knockin allele or the broad and early-expressed P2 allele, they show singularity of OR expression is maintained either with intact OR protein coding region or with a frame-shifted mutation that abrogates OR protein expression. The number of GIH contact sites appears similar across all cases examined. Consistent with previous studies in the literature, transgene expression of a non-protein coding M71 mRNA results in suppression of other OR expression. From all of the above, the authors conclude that OR mRNA, and not OR protein per se, is required to break symmetry and promote singularity in OR expression. Unfortunately, without knowing the configurations of the GIHs in the M71 mRNA over-expression experiment, it cannot be concluded that this effect is due to symmetry breaking at the level of GIHs. Missing is an experimental manipulation that results in persistent polygenic OR expression. This is a critical point that either makes or breaks the interpretation of the final set of experiments in Figure 5. From the data presented I just don't see an incisive experiment that allows the authors to draw their conclusions about mechanism.

It is also interesting to consider how the results in this study relate to the phenomenon of OR gene “switching” that was originally discovered by Sakano and fleshed out mechanistically by this group. The authors' explanation is that this GIH symmetry breaking is the step that determines singularity, and the post-translational OR protein-dependent steps enforce singularity after the fact, once it's established. This is certainly plausible, but the authors seem to conflate their model with the data, which in my mind do not address the role of OR mRNA in the process, one way or the other. Characterizing the changes in chromatin architecture associated with OR gene expression certainly provides the basis for some interesting models. While a technical tour de force on many levels that provides intriguing details about the organization of OR chromatin architecture, the present study nonetheless falls short in illuminating the mechanistic basis of OR singularity.

Minor but important concern: I found a few errors in the figure legends (missing references to panels) and would encourage the authors to review and edit them for clarity. In addition the font sizes of some of the labels are so small as to be unreadable at a magnification corresponding to normal print size.

Author Rebuttals to Initial Comments:

We would like to thank the 3 reviewers for their thoughtful and constructive critiques. We were pleased that reviewers found our experimental approach “a tour de force effort” (reviewer 2), with “an impressive array of technically dizzying single cell genomics” (reviewer 3), that produced “an elegant and important new mechanism that explains a long-standing enigma” (reviewer 1). Reviewers raised important points, had reasonable concerns, and suggested insightful new experiments and analyses of the existing data that made the paper significantly stronger and more interesting. I hope that you and the reviewers will appreciate that we did everything humanely possible to address every concern with new experiments or additional analysis. In brief, our revision involves new computational analyses of our multiome and Dip-C data that highlights the remarkable lack of deterministic patterns in the combination of olfactory receptor (OR) enhancers that remain accessible in each neuron, and the lack of stereotypic patterning between enhancers converging in the active hub and the identity of the chosen OR allele, highlighting the remarkably probabilistic nature of OR gene regulation (revised figures 1 and 2). Further, with a new genetic experiment, we identify a population of olfactory neurons that transiently deploy two independent enhancer hubs and transcribe two OR alleles, before switching to a stable utilization of a single hub and transcriptional singularity, providing direct support for the model of symmetry breaking (revised Figure 4). Below please find our point-by-point response to every issue raised by the 3 reviewers.

Referee #1 (Remarks to the Author):

Lomvardas and colleagues aim to address a fundamental question about how olfactory receptor neurons transcribe only one of ~2000 distinct olfactory receptor genes. This olfactory receptor choice model represents a paradigm example of allelic competition where stochastic choice leads to a quantitative and deterministic outcome that is essential for function. Yet, what mechanisms exist to explain such random counting problems in olfactory receptor choice or other contexts is still largely unknown. To address this, the authors generate a comprehensive single cell atlas of 3D genome conformation changes and chromatin accessibility that culminates in monoallelic olfactory receptor (OR) gene expression. Olfactory receptor genes organize around enhancer dense regions called “Greek Islands” (GI). The authors show that the active GI localizes specifically with the active OR, whereas inactive GIs form less specific interactions with inactive ORs. To explore how competition of protein factors between GIs might impact stochastic choice, they present a physical model whereby concentration leads to a phase transition and symmetry breaking driving accumulation over a single GI. The authors explore the idea that the olfactory receptor pre-mRNA might be required for this symmetry breaking. They test this using transgenes with coding and frameshifted ORs to distinguish between the RNA and protein products. Based on these observations, the authors propose a model whereby the RNA transcribed from the olfactory receptor gene acts to select the monoallelically expressed allele and silence the other olfactory receptor alleles.

Overall, this paper and the proposed model represents an elegant and important new mechanism to explain a long-standing enigma in gene regulation. The idea that the RNA product of a protein-coding gene can play a direct role in symmetry breaking – reinforcing activation of itself and ensuring silencing of the other alleles – is a novel mechanism that will have important implications far beyond the olfactory receptor system. While I am extremely enthusiastic about the concepts presented in this work, I do have some concerns about how this paper is presented and some of the technical aspects of the experiments that I think will be important to address in order to fully demonstrate this model.

As currently written, the paper appears to represent two distinct parts that do not obviously connect. The first part (Figures 1-3) that focuses on the single cell atlas of structural changes and enhancer interactions between the GIs and the second part (Figures 4 and 5) which focuses on the physical model and the role of RNA in monoallelic expression. While the authors may have some logic of why they felt the former components were critical in leading to the model and the RNA components, it was not obvious to me in reading the paper why these descriptions of GI dynamics were relevant to this mechanism or how they connected. None of these specific dynamics were utilized in establishing the model or exploring the RNA role.

Reviewer 1 raises valid points. Additional analyses in the revised version of the manuscript clarifies why the first part of the paper informs the experiments in the second part. By uncovering the absence of stereotypic enhancer patterns in each mOSN, we exclude the possibility that the presence of certain enhancers gives *a priori* advantage to one of the hubs assembled during differentiation. Thus, lack of stereotypy implies that GI hubs are initially equipotent, supporting the “symmetry breaking” model presented in the second half of the paper. Further, with the new genetic experiment of “forced” switching from *mor28* to *P2*, we demonstrate the transient existence of two active hubs, and the rapid transition to a single transcriptionally engaged hub, further supporting a symmetry breaking process. Importantly, symmetry breaking cannot be explained by the feedback elicited by *mor28* and *P2* proteins in the ER, because this signal should be agnostic to the OR gene sequence that produces the OR protein, and, thus, should have the same effect for both hubs. Thus, the simplest explanation for this “counting” mechanism that tolerates only one active hub per OSNs is an RNA-mediated symmetry breaking process, which leads to our critical observations in figure 5.

Related to this point, it was also not obvious how they went from the proposed model to the RNA hypothesis that they tested. Based on my understanding of the model as presented, there is no “RNA component” included. As such, how the idea of GIs serving as a scaffold for a self-aggregating protein led to them testing a role for RNA is unclear. Indeed, the Figure 4C illustration suggests that it’s the GI hubs that drives aggregation/symmetry breaking. More generally, the authors never test the predictions of this model beyond the gradual silencing of all but one GI. The central mechanistic claims of the model, including whether there is a concentration-limited protein factor, whether this protein factor changes affinity in a local hub, or whether this factor is essential for symmetry breaking, remain untested. As such, it is not clear to me why this model is presented in the Results section since it is not directly tested and does not directly relate to the previous mapping observations or the subsequent RNA hypothesis.

An alternative approach would be to explore this model (which is an adaptation of the symmetry breaking model presented by Nicodemi and Prisco 2007 in the context of X inactivation) in the Discussion as a potential explanation linking their observed 3D genome and RNA results. This would allow the authors to more thoroughly explore/highlight many of the remaining unknowns that still need to be determined and tested. For example, how the RNA component might actually lead to symmetry breaking. Based on my inference from their model, there is presumably a missing protein factor that the RNA likely binds to. What is this protein factor? While I understand that asking the authors to find this protein and characterize it is beyond the scope of this paper, they should certainly explain what some properties of such a protein might be (e.g. RNA binding, ability to undergo concentration-dependent phase transition, etc.) and note the need to identify these components in future work.

Reviewer 1 is correct on this assessment. We moved this model to the discussion session. We agree that this strengthens the paper and streamlines the progression of the experiments.

Finally, I have a technical issue with the authors' claim that the non-coding transgene experiment demonstrates the centrality of the RNA to OR choice. The authors perform a series of clever experiments to distinguish between the RNA and protein products. The key experiment uses a Tet-inducible OR allele that they know is preferentially selected when overexpressed to make a version of this allele that impacts protein production but leaves the remaining sequence otherwise mostly intact. While the non-coding allele leads to lower expression of other OR genes, it does not lead to Tet-independent expression of the transgene itself. If indeed the RNA is critical for monoallelic expression, the non-coding transgene should reduce expression elsewhere and sustain expression of itself. The authors hypothesize that the encoded protein is likely required for sustained expression of the active OR gene, if not for its initial selection. This is certainly a reasonable explanation, but it is never actually tested. Without this demonstration, the authors have not actually demonstrated that the RNA can drive monoallelic choice. One simple experiment that would address this is to express the OR protein exogenously (e.g. by mRNA or protein transfection) alongside the mutant allele and see if this causes sustained expression of the non-coding OR transgene. There might be other experiments to address this that the authors are probably better suited to define in their system and I would be amenable to any that would address this concern.

Reviewer 1 proposes an elegant and important experiment. However, in this *in vivo* system it is practically impossible to deliver the OR protein without producing the OR coding RNA in the nucleus, either as a transgene or via viral infection. Viruses that efficiently infect olfactory neurons have a life cycle of nuclear RNA transcription that is pre-requisite for OR protein translation in the ER. RNA viruses that could allow us to only translate OR RNA without nuclear transcription (for example SARS-CoV-2) do not infect olfactory neurons to our knowledge. Therefore, it is impossible, currently, to conduct this clever experiment.

Beyond these technical difficulties, we would like to clarify the contribution of OR RNA vs OR protein. We do not doubt the importance of a protein mediated feedback in stabilizing singular OR transcription. However, as shown for OSNs that transiently have both P2- and *mor28*-containing active hubs (new figure 4), a protein signal from the ER would not distinguish between them, and it would either lock them both in an active state or dissolve them. In contrast, if local RNA synthesis breaks the symmetry between the two competing hubs first, then the OR protein-elicited feedback would act upon already established singular transcription, stabilizing it for the life of the neuron. This explains why induction of the sterile P2 allele, in *gg8-tTA>tetOP2(nc)* mice, silences all the other ORs while P2 transcription is reinforced by tTA, yet it does not result in tTA-independent expression because the protein feedback is never induced. In fact, this is the reason for which the non-coding function of OR mRNAs had been previously missed by our field; mutations that prevent OR protein expression in endogenous OR genes that cannot remain active by tTA result in switching and rapid transcriptional termination of sterile OR genes, preventing the realization of a protein-independent process for transcriptional singularity. We have clarified this issue in the revision, hopefully to the reviewer's satisfaction.

Referee #2 (Remarks to the Author):

In a tour de force effort, Pourmorady et al address an important unanswered question in genome biology, of which the answer will have broad impact across multiple biological systems. Specifically, in olfactory sensory neurons (OSNs) a single olfactory receptor gene must be chosen and expressed in each neuron, and this requires the selection from 126 possible enhancers (so-called Greek islands – GIs) distributed across the genome. In post-mitotic OSN progenitors, multiple OR genes are expressed, and then an unknown mechanism later in OSN maturation kicks in to ensure that only one OR allele is expressed for the remaining lifetime of each given neuron. In this work, the authors chase down the answer to this mechanism using multiple bleeding edge single-cell and ensemble sequencing technologies as well as biophysical modeling. The impact of the work is clearly at the level appropriate for Nature, and the improvements for revision can be focused on gathering critical multimodal single-cell data to test the authors' compelling hypothesis. Below we provide comments and ideas which we hope are useful to the authors in revising the work.

Comments

>>Figure 1 text is well written and important conclusions are shown. First, using multi-modal single-nucleus ATAC-seq and RNA-seq in the same nucleus for neurons from the main olfactory epithelium (MOE), authors show that: (1) Greek island enhancers are much more variable in accessibility across a cohort of single mOSNs compared to conventional enhancers. (2) GI enhancers exhibit the highest accessibility during the INP \diamond iOSN developmental stage when multiple OR alleles are known to be expressed. (3) Overall summed GI enhancer accessibility declines substantially in mOSNs during which the OR single allele choice is maintained. Overall, the data together are consistent with the authors' proposed model that the reduction in total summed enhancer accessibility (pruning). A bit of

additional re-analysis will help to further support the concept that sustained accessibility of a single enhancer/GI across single cells coincides with single OR allele choice.

1. Figure 1a- could clusters be determined only from the RNAseq, thus allow pure clusters from transcriptional markers and then pure analysis of ATAC at promoters and GIs in 1b and following figures? The reason this might be helpful is that ATACseq signal at promoters is often not a correlative read out of expression, and ATAC signal at enhancers is only predictive of enhancer activity in key developmental contexts (as might be the case in the OSN allele choice system). Minor: It might be helpful to explicitly state the arrow is the direction of lineage within the figure caption.

We agree that clustering cells using both RNA and ATAC modalities may produce different cell cluster assignments than if we were to use RNA data alone. We perform cell clustering using a weighted nearest neighbor (WNN) analysis as described in Hao et al. 2021¹. For cell clustering by WNN analysis, the relative information content of both RNA and ATAC modalities is used to generate cell cluster assignments. Hao et al. find that WNN analysis leads to substantial improvements in accurately characterizing cellular diversity, an important requirement for our analysis given that the olfactory neuronal lineage contains many cell types of relatively low abundance. Furthermore, we would like to clarify that while Fig. 1B and 1C are based off of the mOSN cluster assignment generated by WNN analysis, all panels downstream are purely based off of a pseudotime trajectory constructed from the RNA data alone.

To address Reviewer 2's point, we re-clustered the cells using the RNA data alone (Seurat Clusters) and generated a UMAP projection of the gene expression data (Rebuttal Fig. 1a) and compared it to the WNN clustering and graph presented in the paper and (Rebuttal Fig. 1d). We find that both WNN and Seurat clusters follow similar patterns and separate cells with similar specificities (Fig. 1 b, c, e, f). While mOSNs fall into one cluster, Cluster 0, by WNN, they fall into two clusters using RNA alone, Clusters 0 and 6; and both clusters highly express *Omp*, an mOSN identifying gene. We then repeated our analysis of accessibility frequency in mOSNs and included OR promoters. We find that the patterns of accessibility we observed for mOSN cCREs and GIs to be the same, and OR promoters are almost completely inaccessible, which is the same both for RNA- and WNN-based clustering (Rebuttal Fig. 1g, h). While we appreciate the nuances of the different clustering methods, for the scope of this manuscript they reach the same conclusion, thus we opted to keep our original approach.

Rebuttal Fig. 1 MOE Multiome Revision Analysis (a-c) UMAP projections of snATAC/snRNA data clustered using RNA data alone (Seurat), showing (a) RNA produced cluster assignments, (b) weighted nearest neighbors (WNN) clusters assignments projected onto Seurat UMAP, (c) RNA counts of OMP expression. **(d-f)** UMAP projections of snATAC/snRNA data generated by WNN neighbors analysis, showing (d) Seurat cluster assignments projected onto WNN UMAP, (e) WNN produced cluster assignments, (f) RNA counts of OMP expression. **(g-h)** Percent accessibility of mOSN cCREs, Greek Islands, and OR promoters was performed in mOSNs clustered by RNA alone (g), and multimodal RNA/ATAC data (h). **(i-j)** Lineplots depicting the average accessibility per Greek Island (i) or per OR promoter (j), per integer pseudotime.

2. Figure 1b is rather exciting – readers would benefit from a supplementary figure containing 1b implementation for all 126 GI enhancers as well as for the promoters of all OR gene alleles. Annotating 1b to be clear it is only in mOSN cell type and not for the earlier lineages would be helpful.

We provide a supplemental figure 1 and we modified the text according to the reviewer's suggestion. We worry that putting all 63 GIs and 79 cCREs may be too many panels on one page and therefore selected 20 of each at random to incorporate in Fig. S1d-e.

3. It would be useful and informative to see the same type of Figure 1e in the supplement but for a line plot for every GI and OR promoter overlaid in the same 1-2 plot(s). Also, if this format replaced Figure 1f, there might be less mental overhead of seeing the trends match the average/sum in Figure 1e.

We made a mockup figure of accessibility per GI or OR promoter in a lineplot format (Rebuttal Fig. 1i, j). While the figure does also demonstrate that the pattern of accessibility observed for aggregated GIs/OR promoters is also observed on an individual level, we worry that having so many superimposed lines on such a small plot may be too busy. We could include it in the manuscript, but we hope that the format for Fig. 1E is satisfactory.

4. Do the colors (blue, green, red) match the colors in the line plot & marker genes in Figure 1d and 1e? I would have expected green iOSNs to be placed prior to blue mOSNs in pseudotime in Figure 1e.

We have fixed the colors so that they are consistent with the colors from our pseudotiming.

5. The critical concept in 1g-i is that the computation involves cumulative/summed accessibility. In pink, the accessibility over the top 71 GIs is shown as plateauing in the last mOSN stage – yet in blue the same pattern is shown only for the top OR. Is the idea that these 71 enhancers regulate 71 different ORs across this cellular population (1 GI per 1 OR per cell – many cells)? If so, it might be more relevant to show in blue the plot for the top ORs that would be relevant for these 71 GIs.

We apologize for creating confusion; the plateauing enhancers in pink and red were general cCREs and Lhx2/Ebf-bound regulatory elements, respectively, that are not involved with OR gene regulation, while in each graph we keep GIs (the OR-regulating regulatory elements) in black for reference and direct comparison. The point of these graphs is that GIs have a unique pattern of accessibility fluctuation that is distinct from all the other potential enhancers in this population, even if they are bound by the same transcription factors.

In regards of the other point raised by Reviewer 2, whether GIs are also involved in stabilizing OR expression during the developmental process of OR choice is not known. OR expression patterns transform over the course of OSN development (Hanchate et al. Tan et al.). In OE stem cells, OSNs express no ORs. As cells develop into INP stages and iOSNs, they can express up to ~11 ORs simultaneously all at low levels. Finally, OSNs choose a single OR to express and differentiate into mOSNs; this OR choice coincides with a strong ramp in expression of a single OR. Since the definitive transcriptomic marker for choice is an increase in expression of the most highly expressed OR, we believe that the best way to contextualize our understanding of Greek Island genomic transformations during this process is to superimpose GI accessibility dynamics over the average expression of the most highly expressed OR per cell over pseudotime.

To clarify this transition, we also plotted in the revised version cumulative distribution of OR promoter accessibility. While there are several promoters accessible in the progenitor cells, there is a sharp decline in cumulative OR promoter accessibility that reflects the fact that only a single OR promoter is accessible in every OSN, supporting expression of a single OR.

Moreover, there is no 1GI-1OR pattern in this system, as 6 GIs, on average, converge for the activation of a single OR (Figure 2g).

6. Since this particular data is at single cell resolution, can the authors show that in each cell there is a top GI and a top OR that correlate with high accessibility in mOSNs – and that this is different in each cell? To me, this seems like a small reanalysis which would have outsized exciting impact on the paper.

This suggestion inspired a new analysis that demonstrates the lack of determinism between GI accessibility or GI hub patterning and OR expression. Beginning with our multiome, we used our 1,010 mOSNs to generate ~1M unique cell pairs that were tested for overlap in their accessible GI repertoire and their OR expression. With this analysis we were able to test the following question: are OSNs that share greater overlap in their combination of accessible GIs more likely to also be expressing the same OR? We find that while some cells can have up to 12 GIs in common, ~75% of cell pairs have no overlap in their accessible GIs (Fig. 1l). When we then measure the complexity of OR representation among cell pairs, we observe that OR complexity nearly always matches cellular complexity, independent of the number of GIs in common (Fig. 1m). These new data add further support to our conclusion that the combination of accessible, active GIs does not determine OR choice.

We performed a similar version of this analysis in our Dip-C data. We compared within and between P2 and mor28 expressing cells to determine if the composition of *trans* GIs in their active GI hubs varied based off the OR being expressed. We observe that cell pairs expressing the same OR vs. different ORs are just as likely to have same combinations of GIs in common, suggesting that the patterning of *trans* GIs in the active GI hub itself also does not dictate OR expression (Fig. 2e). From our Dip-C and multiomic analysis we conclude that OR choice is not linked to the combinatorial identity of GIs in a hub, and that OR alleles appear to be indiscriminately utilizing *trans* GIs that happen to be in their proximity to support their expression. We agree with Reviewer 2 that this re-analysis has outsized exciting impact on the paper and we are grateful for this suggestion.

>>Figure 2 – Using elegant transgenic models, the authors conclude that for hubs of enhancer-to-OR gene contacts the number of hubs per cell and the size of hubs (number of GI enhancers per hub per cell) does not change whether the OR gene is active or inactive. They also conclude that cis contacts between the OR gene and GI enhancers are stronger in hubs when the OR genes are active vs. inactive.

7. Readers would benefit from a different resolution for panel Figure 2c – difficult to see text and data.

We made appropriate changes.

8. The 5 particle limit was based on imaging data that suggests that enhancer-promoter distances below 300-400nm correlate with transcription. A supplemental figure showcasing this relationship would be very helpful for the readers.

We used some of the published work from the Levine/Gregor laboratories, who monitor transcription bursting in relationship to enhancer-promoter distance by live imaging, to draw this conclusion. However, we agree with Reviewer 2 that there is uncertainty on the actual mode by which enhancers activate transcription, thus we removed this justification. That said, beyond 5 p.r. active and inactive OR alleles are indistinguishable in regards of GI distribution, thus we reasoned that 5p.r. is the limit for transcriptionally meaningful contacts by Dip-C.

9. The writing on the cellular populations was not crystal clear for a generalized audience and would benefit from a schematic. I could gather that there were two OR genes under consideration – P2 and mor28. After reading the description of the genetic system several times, I could guess that mOSNs expressing either of these 2 OR genes were labeled with GFP via an IRES. For mor28, it is a naturally highly expressed OR gene and GFP+ cells could be sorted. For P2, it is tet inducible and the P2 OR allele remains on long after tet is removed, so the mOSNs with this gene on stay GFP positive and can be sorted. The breeding to another mouse genetic background allows alleles to be separated. As a reader, it took a lot of time to think through this independent from the brief text, so revising this part for clarity on the models might keep readers engaged to keep reading the rest of the paper.

We apologize and we generated a new schematic explaining the experimental pipeline (Supplemental Figure 2).

10. It was not fully clear what should be deduced from Figure 2c – is it meant to claim that a GI enhancer on chr14 is physically proximal (proximity score as predicted by the polymer model) to the mor28 gene in the majority of sorted cells with the mor28 gene expressed? And the GI enhancer on chr7 is physically proximal to the P2 in the majority of sorted cells with the P2 gene expressed? I think I would have benefited from knowing more clearly if this plot was measuring proximity between the one enhancer and the OR gene allele of interest – or if it was measuring proximity among multiple GIs on that chromosome and not assessing contact with the OR gene at all. This figure would increase in value if the same plots were made for a negative control of 70-80 sorted cells that are not GFP+ -- they should not have the pattern shown in Figure 2c. Would the signal on chr14 and chr7 be gone completely or only reduce in the number of cells?

The point of Figure 2c was to demonstrate that in each P2 and mor28 expressing OSN, *cis* enhancers are stereotypically proximal to the active OR allele whereas *trans* enhancers exhibit a probabilistic pattern of convergence on the active OR. Reviewer 2 is correct, however, in asking us to analyze these distances in a negative control population. We have revised Figure 2C and added negative control data in order to clarify the purpose of this figure by using mOSNs that are not selected for specific OR expression (Fig. 2c, d). This mixed mOSN population serves as a good negative control because P2 and mor28 would be expressed in 1/1000 cells. This control clarifies that in the case of P2, frequent proximity with *cis* enhancers correlates with active transcription. For mor28, however, where the closest enhancer is 40Kb apart, we detect the *cis* enhancer near mor28 also in the negative control. Distant *cis* and *trans* enhancers approach the mor28 allele more frequently in the active state compared to our negative control data.

11. The purpose of Figure 2d might be achieved more directly by showing the full distributions of the number of active vs. inactive hubs per cell. I struggled to interpret data showing the distribution only within or above a certain average threshold – I'd prefer to see all of the data and the comparison across conditions. Would like to see explicitly the metrics of Number of Hubs per cell and Number of Enhancers per Hub per cell – full distributions for both Inactive and Active hubs.

This is a great suggestion; we included this analysis in revised Figure 2g.

12. The text for this section was a bit difficult to parse in terms of weighing it against the figure. My understanding is that the main conclusion is that there is very similar distributions of the number of enhancers per hub/cluster per cell whether the hub is inactive (i.e. with an inactive OR gene) vs. active (i.e. with an active OR gene). However, when I look at panel 2h/2i – the inactive OR genes are not forming contacts with inactive GIs – so is there any evidence for inactive GI – inactive OR hubs from these data? All I could gather is that 2f shows inactive GIs are together, 2f shows inactive ORs can be together, but 2h-1 shows inactive GIs and inactive ORs are actually not in hubs. Could this be clarified to support the claims with a bit of tweaks in the analysis?

We apologize for the potential confusion that may have been created in this part of the text. The core of Fig. 2f-m lies in the distinction between GI-OR topology (modeling) and GI-OR physical interactions (contacts). For the purpose of this figure, we define hubs topologically, as ~5 p.r. wide foci of GI clusters (Fig. 2h). We verify that for the unique GI hub containing the active OR, GIs form strong contacts with each other (Fig. 2i, top left) and with the active OR (Fig. 2j). However, we also observe many other GI complexes per cell (Fig. 2f, g) which are presumed inactive, since they are not in contact with the single OR allele that is transcribed in this OSN (we sorted OSNs based on the singular expression of P2 or mor28 and transcription at this stage is strictly monogenic and monoallelic). If we select an inactive GI hub containing the same or similar number of GIs as the active GI hub in every cell, we observe that these GIs will again form strong interactions with each other, like in the active GI hub (Fig. 2i, top right). However, unlike active GI hubs, GIs in inactive hubs will only form very weak contacts with any of the OR genes that they appear to contain from the model (Fig. 2k, l). So, while the GI-OR spatial relationships within the active and inactive GI hubs are defined to be the same by their size and GI density from

Dip-C modeling (Fig. S4b, c), they do not reproduce the same frequency of physical interaction as detected by scHi-C. Herein lies the heart of Figure 2: while there are many GI hubs per mOSN nucleus, some with remarkable similarity to the active GI hub, only the active GI hub has the distinct ability to form exquisitely strong and specific physical interactions with its resident OR allele, being the active OR. We hope that with this explanation added to the text we will clarify any confusion the reader might have. We would like to also reiterate that the discrepancy between the model and HiC contacts in the inactive GI hubs does not reflect a weakness of the model; we have done numerous positive and negative controls, and predictions from the model faithfully recapitulate HiC contacts, as described in the original DipC paper.

13. How are inactive GIH determined in DIPC data since it is not multimodal? Is it assumed that only one OR gene can be active (the sorted population GFP allele) – so all other GI-OR contacts can automatically be inferred to be inactive?

Reviewer 2 is correct. OR expression is monogenic and monoallelic. By sorting a pure population of OSNs expressing a known OR from a known allele, we can by definition locate the active GIH in every cell, given its known physical association with the active OR locus. On the other hand, GIHs that exclude the active OR allele are assumed to be inactive. GI-OR contacts examined in Fig. 2h-i are only measured in inactive GIHs and by proxy also inferred to be inactive.

14. Figure panels 2h/i are not labeled in the 2g caption.

We apologize and we corrected the captions.

15. Figure 2j – add yellow sphere as inactive GI enhancers to key – I did not see congruence between the drawn model and results from Figure 2g-i. At first glance, inactive ORs are not in hubs with inactive GIs?

This figure panel, now Fig. 2m, has been edited. Our Dip-C modeling does show that inactive ORs exist within inactive GI hubs, while our Dip-C contact analysis demonstrates that they do not form contacts with these GIs that are as specific or as frequent, as the contacts between the active OR and the GIs of its hub.

16. Supplemental Figure 2: One thing I am curious on is why it is first needed to project DIP-C onto a polymer model to identify active and inactive hubs? Is it not possible to identify hubs directly from the Dip-C contact matrix? It would be excellent to see and observe the same hubs directly via the Dip-C contact matrix – it would be useful to clarify in the text if this is not possible in single cell data and the intermediate of the polymer model is required for transitioning a sparse matrix to insight.

Like most single cell genomics, scHi-C data is inherently sparse and noisy. With limited genome coverage, single-cell contacts alone could not well be used to examine nuclear architecture at high resolution. However, by leveraging Dip-C technology, sparse single cell contacts are input for reconstructing 3D-models which map all observed and inferred genomic interactions at 20-kb resolution. In Dip-C, 3D reconstruction of the genome is based off a bead-and-spring model of chromatin folding and has been validated to regenerate observed features of the 3D genome (Tan et al., Stevens et al). After multiple rounds of unsupervised 3D-reconstruction, only nuclei with low median root mean squared deviation of 3D genomic positions are included in our analysis, leaving cells that reliably generate the same Euclidean genomic relationships after 3D-reconstruction. By priming our study of single-cell contacts with an understanding of likely conformations of the entire genome in 3D-space, we can focus our analysis on only interactions that are occurring within GI hubs reducing the effects of noise and sparsity from single-cell contacts and spurious GI-OR interactions on our analysis. We agree with Reviewer 2 that this is an important point to clarify and we made it obvious in our revision.

>>Figure 3 – the overarching conclusion is that hubs with active OR alleles are enriched in H3K27ac, whereas hubs with inactive OR alleles are enriched in heterochromatin.

17. Can a single-cell H3K9me3 Cut and Run assay confirm that heterochromatin is enriched at GI enhancers when they are in a hub with inactive OR alleles? This would be a direct test of the hypothesis. There might be a technical barrier to doing this because you would need a multi-modal assay to do single-allele gene expression and heterochromatin ChIP in the same cell. If so, it would make sense that this is not feasible at this time.

This is an excellent suggestion but we are not currently equipped for this analysis.

>>Figure 4 – the conclusion and model from Figure 4 is perhaps the most important of the paper. The model asserts that GI enhancers are ubiquitously activated at a low level along with sporadic OR transcription in the INP stage, and then subsequently a few choice enhancers and OR alleles are upregulated along with multi-enhancer transcriptional hubs in iOSNs. It is then only in mOSNs that many hubs are shut off and the genes silenced, whereas one hub of enhancers and the singular chosen OR allele stays on whilst others are all silenced.

18. The model is compelling and exciting – yet, the data to support is sparse in the current form. This is the one part of the paper that would really benefit from addition of another experimental assay given the sweeping impact of the model.

Please see our response to Reviewer 3, where we address how, with the addition of our OR switching model, we provide a direct observable demonstration of symmetry breaking as a mechanism for OR choice (revised Figure 4d-k). We hope that the addition of this experiment, which required HiC, RNA-seq and ATAC-seq analysis of FAC-sorted neuron of a mouse with 4 different modified alleles, would reinforce support for this model.

19. It is particularly difficult to infer the model from the data shown in Figure 4a. Although not introduced explicitly, I inferred that the authors are showing either scDipC data or new ensemble Hi-C data for these 4 developmental populations introduced in Figure 1a (same cells also used for the 10x multiome analysis). Can the authors make it clear what data was generated and used to make these images? Did they go back to the 4 cell types from Figure 1 and create more DipC libraries?

We have revised Fig. 4 and introduced a modified Fig. S6 to help address Reviewer 2's concerns. Fig. S6a-d is a combination of previously published (HBC, INP, mOSN) and new (iOSN) bulk Hi-C data demonstrating that (1) GI contacts initially form in INPs, (2) reach full strength in iOSNs, when OR expression is still polygenic, and (3) finally stabilize in mOSNs once OR choice has been made. To contextualize our understanding of GI interactions over development in single cells we took advantage of a previously published Dip-C data set, made by the creator of Dip-C². Tan et al. perform Dip-C in newborn and adult MOE, thus generating a Dip-C library with cellular coverage over the entire olfactory neuronal lineage. They demonstrate in their paper that by performing PCA on vectors of genome compartment score, for all cells in their study, they are able to create an axis for neuronal development across PC1. They validate this finding by multiple methods, one being that they take published bulk Hi-C data from our lab, generate the same vectors of compartment score, and project it onto their PCA.

We leveraged this analysis to comment on the nature of GI topologies across cellular stages. We projected the Hi-C data from Fig. S6a-d onto Tan et al.'s Dip-C data set and measured the quantity and density of GI hubs per cell across development (Fig. S6e, f). We find that during the time period when OR expression is polygenic, GI hubs are forming and concentrating themselves with more and more GIs (Fig. S6f). However, once choice has occurred, there is little change to GI architecture generally. These findings are validated by DNA FISH data showing GIs concentrating themselves into multiple GI hubs over development. We hope that with the addition of this new supplemental figure, any confusing regarding the nature of our model in Fig. 4a will be cleared.

20. Figure 4a and 4b – these two analyses do not help support the models because we cannot parse inactive OR-inactive GI hubs from active OR-active GI hubs, nor active GI-GI hubs from inactive GI-GI hubs. Yet, if this was explicitly done, it would transform the paper and back up the conclusions. There is indeed a technical challenge of gathering the data to support this claim – as it would require allele level gene expression in the same cell as single-cell Hi-C like measurements between all GI enhancers and all OR gene isoforms in an allele-specific manner. Since this model is very important to the paper: Can the authors pursue a multi-modal genomics or imaging technology to drive these conclusions home? For example, sequential Oligopaints for a few OR alleles and the GI enhancers coupled with merFISH for the RNA of those chosen OR alleles? If the authors deem that a bulk multiomic technique might be useful to gather the data support – Hi-Car is another option: <https://www.sciencedirect.com/science/article/pii/S1097276522000983>

21. If the authors deem that conducting multimodal techniques is out of scope, the wording can be revised to indicate that Figure 4A is a model that is of high impact yet cannot be fully tested to conclusion at this time in science and technology development.

22. The authors present the intriguing idea "at the earlier stage of polygenic OR transcription when Lhx2 and Ldb1 concentrations are highest (Fig. S5a) each of the multiple hubs may have similar protein composition. As the concentration of these two factors decreases with differentiation, the symmetry between hubs becomes broken, with Lhx2/Ebf/Ldb1 accumulation in one GI hub". Multimodal imaging of Lhx2/Ldb1 along with Oligopaints would likely be needed for single cell evidence of equal Ldb1 concentration at all GI-OR hubs prior to single allele choice. Can the wording be revised?

We appreciate the suggestions from Reviewer 2 for an unprecedented deployment of every state of the art technology available in single cell genomics. Obviously, we aspire to perform the experiments proposed in points 20-22. However, given the time consuming and expensive mouse genetics behind this project, we cannot perform these experiments in a single paper. We will, instead, point out weaknesses of our model in the text.

Referee #3 (Remarks to the Author):

The manuscript by Lomvardas and colleagues uses an impressive array of technically dizzying single cell genomics techniques to probe the mechanisms driving the singularity of odorant receptor (OR) expression by mature olfactory sensory neurons (mOSNs). Previous work by this group has demonstrated the presence of ~70 OR cluster enhancers called Greek Islands (GIs) that mediate intra- and inter-chromosomal interactions that organize the chromatin of mature OSNs into discrete clusters or Greek Island Hubs (GIHs) in an Lhx2/Ebf/Ldb1-dependent mechanism. This architecture is required for high-level expression of a single OR per mOSN. A curiosity arises from other studies, which demonstrated that during the transition from immediate neuronal precursors (INPs) to immature OSNs (iOSNs), multiple ORs are expressed at a low level before one is selected for singular, high-level expression in mOSNs. The authors posit that the properties of the GIHs must change during this transition.

Consistent with this hypothesis, ATAC-seq shows peak accessibility of GIs at the INP-iOSN transition, corresponding to the onset of multi-OR expression per cell, with accessibility falling yet maintained at supra-basal levels through the mOSN stage. Using Dip-C to reconstruct the 3D nuclear chromatin architecture, they further find that each mOSN has on average five GIHs: one active hub that includes the OR chosen for expression (experimentally defined, as these were the cells that were purified by FACS), and ~four "inactive" hubs containing GIs in which the active OR allele is absent. Further analysis by Hi-C, liquid Hi-C, and H3K27ac ChIP-seq demonstrate differences in the number of inter- and intra-chromosomal contacts and biochemical properties of active and inactive GIHs.

The above observations raise the question of how the active GIH is ultimately favored over the other GIHs that wind up being

suppressed. The authors conceived of a symmetry breaking model to account for the transition to enhanced contacts within the active GIH over those in the inactive GIHs. On one level, the model seems attractive and plausible enough. However, as intriguing as they may be, it is unclear to this reviewer whether the experiments that follow address the model, at least in terms of specifics. Using the endogenous MOR28 knockin allele or the broad and early-expressed P2 allele, they show singularity of OR expression is maintained either with intact OR protein coding region or with a frame-shifted mutation that abrogates OR protein expression. The number of GIH contact sites appears similar across all cases examined. Consistent with previous studies in the literature, transgene expression of a non-protein coding M71 mRNA results in suppression of other OR expression. From all of the above, the authors conclude that OR mRNA, and not OR protein per se, is required to break symmetry and promote singularity in OR expression. Unfortunately, without knowing the configurations of the GIHs in the M71 mRNA over-expression experiment, it cannot be concluded that this effect is due to symmetry breaking at the level of GIHs. Missing is an experimental manipulation that results in persistent polygenic OR expression. This is a critical point that either makes or breaks the interpretation of the final set of experiments in Figure 5. From the data presented I just don't see an incisive experiment that allows the authors to draw their conclusions about mechanism.

It is also interesting to consider how the results in this study relate to the phenomenon of OR gene "switching" that was originally discovered by Sakano and fleshed out mechanistically by this group. The authors' explanation is that this GIH symmetry breaking is the step that determines singularity, and the post-translational OR protein-dependent steps enforce singularity after the fact, once it's established. This is certainly plausible, but the authors seem to conflate their model with the data, which in my mind do not address the role of OR mRNA in the process, one way or the other. Characterizing the changes in chromatin architecture associated with OR gene expression certainly provides the basis for some interesting models. While a technical tour de force on many levels that provides intriguing details about the organization of OR chromatin architecture, the present study nonetheless falls short in illuminating the mechanistic basis of OR singularity.

Minor but important concern: I found a few errors in the figure legends (missing references to panels) and would encourage the authors to review and edit them for clarity. In addition the font sizes of some of the labels are so small as to be unreadable at a magnification corresponding to normal print size.

We would like to start our response with an important clarification: although previous experiments have shown that OR transgenes can suppress the previously chosen ORs, this is the first study ever to show that non-protein coding, sterile OR transgenes can repress OR transcription. We apologize for the confusion, but by clarifying this point we hope that we convince Reviewer 3 on the conceptual advancement of this manuscript, since it provides the first demonstration of non-coding functions of an OR.

Beyond this critical point, we agree that revealing the mechanistic underpinnings of symmetry breaking in this system is important. In the revision we provide a critical new experiment that captures OSNs at the transient stage of GI hub symmetry and biallelic OR expression, followed by symmetry breaking and transition to a single OR-engaged GI hub. We combined 4 genetically modified alleles to generate the following strain: *mor28iCre>tdT fl/+; OMPtTA>tetOP2iGFP*. As explained in the text and Figure 4F, the MOE of these mice will contain 3 fluorescent cell populations: GFP⁺ single positive mOSNs that express P2 (green); tdT⁺ single positive OSNs that express *mor28* (red); and GFP⁺/tdT⁺ OSNs that express P2 but have a prior history of *mor28* expression (yellow). We performed ATAC-seq, RNA-seq, and *in situ* Hi-C in all 3 cell populations to study the genomic and transcriptomic transformations that take place during switching in cell populations where we know both the OR selected for choice (P2) and an OR inactivated from choice (mor28).

As explained in the revised manuscript, we observe that GFP⁺ and GFP⁺/tdT⁺ OSNs possess all markers of a cell having stably chosen P2: a highly accessible P2 locus, robust P2 transcription, and a GI hub around the P2 allele (Fig. 4g, h, i, k). Similarly, tdT⁺ cells also appear to have chosen *mor28*, given the presence of the same three features: an accessible *mor28* locus, robust *mor28* transcription, and a GI hub around the *mor28* allele (Fig. 4g, h, m). Remarkably, although red cells are GFP negative, they have an additional GI hub forming over the P2 locus which also exhibits increased accessibility and transcription above background levels (Fig. 4g, h, j). In other words, red cells have initiated the process of P2 activation by tTA and GI hub recruitment, but they have not reach sufficiently high levels of P2 expression to become GFP⁺. The result is clear: at the onset of switching, we detect two hubs, one transcribing *mor28* at high levels and one transcribing P2 at low levels. Importantly, because *mor28* and P2 do not exhibit elevated contacts with each other in tdT⁺ cells, we are confident that these two alleles are utilizing different hubs (Fig. S7). Once P2 expression increases enough to permit GFP detection (in GFP⁺/tdT⁺ cells), the *mor28* locus experiences a marked reduction in accessibility and falls out of association with its own GI hub (Fig. 4g, n). We hope that Reviewer 3 will appreciate that this experiment provides direct demonstration that GI hub-OR interactions are the genomic contacts most sensitive to the symmetry breaking process. This set of experiments also answers Reviewer's 3 question about OR gene switching.

To address Reviewer 3's second concern:

"Missing is an experimental manipulation that results in persistent polygenic OR expression."

This represents a tantalizing experimental goal. An experiment leading to persistent polygenic OR expression would require finding the presumed RNA-binding factor OR genes are competing for to stabilize their association with a GI hub and to prevent OR transcription from other hubs. Only upon identification and overexpression of this missing factor we would be able to achieve a state of stable and robust polygenic OR transcription. We hope that Reviewer 3 would agree that this is beyond the scope of a revision, as it would take years to identify such factor and to perform the required genetic manipulations that would result in stable OR coexpression. For reference, it took decades to identify such factors for X-chromosome inactivation, despite the existence of cell culture systems that are more tractable than our *in vivo* model.

- 1 Hao, Y. *et al.* Integrated analysis of multimodal single-cell data. *Cell* **184**, 3573-3587. e3529 (2021).
- 2 Tan, L., Xing, D., Daley, N. & Xie, X. S. Three-dimensional genome structures of single sensory neurons in mouse visual and olfactory systems. *Nat Struct Mol Biol* **26**, 297-307, doi:10.1038/s41594-019-0205-2 (2019).

Reviewer Reports on the First Revision:

Referees' comments:

Referee #1 (Remarks to the Author):

The authors did a great job addressing my concerns in the initial submission. In particular, the revised manuscript now more clearly explains the motivations and results and connects these up into a clear and compelling model. While I understand why the authors were unable to perform the protein versus RNA experiment I proposed, the clarification of the model related to this point in the Discussion more than addresses my concern.

One small suggestion for the authors is that they might wish to move the "two possible models" mentioned in the reply and now described at the transition point between the single cell omics and RNA sections into the Introduction to setup the rationale for exploring the genomic structure and framing of the paper. This would also provide the reader some additional context for the current state of knowledge in the field about this question. However, this is obviously a decision I leave to the authors.

I remain convinced that this work is an important contribution that uncovers a beautiful new mechanism in gene regulation. With these revisions, I would be excited to see this published in Nature!

Referee #2 (Remarks to the Author):

Lomvardas and colleagues submit a strong revised manuscript. I have read the rebuttal in detail, and it is clear that the team has rigorously and excellently addressed all of the matters related to minor typos, figure panel errors, and sources of confusion. It is also clear that figures have been upgraded in quality and clarity for a general audience. Although not included for my requests, I noted the new data provided for Review 3 in Figure 4, which, to me, represents a difficult experiment and a substantial contribution to the literature. Through use of elegant triple- and quadruple-transgenic mice, rare sorted neuron populations, and difficult genomics assays, the authors convincingly show that ectopic expression of the P2 allele in iNP/iOSNs or mOSNs can outcompete the existing hub and instead allow the P2-GI hub to dominate and the former hubs to either break apart or silence. This observation alone opens up many new questions and curiosities - a hallmark of work published in Nature.

I still take pause on the data provided for some of the claims as the manuscript progresses to Figure 5 to the end. However, it is true that the way forward for testing the authors' ideas is for bleeding edge multimodal assays that are only in nascent form and not readily available to the international community.

Referee #3 (Remarks to the Author):

The experiments in the new Figure 4 that literally catch cells in the act of OR gene switching elegantly provide compelling evidence that symmetry breaking occurs at the level of the Greek Island hubs. These experiments more logically set the stage for what is now Figure 5, which presents data using the "broken" P2 OR sequence and concludes with the authors' model. For this section of the manuscript, the data horse is now appropriately in front of the model cart. The revised manuscript is much improved and addresses my main concerns about the Greek Island hubs being the locus of symmetry breaking and the logical flow of experiment and model (arguably articulated more cogently by Reviewer 1). The authors are to be commended for so creatively dissecting the complex mechanisms underlying the regulation of the large OR gene family.

Author Rebuttals to First Revision:

We are grateful to the 3 reviewers for finding our study significant, novel, convincing and rigorous enough for publication in Nature. Since there were no additional requests, we do not have any specific responses at this time, other than thanking them for improving the manuscript with their insightful comments.